# EmBrace: A Collective Knowledge Fusion Framework Toward Unified EEG Foundation Models

**Ziyu Jia** [1 2] **Junyi Lin** [3] **Pu Wan** [3] **Jinxin Pi** [3] **Jingying Ma** [4] **Peiliang Gong** [5] **Xinliang Zhou** [5] **Yi Ding** [5] **Chenyu Liu** [5]

## Abstract

Electroencephalography (EEG) foundation models (EFMs) have achieved strong performance across a wide range of downstream EEG tasks via pretraining and fine-tuning. Through empirical analysis, we observe that (i) no single EFM consistently dominates all tasks, yet identifying the task-specific optimal model by fine-tuning all EFMs introduces substantial computational overhead; and (ii) models with inferior task-level performance still exhibit strengths at the sample level as distinct architectures induce diverse inductive biases. These observations motivate EmBrace, a representation-centric framework for sample-aware knowledge fusion that avoids the constraints of parameter-level or output-level alignment. EmBrace synchronizes discriminative intermediate representations into a unified manifold and adaptively weights multiple EFMs at the sample level while selecting the most compatible model as the carrier. Extensive experiments across multiple EEG benchmarks demonstrate that EmBrace consistently improves over SOTA EFMs and generalizes effectively under cross-task settings. Our code is available at GitHub.

## 1. Introduction

Electroencephalography (EEG) is a non-invasive technique for recording brain activity and is widely used in affec-

tive computing (Pillalamarri & Shanmugam, 2025), brain computer interfaces (Värbu et al., 2022), cognitive state assessment (Gu et al., 2021b), and neurological disorder diagnosis (Alturki et al., 2020). Inspired by the success of pretraining paradigms in natural language processing, EEG foundation models (EFMs) leverage pretraining followed by fine-tuning to achieve strong performance across multiple EEG tasks, establishing this paradigm as a dominant trend in EFM research (Zhou et al., 2025a).

Although EFMs perform well on diverse EEG tasks, evaluations on multiple datasets reveal a consistent phenomenon: **(I) No single EFM dominates all EEG tasks**. As shown in Figure 1(a, top), the optimal model varies depending on the task; even the latest state-of-the-art CodeBrain (Ma et al., 2026) fails to achieve consistently optimal performance. Moreover, the substantial parameter count in EFMs makes fine-tuning all models computationally expensive (Tang et al., 2026), while relying on a single model inevitably sacrifices performance compared to task-specific optimal models (Figure 1(b)); even CodeBrain experiences a 26.2% performance loss across multiple evaluation tasks.

Through a deeper analysis, we further uncover a critical observation beyond task-level performance: **(II) EFMs with weaker task-level performance may still exhibit strengths at the sample level**. More concretely, even the model with the lowest task-level performance can correctly classify samples that the overall best-performing model misses (Figure 1(a, bottom)) (Liu et al., 2025). One key reason is that architectural differences among EFMs induce distinct inductive biases. For instance, models that explicitly separate temporal and spatial streams (e.g., CBraMod (Wang et al., 2025)) tend to more effectively exploit the information embedded in samples with strong spatiotemporal coupling, whereas models incorporating state-space modeling or continuous dynamical mechanisms (e.g., CodeBrain) tend to be more effective at extracting information from samples characterized by long-range temporal dependencies (Gu et al., 2021a). These two observations jointly raise a critical question:

***Can we better exploit the strengths of different EFMs to achieve stable performance across EEG tasks?***

---

[1] Beijing Key Laboratory of Brainnetome and Brain-Computer Interface, Institute of Automation, Chinese Academy of Sciences, Beijing, China. [2] Brainnetome Center, Institute of Automation, Chinese Academy of Sciences, Beijing, China. [3] School of Computer Science and Technology, Beijing Jiaotong University, Beijing, China [4] Saw Swee Hock School of Public Health, National University of Singapore, Singapore [5] College of Computing and Data Science, Nanyang Technological University, Singapore . Correspondence to: Yi Ding <ding.yi@ntu.edu.sg>, Chenyu Liu <chenyu003@e.ntu.edu.sg>.

*Proceedings of the 43rd International Conference on Machine Learning*, Seoul, South Korea. PMLR 306, 2026. Copyright 2026 by the author(s).

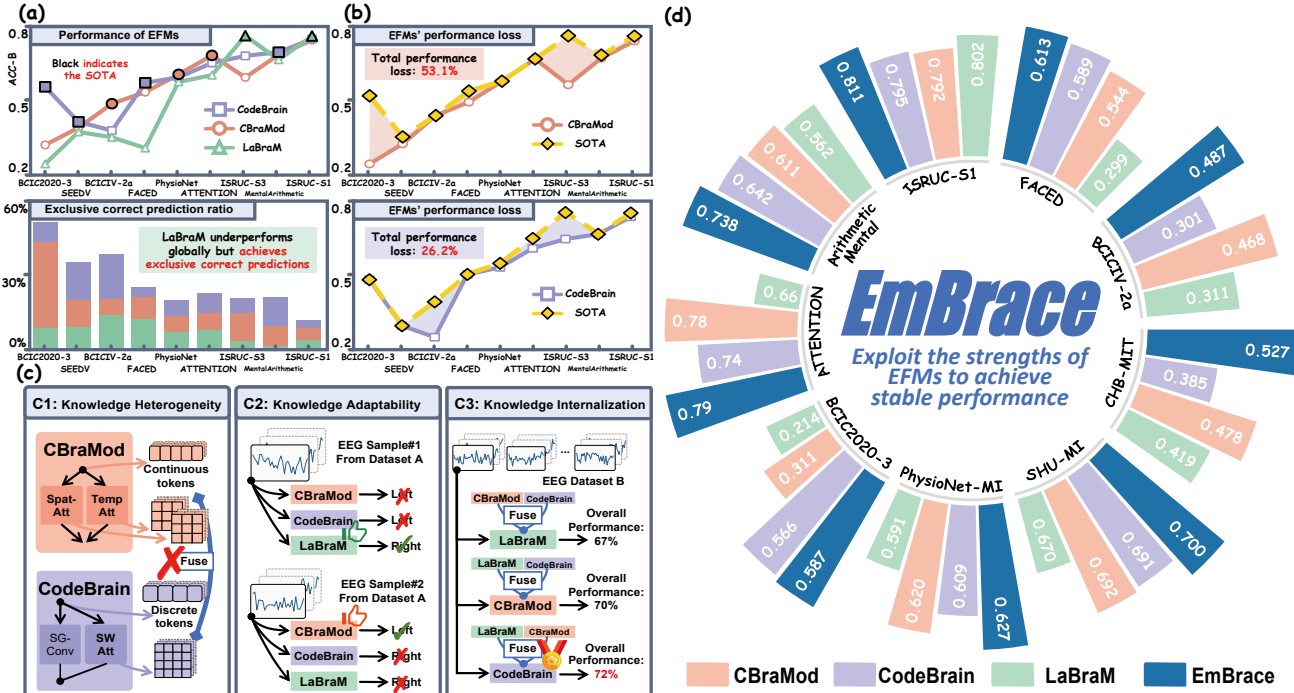

*Figure 1.* **(a, top)** Task-level observations indicating that no single EFM consistently dominates all tasks. **(a, bottom)** Sample-level observations where models with lower performance correctly classify samples that the best model fails to. **(b)** Deploying a single model leads to substantial cumulative performance loss compared to the task-specific SOTA selection. **(c)** Three technical challenges. **(d)** Performance comparison of EmBrace and EFMs.

To address this question, knowledge fusion offers a promising direction. In the context of foundation models, knowledge fusion typically uses intermediate representations as transferable knowledge and integrates them into a unified training framework. However, the architectural characteristics of EFMs pose three key technical challenges in its implementation (Figure 1(c)):

**C1: Heterogeneity among EFMs hinders existing fusion methods.** Existing methods for leveraging multiple models cannot be directly applied. Parameter-level methods (Wortsman et al., 2022; Jin et al., 2022) assume similar architectures, which is incompatible with the diverse implementations of EFMs. Output-level techniques (Moslemi et al., 2024; Allen-Zhu & Li, 2020) require comparable decoder outputs or shared prediction distributions, whereas EFMs often differ in task-specific heads, making direct output alignment ill-posed.

**C2: The effectiveness of different EFMs varies dynamically at the sample level.** Due to sample-level performance differences among EFMs, their contributions vary across samples. However, within the EFM domain, existing approaches still lack an explicit, learnable mechanism to assign sample-dependent weights during training, which often leads to suboptimal utilization of sample-level knowledge.

**C3: Substantial performance discrepancies arise when different EFMs serve as knowledge carriers.** EFMs differ significantly in task adaptability and representational flexibility, making the effectiveness of transfer highly dependent on the chosen carrier (Gou et al., 2021). Therefore, there is a need for an explicit, learnable, and data-driven mechanism to identify the most suitable carrier EFM across heterogeneous EEG tasks and subject-dependent data distributions.

To address these challenges, we propose a new framework for knowledge fusion among EFMs, termed EmBrace. In contrast to strategies based on parameter- or output-level alignment, EmBrace adopts **discriminative information encapsulated in intermediate representations** as a unified form of knowledge. It formulates knowledge fusion among EFMs as a representation-centric learning problem. **For C1,** EmBrace **externalizes** multi-scale intermediate representations from heterogeneous EFMs into a unified knowledge manifold, effectively circumventing architectural heterogeneity at both the parameter and output levels. **For C2,** EmBrace **calibrates** the contribution of each EFM through a sample-aware adaptive weighting mechanism, ensuring that the collective intelligence is dynamically tailored to the intrinsic neural dynamics of each input. **For C3,** EmBrace **optimizes** the fusion efficiency by identifying the most compatible EFM as carrier via a structural resonance-based selection method, thereby maximizing the effectiveness of the

fused knowledge across tasks.

To verify the efficacy of EmBrace, we conduct extensive experiments across 12 diverse EEG datasets covering 8 representative BCI paradigms. As shown in Figure 1(d), EmBrace significantly exceeds the scores of the best-performing individual backbones, establishing EmBrace as a robust and unified solution for heterogeneous EEG analysis that successfully breaks the performance ceiling of single-model deployments.

## 2. Preliminaries

### 2.1. EEG Foundation Model

A raw EEG sample is denoted as $\mathbf{X}_{\text{raw}} \in \mathbb{R}^{C \times T}$, where $C$ and $T$ represent the number of channels and sampling points. It is segmented into $N_p$ non-overlapping patches, yielding $\mathbf{X} \in \mathbb{R}^{C \times N_p \times \lfloor T/N_p \rfloor}$. For a downstream dataset $\mathcal{D} = \{(\mathbf{X}^{(n)}, \mathbf{y}^{(n)})\}_{n=1}^N$, $\mathbf{y}$ denotes the ground-truth labels and $N$ is the total number of samples.

The pre-trained EEG Foundation Model is denoted as $\mathbf{F}(\cdot)$, which maps the input $\mathbf{X}$ to latent representations $\mathbf{Z}_L \in \mathbb{R}^{P_{seq} \times d}$, where $d$ is the embedding dimension and $P_{seq}$ is the total sequence length (comprising $C \times N_p$ patches and potential task tokens). The model structure is formulated as:

$$\mathbf{Z}_L = \mathbf{F}(\mathbf{X}; \theta) = \mathcal{T}\big(\mathcal{I}(\mathbf{X}; \theta_{\mathcal{I}}); \theta_{\mathcal{T}}\big), \quad (1)$$

where the final prediction is obtained via $\hat{\mathbf{y}} = \text{Head}(\mathbf{Z}_L)$. The components $\mathcal{I}(\cdot)$ and $\mathcal{T}(\cdot)$ are further detailed as:

- **Input Embedding Layer $\mathcal{I}(\cdot; \theta_{\mathcal{I}})$:** This module tokenizes input signals into an embedding space:

$$\mathbf{Z}_0 = \mathcal{I}(\mathbf{X}; \theta_{\mathcal{I}}) \in \mathbb{R}^{P_{seq} \times d}, \quad (2)$$

where $\mathbf{Z}_0$ denotes the initial embeddings that encapsulate local signal characteristics within each token.

- **Encoder $\mathcal{T}(\cdot; \theta_{\mathcal{T}})$:** This module captures the latent relational between tokens through $L$ successive layers:

$$\mathbf{Z}_L = \mathcal{T}(\mathbf{Z}_0; \theta_{\mathcal{T}}) \in \mathbb{R}^{P_{seq} \times d}, \quad (3)$$

where $\mathbf{Z}_L$ denotes the refined representations that encode intricate spatio-temporal token interactions.

The parameter set $\theta = \{\theta_{\mathcal{I}}, \theta_{\mathcal{T}}\}$ represents the complete set of pre-trained weights. By initializing $\mathbf{F}(\cdot)$ with $\theta$, the model extracts representations informed by pre-trained knowledge to facilitate various downstream predictions.

### 2.2. Knowledge Fusion for EFMs

To move beyond the limitations of a single pre-trained model and achieve stable performance across diverse EEG tasks, we define a joint optimization objective $\mathcal{J}$ to internalize

collective knowledge from a *model pool* $\mathcal{M} = \{\mathbf{F}^{(m)}\}_{m=1}^M$ into an optimal carrier model $\mathbf{T} \in \mathcal{M}$, where $m$ denotes the $m$-th pre-trained EFM. To optimize the parameter space $\theta_{\mathbf{T}}$ of the carrier $\mathbf{T}$, we formulate the objective as follows:

$$\begin{aligned} \mathcal{J}(\theta^{(\mathbf{T})}; \mathcal{D}, \mathcal{K}^{(*)}) =& \lambda \cdot \mathcal{L}_{\text{task}}(\hat{\mathbf{y}}, \mathbf{y}) \\ &+ \sum_{k=1}^{N_k} \gamma_k \cdot \mathcal{L}_{\text{fusion}}\big(\mathbf{K}_k^{(\mathbf{T})}, \mathbf{K}_k^{(*)}\big), \end{aligned} \quad (4)$$

where $\mathcal{K}^{(*)} = \{\mathbf{K}_k^{(*)}\}_{k=1}^{N_k}$ is the set of fused knowledge references, $k$ denotes the $k$-th scale, and $\lambda, \gamma_k$ are trade-off coefficients. In this formulation:

- $\mathcal{L}_{\text{task}}(\cdot, \cdot)$ is a supervised loss (e.g., Cross-Entropy) that ensures the carrier $\mathbf{T}$ maintains discriminative performance on the downstream dataset $\mathcal{D}$.
- $\mathcal{L}_{\text{fusion}}(\cdot, \cdot)$ is a divergence metric (e.g., MSE or KL divergence) that promotes representation convergence between the internal knowledge $\mathbf{K}_k^{(\mathbf{T})}$ extracted from carrier $\mathbf{T}$ and the corresponding fused reference $\mathbf{K}_k^{(*)}$.

By minimizing $\mathcal{J}$, the carrier $\mathbf{T}$ is optimized to both fit the labels $\mathbf{y}$ and converge towards the collective knowledge fused from the *model pool*.

### 2.3. Technical Challenges

To effectively harness this knowledge synergy, we must address three technical challenges derived from the optimization objective Eq. (4):

**C1: Knowledge Heterogeneity.** *How to extract multi-scale knowledge from non-isomorphic EFMs and unify it into a shared space for collective fusion?* The structural and semantic discrepancies between different $\mathbf{F}^{(m)}$ prevent direct knowledge aggregation. This requires an extraction-unification operator $\Psi$ to bridge these gaps:

$$\Psi : (\mathbf{F}^{(m)}, \mathbf{X}) \to \tilde{\mathbf{K}}_k^{(m)} \in \mathcal{V}_k, \quad (5)$$

where $\Psi(\cdot)$ captures essential neural patterns at scale $k$ and reconciles them into the unified knowledge representation $\tilde{\mathbf{K}}_k^{(m)}$ within a shared manifold $\mathcal{V}_k$. The mechanisms for knowledge extraction and cross-modal unification are elaborated in Section 3.1.

**C2: Knowledge Adaptability.** *How to dynamically quantify the reliability of knowledge from each EFM to construct the optimal reference $\mathbf{K}_k^{(*)}$?* The knowledge of each EFM varies depending on the specific input sample. This necessitates an adaptive fusion operator $\mathcal{G}$ to synthesize the collective knowledge into a consolidated reference:

$$\mathcal{G} : (\{\tilde{\mathbf{K}}_k^{(m)}\}_{m=1}^M, \mathbf{X}_{\text{raw}}) \to \mathbf{K}_k^{(*)} \in \mathcal{K}^{(*)}, \quad (6)$$

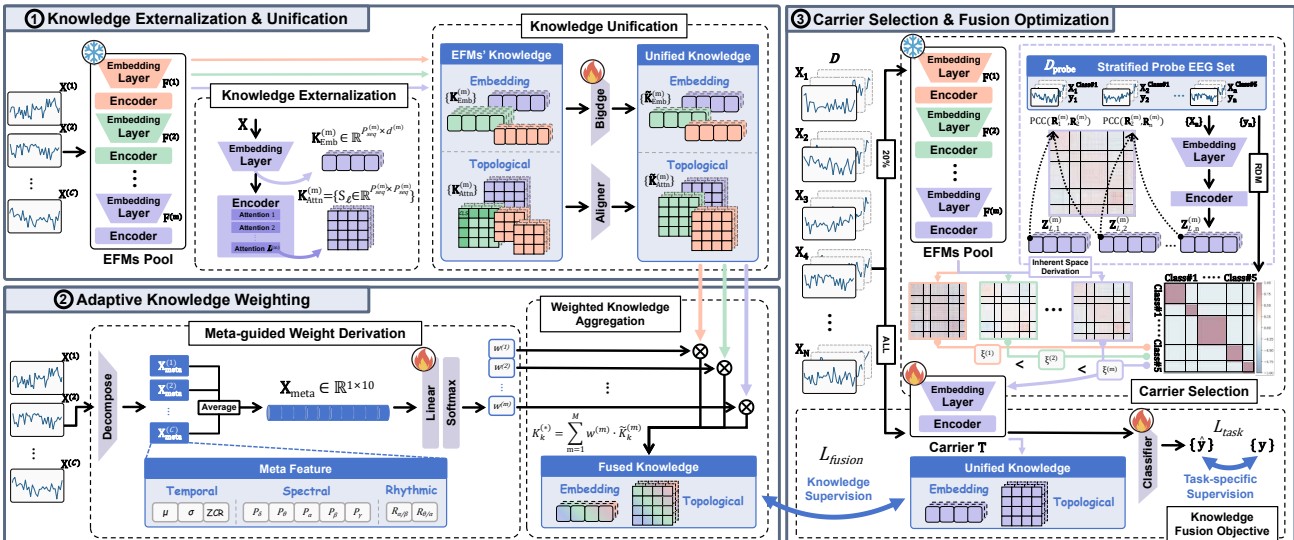

*Figure 2.* Overview of the proposed EmBrace framework. (1) **Knowledge Externalization and Unification** (Section 3.1), (2) **Adaptive Knowledge Weighting** (Section 3.2), and (3) **Carrier Selection and Fusion Optimization** (Section 3.3).

where $\mathcal{G}(\cdot)$ identifies the characteristics of input $\mathbf{X}$ to modulate the contribution of each $\tilde{\mathbf{K}}_k^{(m)}$. The strategy for dynamic knowledge quantification and adaptive weighted fusion is detailed in Section 3.2.

**C3: Knowledge Internalization.** *How to select the most compatible carrier model* $\mathbf{T}$ *from* $\mathcal{M}$ *to facilitate the knowledge convergence?* The efficiency of internalizing collective knowledge is constrained by the carrier's inherent capacity and its architectural fitness. We thus formulate the optimal carrier selection as a scoring and selection problem:

$$Q : (\mathcal{M}, \mathcal{D}) \rightarrow \mathbf{T} \in \mathcal{M}, \tag{7}$$

where $Q(\cdot)$ evaluates candidates within the *model pool* $\mathcal{M}$ to identify the optimal carrier $\mathbf{T}$ that best fits the target dataset $\mathcal{D}$. The selection method and final optimization process are detailed in Section 3.3.

## 3. Method

This section details the EmBrace framework, designed to harness the collective intelligence of heterogeneous EFMs.

### 3.1. Knowledge Externalization and Unification

*Intuition.* Given that heterogeneous EFMs model the same EEG signals $\mathbf{X}$, their internal representations capture the fundamental underlying neural dynamics. This suggests the existence of a common knowledge space where divergent model logics are rendered comparable.

**Lemma 3.1.** Let $\Omega^{(m)}$ be the internal states of EFM $\mathbf{F}^{(m)}$. There exists a sufficient extraction-unification op-

erator $\Psi$ that externalizes and reconstructs these states into unified knowledge $\tilde{\mathbf{K}}_k^{(m)}$ within a shared canonical manifold $\mathcal{V}_k$. This transformation preserves information such that $I(\tilde{\mathbf{K}}_k^{(m)}; \mathbf{Z}_L^{(m)}) \approx I(\Omega^{(m)}; \mathbf{Z}_L^{(m)})$, where $I(\cdot; \cdot)$ denotes mutual information, ensuring that $\tilde{\mathbf{K}}_k^{(m)}$ retains the core representational logic of $\mathbf{F}^{(m)}$.

*Proof.* As $\mathbf{Z}_L^{(m)}$ is a functional transformation of $\Omega^{(m)}$, the latter is a sufficient statistic for the model internal logic. The operator $\Psi$ projects these states into the shared manifold $\mathcal{V}_k$ while minimizing information dissipation. Since the relational knowledge is distributively encoded across the feature support, $\Psi$ acts as a sufficient mapping that reconstructs heterogeneous layouts into a canonical form. Thus, $\tilde{\mathbf{K}}_k^{(m)}$ retains the essential mutual information, ensuring architectural invariance with minimal knowledge loss.

**Remark 3.1.** The operator $\Psi$ thus serves as an interface to map heterogeneous model logics into a homogeneous basis $\tilde{\mathbf{K}}_k^{(m)}$, effectively decoupling neural dynamics from architecture-specific constraints. To operationalize this mapping, we implement $\Psi$ through a two-stage process of externalization and unification.

**Knowledge Externalization.** This stage decouples scale-specific knowledge from the EFM components. For each model $\mathbf{F}^{(m)} \in \mathcal{M}$, the operator retrieves raw patterns on two distinct scales:

- *Embedding Patterns* ($k = Emb$): The operator retrieves the initial patch-level representations from the input embedding layer $\mathcal{I}^{(m)}$ to capture localized neural signatures:

$$\mathbf{K}_{\text{Emb}}^{(m)} = \mathbf{Z}_0^{(m)} \in \mathbb{R}^{P_{\text{seq}}^{(m)} \times d^{(m)}} \tag{8}$$

- *Topological Patterns (k = Attn):* The operator retrieves attention maps $\mathbf{S}_\ell^{(m)}$ from a pre-defined set of layers $\mathbb{L}_{\text{sel}} \subseteq \{1, \ldots, L^{(m)}\}$ within the encoder $\mathcal{T}^{(m)}$ to characterize internal token-to-token pairwise dependencies:

$$\mathbf{K}_{\text{Attn}}^{(m)} = \{\mathbf{S}_\ell^{(m)}\}_{\ell \in \mathbb{L}_{\text{sel}}}, \quad \mathbf{S}_\ell^{(m)} \in \mathbb{R}^{P_{\text{seq}}^{(m)} \times P_{\text{seq}}^{(m)}} \quad (9)$$

For each EFM $\mathbf{F}^{(m)}$, the raw individual knowledge set is defined as $\mathcal{K}^{(m)} = \{\mathbf{K}_{\text{Emb}}^{(m)}, \mathbf{K}_{\text{Attn}}^{(m)}\}$, which remains constrained by the latent dimension of individual model.

**Knowledge Unification.** To enable collective fusion, the operator reconciles the retrieved raw patterns into a consistent semantic and structural space, generating unified individual knowledge $\tilde{\mathbf{K}}_k^{(m)}$. This involves two parallel unification mechanisms:

- *Embedding Unification (k = Emb):* A *bridge* mechanism maps embeddings into the shared manifold $\mathcal{V}_{\text{Emb}} = \mathbb{R}^{P_{\text{seq}} \times d_{\text{align}}}$ to resolve semantic and dimensional heterogeneities:

$$\tilde{\mathbf{K}}_{\text{Emb}}^{(m)} = \text{Bridge}^{(m)}(\mathbf{K}_{\text{Emb}}^{(m)}) \in \mathcal{V}_{\text{Emb}} \quad (10)$$

The bridge is a two-layer MLP with LayerNorm and GELU that projects heterogeneous latent spaces into a unified semantic space.

- *Topological Unification (k = Attn):* An *aligner* mechanism standardizes structural dependencies into a consistent manifold $\mathcal{V}_{\text{Attn}} = \mathbb{R}^{P_{\text{seq}} \times P_{\text{seq}}}$ by reconstructing attention maps to a uniform resolution:

$$\tilde{\mathbf{K}}_{\text{Attn}}^{(m)} = \text{Aligner}^{(m)}(\mathbf{K}_{\text{Attn}}^{(m)}) \in \mathcal{V}_{\text{Attn}} \quad (11)$$

Detailed formulations of architecture-specific topological operators are provided in Appendix F.

The resulting unified individual knowledge set $\tilde{\mathcal{K}}^{(m)} = \{\tilde{\mathbf{K}}_{\text{Emb}}^{(m)}, \tilde{\mathbf{K}}_{\text{Attn}}^{(m)}\}$ provides a homogeneous basis for subsequent collective fusion.

## 3.2. Adaptive Knowledge Weighting

*Intuition.* Despite knowledge unification, EFMs retain architectural biases that cause reliability to fluctuate across signal profiles. Each model contribution must therefore be dynamically calibrated. Unlike raw data that often introduce task-specific noise, intrinsic characteristics provide robust descriptors of signal dynamics. These features allow the fusion process to effectively match model inductive biases with the requirements of each neural sample.

**Lemma 3.2.** For any sample $\mathbf{X}$ with intrinsic characteristics $\mathbf{X}_{\text{meta}}$, there exists a conditional mapping $\mathbf{w} = \mathcal{G}(\mathbf{X}_{\text{meta}})$ yielding a weight distribution for $\mathbf{K}_k^{(*)} = \sum w^{(m)} \tilde{\mathbf{K}}_k^{(m)}$ such that the fused knowledge satisfies the conditional consistency: $I(\mathbf{K}_k^{(*)}; \{\tilde{\mathbf{K}}_k^{(m)}\}_{m=1}^M \mid \mathbf{X}_{\text{meta}}) \geq I(\mathbf{K}_k^{(*)}; \{\tilde{\mathbf{K}}_k^{(m)}\}_{m=1}^M)$, where $I(\cdot; \cdot \mid \cdot)$ denotes the conditional mutual information.

*Proof.* Let $\mathbf{K}_k^{(*)}$ be parameterized by $\mathbf{w} = \mathcal{G}(\mathbf{X}_{\text{meta}})$ via weighted aggregation. Since conditioning on intrinsic characteristics non-increasingly affects system entropy, $H(\mathbf{K}_k^{(*)} \mid \mathbf{X}_{\text{meta}}) \leq H(\mathbf{K}_k^{(*)})$, where $H(\cdot)$ denotes the Shannon entropy. Given the fundamental relation $I(X; Y) = H(X) - H(X|Y)$, this reduction in conditional uncertainty effectively filters task-irrelevant noise. By aligning model knowledge with signal dynamics, the captured mutual information is maximized such that $I(\mathbf{K}_k^{(*)}; \{\tilde{\mathbf{K}}_k^{(m)}\} \mid \mathbf{X}_{\text{meta}}) \geq I(\mathbf{K}_k^{(*)}; \{\tilde{\mathbf{K}}_k^{(m)}\})$, thereby achieving a tighter consistency bound.

***Remark 3.2.*** The mapping $\mathcal{G}$ functions as a meta-feature guided fusor that dynamically assigns weights based on characteristics dependent on the signal profile. This ensures that the collective knowledge $\mathbf{K}_k^{(*)}$ is a sample-adaptive synthesis tailored to the intrinsic physiological properties of the EEG input.

**Meta-guided Weight Derivation.** The process begins by extracting a meta-feature matrix $\mathbf{X}_{\text{meta}} \in \mathbb{R}^{C \times 10}$ from $\mathbf{X}_{\text{raw}}$. For each channel $c$, the characteristics are formulated as:

$$\mathbf{X}_{\text{meta}}^{(c)} = [\mathbf{v}_{\text{time}}, \mathbf{v}_{\text{freq}}, \mathbf{v}_{\text{rhy}}] \in \mathbb{R}^{1 \times 10} \quad (12)$$

where the signal is decomposed into three domains: **1) Temporal:** $\mathbf{v}_{\text{time}} = [\mu, \sigma, \text{ZCR}]$, capturing baseline statistics and signal volatility; **2) Spectral:** $\mathbf{v}_{\text{freq}} = [P_\delta, P_\theta, P_\alpha, P_\beta, P_\gamma]$, capturing power density across five bands; and **3) Rhythmic:** $\mathbf{v}_{\text{rhy}} = [\mathcal{R}_{\alpha/\beta}, \mathcal{R}_{\theta/\alpha}]$, reflecting neural state transitions. Detailed quantification is provided in Appendix G.

To resolve the sample specific contribution of diverse source models, the *fusor* $\mathcal{G}$ computes the weights $\mathbf{w} = [w^{(1)}, \ldots, w^{(M)}]^\top$ using the meta characteristics $\mathbf{X}_{\text{meta}}$:

$$\mathbf{w} = \text{Softmax}\left(\frac{1}{\tau}\mathbf{W}_g \left[\frac{1}{C}\sum_{c=1}^C \mathbf{X}_{\text{meta}}^{(c)}\right]^\top\right) \in \mathbb{R}^M \quad (13)$$

where $\mathbf{W}_g \in \mathbb{R}^{M \times 10}$ is a learnable projection and $\tau$ is the temperature scaling factor.

**Weighted Knowledge Aggregation.** Following the unification in Section 3.1, we synthesize the fused knowledge set $\mathcal{K}^{(*)} = \{\mathbf{K}_{\text{Emb}}^{(*)}, \mathbf{K}_{\text{Attn}}^{(*)}\}$ via weighted summation. The fusion is formulated as:

$$\mathbf{K}_k^{(*)} = \sum_{m=1}^M w^{(m)} \cdot \tilde{\mathbf{K}}_k^{(m)}, \quad \forall k \in \{\text{Emb}, \text{Attn}\} \quad (14)$$

This process ensures the carrier model $\mathbf{T}$ receives a sample adaptive supervision signal that effectively reconciles the representational strengths of the *model pool* $\mathcal{M}$.

*Table 1.* Summary of Downstream BCI Tasks and Datasets

| BCI Downstream Tasks | Datasets | #Samples | Train | Val | Test | Shape | Labels |
|---|---|---|---|---|---|---|---|
| Motor Imagery | BCICIV-2A | 5,088 | 1-5 | 6-7 | 8-9 | 22 channels × 4s | 4-class |
| | PhysioNet-MI | 9,837 | 1-70 | 71-89 | 90-109 | 64 channels × 4s | 4-class |
| | SHU-MI | 11,988 | 1-15 | 16-20 | 21-25 | 32 channels × 4s | 2-class |
| Emotion Recognition | SEED-V | 117,744 | 1-5 | 6-10 | 11-15 | 62 channels × 1s | 5-class |
| | FACED | 10,332 | 1-80 | 81-100 | 101-123 | 32 channels × 10s | 9-class |
| Sleep Staging | ISRUC_S1 | 86,320 | 1-80 | 81-90 | 91-100 | 6 channels × 30s | 5-class |
| | ISRUC_S3 | 8,500 | 1-8 | 9 | 10 | 6 channels × 30s | 5-class |
| Seizure Detection | CHB-MIT | 326,993 | 1-19 | 20-21 | 22-23 | 16 channels × 10s | 2-class |
| Imagined Speech | BCIC2020-3 | 6,000 | 1-60 | 61-70 | 71-80 | 64 channels × 3s | 5-class |
| Mental Disorder Diagnosis | Mumtaz2016 | 7,143 | 1-43 | 44-52 | 53-62 | 19 channels × 5s | 2-class |
| Mental Stress Detection | MentalArithmetic | 1,707 | 1-28 | 29-32 | 33-36 | 20 channels × 5s | 2-class |
| Mental Attention Detection | ATTENTION | 4,680 | 1-16 | 17-21 | 22-26 | 28 channels × 4s | 2-class |

## 3.3. Carrier Selection and Fusion Optimization

***Intuition.*** The efficacy of a carrier depends on the structural congruence between its inherent representational space and the target task relational logic. An EFM whose inductive bias inherently aligns with downstream semantic dependencies provides a superior optimization prior, thereby minimizing the recalibration cost required to internalize knowledge.

**Lemma 3.3.** Let $\mathcal{R}_{\mathbf{T}}$ and $\mathcal{R}_y$ denote the relational structures of carrier $\mathbf{T}$ and the target task, respectively. There exists a compatibility score $\xi^{(m)}$ such that maximizing $\xi^{(m)}$ minimizes the conditional uncertainty of the collective knowledge pool $\mathcal{K}^{(*)}$ relative to the carrier state, satisfying $H(\mathcal{K}^{(*)} \mid \mathcal{R}_{\mathbf{T}}) \leq H(\mathcal{K}^{(*)} \mid \mathcal{R}_{m \neq \mathbf{T}})$.

*Proof.* Let structural resonance satisfy the Markov chain $\mathcal{R}_y \to \mathcal{R}_{\mathbf{T}} \to \mathbf{Z}_L$. According to the Data Processing Inequality (DPI), the mutual information is bounded by $I(\mathbf{Z}_L; \mathcal{K}^{(*)}) \leq I(\mathcal{R}_{\mathbf{T}}; \mathcal{K}^{(*)})$. Since $H(\mathcal{K}^{(*)} \mid \mathcal{R}_{\mathbf{T}}) = H(\mathcal{K}^{(*)}) - I(\mathcal{R}_{\mathbf{T}}; \mathcal{K}^{(*)})$, maximizing $\xi^{(m)}$ directly maximizes $I(\mathcal{R}_{\mathbf{T}}; \mathcal{K}^{(*)})$, thereby minimizing the conditional entropy. This ensures well conditioned gradient trajectories and minimizes information dissipation during knowledge internalization.

***Remark 3.3.*** The compatibility score $\xi^{(m)}$ is empirically evaluated through the zero-training alignment process described in the subsequent section. By identifying $\mathbf{T}$ based on this structural resonance, the subsequent multi-objective optimization is initialized within a region of high semantic density, ensuring the internalization of $\mathcal{K}^{(*)}$ complements rather than disrupts the carrier's pre-trained foundations.

**Carrier Selection (Q).** The selection function $\mathrm{Q}(\mathcal{M}, \mathcal{D})$ identifies the optimal architectural vessel by evaluating structural compatibility. For each candidate $\mathbf{F}^{(m)} \in \mathcal{M}$, we compute a compatibility score $\xi^{(m)}$ to measure the alignment between latent feature geometry and the label space:

$$\xi^{(m)} = \rho_{\text{Spearman}}\big(\text{lt-RDM}(\mathbf{R}^{(m)}), \text{lt-RDM}(\mathbf{Y}_{\text{probe}})\big), \quad (15)$$

where $\rho_{\text{Spearman}}$ is the Spearman rank correlation (Bolya

et al., 2021). To ensure efficiency, we use a stratified probe set $\mathcal{D}_{\text{probe}} \subset \mathcal{D}$ with a 0.2 sampling ratio. The feature matrix $\mathbf{R}^{(m)} \in \mathbb{R}^{B \times d^{(m)}}$ is constructed by average-pooling the final representations $\mathbf{Z}_L^{(m)}$ across the sequence dimension. The lt-RDM$(\cdot)$ operator calculates the lower triangular dissimilarity matrix based on Pearson correlation distance. Before computation, we standardize $\mathbf{R}^{(m)}$ and transform labels into one-hot vectors $\mathbf{Y}_{\text{probe}}$. This approach captures the inherent semantic correlation without requiring costly gradient updates or full-scale training.

The final selection of the carrier $\mathbf{T}$ is formulated as:

$$\mathbf{T} = \mathrm{Q}(\mathcal{M}, \mathcal{D}) = \underset{\mathbf{F}^{(m)} \in \mathcal{M}}{\arg\max} \, \xi^{(m)} \quad (16)$$

**Knowledge Fusion Objective.** The objective $\mathcal{J}$ defined in Eq. (4) is instantiated by coupling the task-specific supervision with the multi-scale extraction and fusion operators:

$$\begin{aligned}
\mathcal{J}(\theta^{(\mathbf{T})}; \mathcal{D}, \mathcal{K}^{(*)}) = &\lambda \cdot \mathcal{L}_{\text{task}}(\hat{\mathbf{y}}, \mathbf{y}) \\
&+ \sum_{k \in \{\text{Emb, Attn}\}} \gamma_k \cdot \mathcal{L}_{\text{fusion}}\Big(\Psi_k^{(\mathbf{T})}(\mathbf{F}^{(\mathbf{T})}, \mathbf{X}), \\
&\quad \mathcal{G}(\{\mathbf{K}_k^{(m)}\}_{m=1}^M, \mathbf{X}_{\text{raw}})\Big),
\end{aligned} \quad (17)$$

where $\Psi_k^{(\mathbf{T})}(\cdot)$ and $\mathcal{G}(\cdot)$ follow the definitions in Section 3.1 and Section 3.2, respectively.

## 4. Experiments

### 4.1. Experiment Setup

**Downstream BCI Tasks.** To evaluate EmBrace across diverse neural contexts, we select 12 datasets covering **8 representative BCI paradigms**: **1) Emotion Recognition**, **2) Motor Imagery**, **3) Sleep Staging**, **4) Seizure Detection**, **5) Imagined Speech**, **6) Mental Disorder Diagnosis**, **7) Mental Stress Detection**, and **8) Mental Attention Detection**. All EEG signals are resampled to 200 Hz. The

*Table 2.* Comparison results of different methods on downstream tasks. **Bold** and underline denote the best and second-best performance among the foundation models, respectively.

| Methods | BCIC-IV-2A (4-class) | | | FACED (9-class) | | |
|---|---|---|---|---|---|---|
| | ACC-B | W-F1 | Kappa | ACC-B | W-F1 | Kappa |
| EEGNet | $0.5668 \pm 0.0112$ | $0.5498 \pm 0.0161$ | $0.4225 \pm 0.0149$ | $0.2282 \pm 0.0167$ | $0.1781 \pm 0.0248$ | $0.1310 \pm 0.0175$ |
| EEG-Deformer | $0.5488 \pm 0.0170$ | $0.5260 \pm 0.0227$ | $0.3984 \pm 0.0227$ | $0.3439 \pm 0.0251$ | $0.3450 \pm 0.0257$ | $0.2614 \pm 0.0269$ |
| BIOT | $0.4080 \pm 0.0173$ | $0.3403 \pm 0.0213$ | $0.2106 \pm 0.0231$ | $0.2590 \pm 0.0064$ | $0.2556 \pm 0.0057$ | $0.1642 \pm 0.0066$ |
| LaBraM | $0.3453 \pm 0.0089$ | $0.3110 \pm 0.0449$ | $0.1271 \pm 0.0119$ | $0.2983 \pm 0.0357$ | $0.2986 \pm 0.0359$ | $0.2096 \pm 0.0406$ |
| CBraMod | $0.4922 \pm 0.0354$ | $0.4679 \pm 0.0459$ | $0.3229 \pm 0.0473$ | $0.5425 \pm 0.0077$ | $0.5444 \pm 0.0077$ | $0.4833 \pm 0.0089$ |
| CodeBrain | $0.3747 \pm 0.0233$ | $0.3007 \pm 0.0295$ | $0.1662 \pm 0.0311$ | $0.5835 \pm 0.0079$ | $0.5891 \pm 0.0094$ | $0.5290 \pm 0.0093$ |
| **EmBrace** (Ours) | **$0.5047 \pm 0.0203$** | **$0.4870 \pm 0.0240$** | **$0.3396 \pm 0.0271$** | **$0.6136 \pm 0.0055$** | **$0.6131 \pm 0.0060$** | **$0.5608 \pm 0.0062$** |

| Methods | ISRUC-S1 (5-class) | | | CHB-MIT (2-class) | | |
|---|---|---|---|---|---|---|
| | ACC-B | W-F1 | Kappa | ACC-B | AUC-PR | AUROC |
| EEGNet | $0.6504 \pm 0.0181$ | $0.6825 \pm 0.0158$ | $0.6236 \pm 0.0229$ | $0.6093 \pm 0.0429$ | $0.2890 \pm 0.0952$ | $0.9194 \pm 0.0114$ |
| EEG-Deformer | $0.7095 \pm 0.0169$ | $0.7360 \pm 0.0100$ | $0.6571 \pm 0.0148$ | $0.6255 \pm 0.0253$ | $0.3551 \pm 0.0095$ | $0.9081 \pm 0.0130$ |
| BIOT | $0.7795 \pm 0.0061$ | $0.8028 \pm 0.0071$ | $0.7451 \pm 0.0084$ | $0.5764 \pm 0.0693$ | $0.2981 \pm 0.0088$ | $0.8734 \pm 0.0095$ |
| LaBraM | $0.7880 \pm 0.0067$ | $0.8023 \pm 0.0107$ | $0.7455 \pm 0.0124$ | $0.5776 \pm 0.0089$ | $0.4192 \pm 0.0929$ | $0.8997 \pm 0.0384$ |
| CBraMod | $0.7701 \pm 0.0070$ | $0.7915 \pm 0.0093$ | $0.7323 \pm 0.0061$ | **$0.7166 \pm 0.0241$** | $0.4779 \pm 0.1830$ | $0.9025 \pm 0.0355$ |
| CodeBrain | $0.7751 \pm 0.0078$ | $0.7945 \pm 0.0104$ | $0.7393 \pm 0.0147$ | $0.6265 \pm 0.0432$ | $0.3848 \pm 0.0588$ | $0.8690 \pm 0.0271$ |
| **EmBrace** (Ours) | **$0.7927 \pm 0.0119$** | **$0.8110 \pm 0.0107$** | **$0.7617 \pm 0.0111$** | $0.6157 \pm 0.0913$ | **$0.5274 \pm 0.1725$** | **$0.9296 \pm 0.0204$** |

| Methods | BCIC2020-3 (5-class) | | | Mumtaz2016 (2-class) | | |
|---|---|---|---|---|---|---|
| | ACC-B | W-F1 | Kappa | ACC-B | AUC-PR | AUROC |
| EEGNet | $0.2600 \pm 0.0151$ | $0.2573 \pm 0.0163$ | $0.0750 \pm 0.0189$ | $0.8867 \pm 0.0542$ | $0.9586 \pm 0.0219$ | $0.9505 \pm 0.0211$ |
| EEG-Deformer | $0.4563 \pm 0.0113$ | $0.4567 \pm 0.0116$ | $0.3203 \pm 0.0141$ | $0.9097 \pm 0.0314$ | $0.9768 \pm 0.0023$ | $0.9705 \pm 0.0030$ |
| BIOT | $0.3592 \pm 0.0187$ | $0.3579 \pm 0.0191$ | $0.1990 \pm 0.0234$ | $0.8866 \pm 0.0158$ | $0.9763 \pm 0.0149$ | $0.9737 \pm 0.0171$ |
| LaBraM | $0.2288 \pm 0.0255$ | $0.2139 \pm 0.0347$ | $0.0360 \pm 0.0319$ | $0.9053 \pm 0.0069$ | $0.9767 \pm 0.0043$ | $0.9728 \pm 0.0071$ |
| CBraMod | $0.3120 \pm 0.0155$ | $0.3112 \pm 0.0158$ | $0.1400 \pm 0.0193$ | $0.8867 \pm 0.0036$ | $0.9784 \pm 0.0050$ | $0.9791 \pm 0.0056$ |
| CodeBrain | $0.5659 \pm 0.0110$ | $0.5659 \pm 0.0115$ | $0.4573 \pm 0.0138$ | **$0.9054 \pm 0.0068$** | $0.9806 \pm 0.0018$ | $0.9786 \pm 0.0023$ |
| **EmBrace** (Ours) | **$0.5867 \pm 0.0115$** | **$0.5869 \pm 0.0115$** | **$0.4833 \pm 0.0144$** | $0.9048 \pm 0.0039$ | **$0.9811 \pm 0.0040$** | **$0.9798 \pm 0.0049$** |

| Methods | MentalArithmetic (2-class) | | | ATTENTION (2-class) | | |
|---|---|---|---|---|---|---|
| | ACC-B | AUC-PR | AUROC | ACC-B | AUC-PR | AUROC |
| EEGNet | $0.5847 \pm 0.0619$ | $0.5770 \pm 0.1300$ | $0.7875 \pm 0.0633$ | $0.7344 \pm 0.0369$ | $0.8046 \pm 0.0367$ | $0.8271 \pm 0.0386$ |
| EEG-Deformer | $0.5090 \pm 0.0129$ | $0.4899 \pm 0.0323$ | $0.7297 \pm 0.0231$ | $0.5664 \pm 0.0457$ | $0.8021 \pm 0.0080$ | $0.8138 \pm 0.0091$ |
| BIOT | $0.6229 \pm 0.0743$ | $0.5428 \pm 0.1534$ | $0.7717 \pm 0.0744$ | $0.5784 \pm 0.0402$ | $0.6839 \pm 0.0542$ | $0.7081 \pm 0.0310$ |
| LaBraM | $0.6840 \pm 0.0673$ | $0.5619 \pm 0.1650$ | $0.7073 \pm 0.1132$ | $0.6170 \pm 0.0287$ | $0.6610 \pm 0.0176$ | $0.6789 \pm 0.0255$ |
| CBraMod | $0.7063 \pm 0.0613$ | $0.6109 \pm 0.0686$ | $0.7979 \pm 0.0500$ | $0.7033 \pm 0.0139$ | $0.7803 \pm 0.0054$ | $0.7915 \pm 0.0045$ |
| CodeBrain | $0.7174 \pm 0.0144$ | $0.6417 \pm 0.0507$ | $0.8487 \pm 0.0190$ | $0.6704 \pm 0.0082$ | $0.7363 \pm 0.0070$ | $0.7503 \pm 0.0055$ |
| **EmBrace** (Ours) | **$0.7292 \pm 0.0391$** | **$0.7377 \pm 0.0415$** | **$0.8813 \pm 0.0201$** | **$0.7217 \pm 0.0036$** | **$0.7884 \pm 0.0129$** | **$0.8031 \pm 0.0104$** |

structural properties and paradigms are summarized in Table 1. Further dataset specifications, preprocessing details, and extended results are provided in Appendix. H.

**Baselines.** We benchmark EmBrace against a comprehensive model pool, categorized into: **1) Foundation Models:** including our fused constituents (LaBraM (Jiang et al., 2024), CBraMod (Wang et al., 2025), CodeBrain (Ma et al., 2026)) and BIOT (Yang et al., 2023); **2) Task-specific Models:** including 2 representative architectures (EEGNet (Lawhern et al., 2016), EEG-Deformer (Ding et al., 2024)) and 6 additional established baselines. Further technical specifications and baseline list are provided in Appendix B.

**Metrics.** For multi-class classification tasks, we use Cohen's Kappa (designated as the monitoring score), Weighted F1 Score, and Balanced Accuracy. For binary classification

tasks, the metrics are AUROC (designated as the monitoring score), AUC-PR, and Balanced Accuracy. All experiments are conducted using five different random seeds with the mean and standard deviation reported.

**Implementation Details.** Our framework is implemented in PyTorch 2.0.0 and CUDA 11.4 on NVIDIA GeForce RTX 2080 Ti GPUs. We utilize a *model pool* $\mathcal{M} = \{LaBraM, CBraMod, CodeBrain\}$ for knowledge extraction, where *attention knowledge* is specifically derived from the final layer $L^{(m)}$ of each model. We set $d_{align} = 256$, $\tau = 8$, and loss weights $\lambda = 0.6, \gamma_{Emb} = \gamma_{Attn} = 0.4$. To reconcile structural differences, a Block-Diagonal strategy is employed for the unification of extracted attention patterns. We employ Mean Squared Error (MSE) as the optimization criterion for both $\mathcal{L}_{emb}$ and $\mathcal{L}_{attn}$. Detailed configurations

and hyperparameters are summarized in Appendix C.

## 4.2. Performance Comparison with Baselines

**Overall Performance.** Table 2 summarizes the main results across eight representative BCI paradigms. Across the 24 reported evaluation metrics, EmBrace achieves the best performance among foundation models on 21 metrics, demonstrating strong generalization across heterogeneous EEG tasks. In particular, EmBrace consistently outperforms individual EFMs on multi-class tasks such as BCIC2020-3, FACED, and ISRUC-S1, indicating that representation-level knowledge fusion can effectively integrate complementary strengths from heterogeneous EEG foundation models.

**Superiority over Foundation Models.** Compared with individual EFMs, EmBrace provides consistent improvements in most paradigms. For example, on FACED, EmBrace improves the Kappa score from 0.5290 to 0.5608 over the strongest EFM baseline, CodeBrain. On BCIC2020-3, EmBrace surpasses CodeBrain by 2.08 percentage points in ACC-B, showing its advantage in challenging imagined speech decoding. For CHB-MIT and Mumtaz2016, although EmBrace does not obtain the highest ACC-B among EFMs, it achieves the best probabilistic discrimination metrics, including AUC-PR and AUROC, suggesting stronger calibration and ranking ability in clinically relevant binary tasks.

**Comparison with Task-Specific Models.** Task-specific lightweight models remain competitive in several paradigms, especially when the dataset structure closely matches their architectural assumptions. For instance, EEGNet performs strongly on BCIC-IV-2A, while EEG-Deformer shows competitive results on Mumtaz2016. This observation is consistent with recent findings that EEG foundation models do not always outperform compact neural or classical decoders across diverse BCI tasks (Yang et al., 2026). Nevertheless, such task-specific models often rely on dataset-dependent architectural biases, whereas EFMs provide a unified and scalable backbone for general-purpose EEG decoding. Under this setting, EmBrace improves the reliability of EFMs by adaptively integrating complementary knowledge from multiple pre-trained architectures, maintaining robust performance across substantially different signal settings, task objectives, and clinical or cognitive scenarios.

## 4.3. Ablation Study

We conduct extensive ablation studies and sensitivity analyses to justify the design choices of EmBrace. Ablation setups and results are detailed in Appendix I and Appendix J.

- **Sensitivity Analysis:** Analyses in Appendix I demonstrate that balancing task-specific and fusion-oriented terms in $\mathcal{J}$ via $\lambda = 0.6, \gamma_{\text{Emb}} = \gamma_{\text{Attn}} = 0.4$ ensures

a well-conditioned optimization landscape and robust gradient trajectories during joint training. Further investigations in Appendix J.2 and Appendix J.8 reveal that a moderate temperature $\tau$ and an alignment dimension $d_{\text{align}} = 256$ effectively calibrate the multi-source knowledge set $\mathcal{K}^{(*)}$ while avoiding representation dilution.

- **Architecture Design:** Comparative experiments in Appendix J.1, J.5, and J.7 confirm that the combination of shared dynamic weighting via $\mathcal{G}$, a shared MLP bridge, and the block-diagonal unification strategy best preserves the inherent neural dependencies within heterogeneous EEG manifolds. Notably, we observe that terminal layers ($\mathbb{L}_{\text{sel}} = \{L^{(m)}\}$) provide the most refined semantic knowledge, whereas incorporating shallower layers tends to introduce redundant noise that hampers convergence.

- **Selection & Optimization:** Results in Appendix J.3 indicate that the selection method (Bolya et al., 2021) significantly outperforms NCE and LogME in identifying the most compatible carrier model $\mathbf{T}$ for specific downstream tasks. Compared to standalone fine-tuning, EmBrace yields universal performance gains across all candidate backbones (Appendix J.6), with MSE identified as the most numerically stable fusion criterion $\mathcal{L}_{\text{fusion}}$ for aligning non-isomorphic knowledge (Appendix J.4).

## 4.4. Analysis of Fusion Synergy and Carrier Selection

We evaluate the efficacy of EmBrace by comparing each EFM backbone under two settings: individual fine-tuning and acting as the carrier within our knowledge fusion framework. As shown in Table 3, EmBrace yields universal fusion gains, consistently surpassing standalone FT regardless of the model architecture. This confirms that integrating heterogeneous expertise effectively compensates for individual model limitations. Furthermore, the optimal carriers identified by our selection method consistently achieve peak performance across all tasks (e.g., 0.8813 AUROC on MentalArithmetic), validating the framework's ability to identify the most compatible carrier for synchronization. Detailed results are provided in Appendix J.6.

*Table 3.* Ablation results comparing single-model fine-tuning (FT) and Knowledge Fusion (Fusion). We report **Kappa** metric for ISRUC-S3 and BCIC-IV-2A, and **AUROC** for MentalArithmetic. Shaded cells indicate the carrier identified by our selection method. **Bold** denotes the better performance in each pair.

| Model | ISRUC-S3 | | BCIC-IV-2A | | MentalArith. | |
|---|---|---|---|---|---|---|
| | FT | Fusion | FT | Fusion | FT | Fusion |
| LaBraM | 0.7035 | **0.7098** | 0.1271 | **0.1285** | 0.7073 | **0.7234** |
| CBraMod | 0.4463 | **0.4678** | 0.3229 | **0.3396** | 0.7979 | **0.8060** |
| CodeBrain | 0.5807 | **0.6090** | 0.1662 | **0.1808** | 0.8487 | **0.8813** |

*Table 4.* Efficiency comparison across three EFMs. $\Sigma_3^\dagger$ denotes the summed efficiency metrics of LaBraM, CBraMod, and CodeBrain.

| Method | FACED | | ATTENTION | | SEED-V | | ISRUC-S1 | |
|---|---|---|---|---|---|---|---|---|
| | Train Time /Epoch (s) | Latency (ms) | Train Time /Epoch (s) | Latency (ms) | Train Time /Epoch (s) | Latency (ms) | Train Time /Epoch (s) | Latency (ms) |
| LaBraM | 14.58 | 0.88 | 6.16 | 0.60 | 19.40 | 0.20 | 132.60 | 0.83 |
| CBraMod | 11.12 | 0.60 | 4.70 | 0.23 | 29.05 | 0.22 | 116.68 | 0.86 |
| CodeBrain | 33.84 | 1.65 | 13.42 | 0.82 | 75.15 | 0.72 | 194.40 | 2.89 |
| $\Sigma_3^\dagger$ EFMs | 59.54 | 3.13 | 24.28 | 1.65 | 123.60 | 1.14 | 443.68 | 4.58 |
| **EmBrace** (Ours) | 57.11 | 1.65 | 12.95 | 0.29 | 122.81 | 0.93 | 188.96 | 0.85 |

## 4.5. Computational Efficiency Analysis

To evaluate the deployability of EmBrace, Table 4 profiles its training and inference efficiency against three constituent EFMs and their cumulative cost $\Sigma_3^\dagger$. Although EmBrace leverages knowledge from multiple heterogeneous EFMs, it avoids deploying or optimizing all source models independently at inference time. Compared with $\Sigma_3^\dagger$, EmBrace reduces inference latency by 47.3%, 82.4%, 18.4%, and 81.4% on FACED, ATTENTION, SEED-V, and ISRUC-S1, respectively. Its training cost is also comparable to or lower than the cumulative baseline, with reductions of 4.1%, 46.7%, 0.6%, and 57.4% across the four datasets. These advantages demonstrate that EmBrace enables efficient multi-EFM knowledge fusion with a single deployable carrier model.

## 4.6. Visualization of Meta-Guided Fusor Weights

We visualize the learned projection matrix $\mathbf{W}_g$ in Eq. (13) to inspect how physiological meta-features are associated with the weighting of unified knowledge from each source EFM. On the PhysioNet-MI dataset, as shown in Figure 3, ZCR exhibits a prominent positive association with the score assigned to knowledge from LaBraM, whereas the $\theta/\alpha$ power ratio shows a strong negative association with that of CBraMod. These patterns suggest that the Fusor learns source specific feature-to-knowledge weighting relations from temporal, spectral, and rhythmic signal properties, providing an interpretable view of how physiological meta-features participate in heterogeneous knowledge fusion.

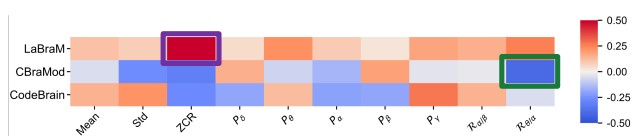

*Figure 3.* Visualization of Meta-Guided Fusor Weights.

## 5. Related Work

**EEG Foundation Models.** Recent EEG Foundation Models (EFMs) such as LaBraM (Jiang et al., 2024), CBraMod (Wang et al., 2025), and CodeBrain (Ma et al., 2026) have significantly advanced cross-task generalization through large-scale pre-training. Despite these gains, most EFMs rely on a single shared encoder, which often struggles to decouple overlapping cognitive processes across diverse paradigms. As no single architecture is universally optimal for all brain signal variations, there is a critical need to transition from individual encoders to a unified framework that integrates the specialized expertise of heterogeneous architectures.

**Knowledge Fusion.** Traditional integration methods such as model ensembles (Van, 2012) and parameter-level fusion (Wortsman et al., 2022) are often constrained by high inference costs or rigid architectural requirements. Furthermore, aligning output distributions (Wan et al., 2024) is mathematically inconsistent for EFMs due to the divergence between pre-training reconstruction and fine-tuning classification objectives. Consequently, our work shifts toward fusion at the representational level. We initiate by formulating a systematic feature-label compatibility criterion (Bolya et al., 2021) to identify an optimal carrier model. A unified manifold is then established via specialized unification mechanisms to align heterogeneous embeddings and attention patterns from the model pool. Unlike generic time-series fusion (Liu et al., 2025; Shentu et al., 2025), we incorporate temporal-spectral-rhythmic biological priors to drive a dynamic weighting mechanism, ensuring the fusion remains neuroscientifically interpretable and robust in complex BCI scenarios.

## 6. Conclusion

This paper presents EmBrace, a novel representational knowledge fusion framework designed to harness the collective intelligence of heterogeneous EFMs. By externalizing multi-scale intermediate representations into a unified manifold, EmBrace effectively circumvents the structural barriers that hinder traditional fusion methods. We introduce a sample-aware adaptive weighting mechanism guided by temporal-spectral-rhythmic biological priors, ensuring that the contribution of knowledge is dynamically tailored to intrinsic neural dynamics. Furthermore, by identifying the optimal carrier model through a structural resonance-based selection strategy, we maximize the efficiency of knowledge internalization. Extensive evaluations across 12 datasets demonstrate that EmBrace consistently achieves state-of-the-art performance, providing a robust and neuroscientifically interpretable solution for complex BCI applications.

## Acknowledgements

This work is supported by the Youth Science Fund Project of National Natural Science Foundation of China (Grant No. 62306317), Beijing Nova Program (Grant No. 20250484804), and Young Elite Scientists Sponsorship Program of the Beijing High Innovation Plan (Grant No. 20250912).

## Impact Statement

This paper presents work aimed at advancing machine learning for EEG-based brain-computer interfaces and neurotechnology applications by enabling efficient knowledge fusion from heterogeneous EEG foundation models. While EEG-based neurotechnology may raise general concerns such as data privacy, informed consent, demographic bias, and responsible deployment, our work is primarily a technical contribution and does not introduce additional ethical risks beyond those commonly associated with this area.

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

# Appendix

## A. Related Work

### A.1. EEG Foundation Models

The primary challenge in BCI is enhancing model generalization across diverse tasks. Recently, EEG Foundation Models (EFMs) have emerged as a robust solution by pre-training on large-scale unlabeled data. **LaBraM** (Jiang et al., 2024) established a new benchmark by integrating a neural tokenizer with a vanilla Transformer to learn generic spatio-temporal representations. Subsequent advancements such as **CBraMod** (Wang et al., 2025) introduced a criss-cross attention mechanism to efficiently capture dependencies across temporal and channel dimensions, while **EEGPT** (Wang et al., 2024) focused on ensuring reliable representations across universal BCI applications. Most recently, **CodeBrain** (Ma et al., 2026) proposed a multi-scale decoupled architecture to enhance internal feature extraction and interpretability. Recent studies further broaden the design space of EFMs beyond encoder-centric architectures, with **ECHO** (Liu et al., 2026) exploring decoder-centric sequence-to-sequence modeling and **BrainPro** (Ding et al., 2025) introducing brain-state-aware representation learning with flexible spatial relationship modeling.

Despite this growing architectural diversity, existing EFMs are still primarily developed as individual backbones, each with its own architectural bias and task preference. Since brain signals vary substantially across paradigms, no single architecture is universally optimal regardless of its internal complexity. This motivates a transition from individual models to a unified framework that integrates expertise from heterogeneous architectures.

### A.2. Model Selection and Integration

Several traditional methods exist for model integration. **Model Ensembles** (Van, 2012) typically average final predictions but ignore deep structural features and require significant inference power. **Parameter-level Fusion** (Wortsman et al., 2022) blends weights directly but is strictly limited to models with identical architectures. While **Knowledge Distillation** (Gou et al., 2021) can transfer information, it primarily serves model compression and often leads to performance loss when the student fails to capture the full teacher complexity. Recently, **Knowledge Fusion** has been applied to large language models (Wan et al., 2024) by aligning output probability distributions. However, this approach is unsuitable for EFMs. During pre-training, these models use reconstruction heads to recover signals, but these are replaced by different MLP decoders for classification during fine-tuning. This target shift makes prediction-level alignment infeasible.

Our work therefore shifts the focus toward representation-level integration of heterogeneous EFMs. Given that no single architecture is universally optimal across EEG paradigms, we first identify a suitable carrier model using a feature-label compatibility criterion (Bolya et al., 2021). This criterion measures how well a model's feature space aligns with task labels, allowing us to select a discriminative backbone without exhaustive fine-tuning. We then construct a unified manifold to align multi-source embeddings and internal attention maps from diverse pre-trained encoders, enabling complementary knowledge to be fused into the selected carrier model.

### A.3. Physiology-Aware Dynamic Knowledge Fusion

The final objective is to ensure the fusion adapts to varying input characteristics. Recent advancements in general time-series foundation models (Liu et al., 2025; Shentu et al., 2025) have highlighted the necessity of adaptive mechanisms such as flexible information bottlenecks or data-driven descriptors to handle the significant divergence across multi-domain datasets. While these approaches provide a comprehensive view of abstract numerical properties, they do not explicitly account for the governing physiological principles of brain activity.

In contrast, we specialize the fusion for the EEG domain by introducing **temporal-spectral-rhythmic** biological priors. Specifically, we encode neuronal oscillation patterns across five frequency bands (Carskadon & Dement, 2011; Klimesch, 1999; Tallon-Baudry & Bertrand, 1999) and include rhythmic ratios to capture brain state transitions (Raufi & Longo, 2022). This biological information drives the dynamic weighting process, allowing the system to adjust the contribution of knowledge from heterogeneous source models based on identifiable brain rhythms. This approach ensures a more robust and neuroscientifically interpretable fusion in complex BCI scenarios.

# B. Baselines and Metrics

## B.1. Baselines

### B.1.1. TASK-SPECIFIC MODELS

- **EEGNet** (Lawhern et al., 2016): A compact CNN architecture that utilizes depthwise and separable convolutions to capture frequency-specific and spatial filters with minimal parameters.

- **EEG-Deformer** (Ding et al., 2024): A hierarchical CNN-Transformer framework integrating fine-grained temporal learning and dense information purification to capture multi-scale EEG dynamics.

- **SERA** (Jia et al., 2025): A neurophysiology-inspired framework utilizing a multi-stage VAE to disentangle EEG signals into independent source activities while employing a coarse-to-fine alignment block to mitigate inter-subject variability.

- **SPaRCNet** (Page et al., 2017): This deep 1D CNN employs dense residual connections and adaptive mechanisms to ensure robust feature extraction from non-stationary signals.

- **ContraWR** (Yang et al., 2021): A 2D CNN-based framework that converts EEG into multi-channel spectrograms and applies a ResNet-style architecture to exploit time-frequency patterns.

- **CNN-Transformer** (Peh et al., 2022): The dual-component model combines CNNs for local feature extraction with Transformer blocks for capturing long-range global dependencies.

- **FFCL** (Li et al., 2022): A parallel architecture that extracts spatial and temporal features through independent CNN and LSTM branches before final fusion and classification.

- **ST-Transformer** (Song et al., 2021): Applying self-attention across both channel and time dimensions, this pure Transformer network models global spatiotemporal dependencies without convolutional priors.

### B.1.2. FOUNDATION MODELS

- **BIOT** (Yang et al., 2023): **Biosignal Transformer (BIOT)** is a linear Transformer-based foundation model that leverages a domain-invariant attention mechanism to effectively address cross-dataset distribution shifts. To handle the 18-channel pre-training constraint, we partition input channels into groups, extract features via a shared backbone, and aggregate them to preserve the original pre-trained weights.

- **LaBraM** (Jiang et al., 2024): **Large Brain Model (LaBraM)** is a scalable framework that excels at capturing universal spatiotemporal correlations through a neural tokenizer and masked signal modeling. The architecture employs a full-attention Transformer encoder to extract comprehensive characteristics from large-scale neural datasets, enabling effective representation learning.

- **CBraMod** (Wang et al., 2025): **Criss-Cross Brain Foundation Model (CBraMod)** is a spatiotemporal-tailored model that utilizes a criss-cross Transformer backbone to decouple representation learning into parallel spatial and temporal attention paths. This design facilitates the independent yet simultaneous modeling of complex brain topologies and dynamic temporal evolutions.

- **CodeBrain** (Ma et al., 2026): **CodeBrain** is an efficient two-stage model that mimics the brain's small-world topology using a State Space Model (SSM) backbone to capture sparse, long-range dependencies with linear complexity. It integrates a TFDual-Tokenizer and EEGSSM structure to encode both temporal and spectral components for high-fidelity neural decoding.

## B.2. Metrics

Following the benchmarks established by recent EEG foundation models (Jiang et al., 2024; Wang et al., 2025; Ma et al., 2026; Zhou et al., 2025b), we employ five complementary metrics to ensure a rigorous and balanced evaluation. Specifically, performance in multi-class classification tasks is evaluated via Balanced Accuracy, Weighted F1-score and Cohen's Kappa, while performance in binary classification tasks is evaluated via Balanced Accuracy, AUC-PR, and AUROC.

- **Balanced Accuracy (ACC-B)**: This metric normalizes traditional accuracy by averaging Recall across all $C$ categories, where $\text{Recall}_i = \frac{TP_i}{TP_i + FN_i}$ is the ratio of true positives $TP_i$ to the total number of actual positives including missed

false negatives $FN_i$:

$$\text{ACC-B} = \frac{1}{C} \sum_{i=1}^{C} \text{Recall}_i \tag{18}$$

- **Weighted F1 (W-F1)**: Calculated as the support-weighted average of class-specific F1-scores, this measure scales the harmonic mean of $\text{Precision}_i$ and $\text{Recall}_i$ by the ratio of class samples $n_i$ to the total sample size $N$:

$$\text{W-F1} = \sum_{i=1}^{C} \frac{n_i}{N} \left( \frac{2 \cdot \text{Precision}_i \cdot \text{Recall}_i}{\text{Precision}_i + \text{Recall}_i} \right) \tag{19}$$

where $\text{Precision}_i = \frac{TP_i}{TP_i + FP_i}$ represents the positive predictive value accounting for false positives $FP_i$.

- **Cohen's Kappa (Kappa)**: A statistical coefficient that quantifies the agreement between predictions and ground truth beyond random chance by comparing the observed proportional agreement $p_o$ to the expected agreement $p_e$ calculated from the marginals of the confusion matrix:

$$\text{Kappa} = \frac{p_o - p_e}{1 - p_e} \tag{20}$$

- **Area Under the Precision-Recall Curve (AUC-PR)**: The AUC-PR metric evaluates the model's ability to maintain high precision $P$ across the entire range of Recall, providing a robust assessment of the trade-off between positive predictive value and sensitivity:

$$\text{AUC-PR} = \int_0^1 P(\text{Recall}) \, d\text{Recall} \tag{21}$$

- **Area Under the Receiver Operating Characteristic Curve (AUROC)**: Representing the integral of the True Positive Rate $\text{TPR} = \frac{TP}{TP + FN}$ over the False Positive Rate $\text{FPR} = \frac{FP}{FP + TN}$ across all possible decision boundaries, this score quantifies the discriminative power where $TN$ denotes true negatives:

$$\text{AUROC} = \int_0^1 \text{TPR}(\text{FPR}) \, d\text{FPR} \tag{22}$$

## C. Hyperparameter Setting

We summarize the detailed hyperparameter settings of our proposed framework in Table 5 and Table 6.

*Table 5.* General hyperparameter settings.

| Hyperparameters | Values |
|---|---|
| *Bridge* output dimension $d_{\text{align}}$ | 256 |
| *Bridge* hidden dimension | 400 |
| *Aligner* strategy | Block-Diagonal |
| *Attention knowledge* selection $\mathbb{L}_{\text{sel}}$ | $\{L^{(m)}\}$ |
| Temperature $\tau$ | 8.0 |
| Loss weights $(\lambda, \gamma_{\text{Emb}}, \gamma_{\text{Attn}})$ | $(0.6, 0.4, 0.4)$ |
| Epochs | 50 |
| Optimizer | AdamW |
| Adam $\beta$ | $(0.9, 0.999)$ |
| Adam $\epsilon$ | 1e-8 |
| Scheduler | CosineAnnealingLR |
| Minimal learning rate | 1e-6 |
| Label smoothing | 0.1 |

## D. Detailed Algorithmic Implementation of EmBrace

The complete execution flow of our EmBrace framework is detailed in Algorithm 1.

| Datasets | Batch Size | Learning Rate | Weight Decay | Dropout |
|---|---|---|---|---|
| BCICIV-2A | 64 | 1e-4 | 5e-2 | 0.2 |
| PhysioNet-MI | 16 | 1e-4 | 1e-3 | 0.1 |
| SHU-MI | 64 | 3e-5 | 5e-2 | 0.2 |
| SEED-V | 64 | 1e-4 | 5e-2 | 0.1 |
| FACED | 8 | 1e-4 | 5e-2 | 0.1 |
| CHB-MIT | 64 | 5e-5 | 1e-2 | 0.3 |
| ISRUC-S1 | 2 | 1e-5 | 1e-2 | 0.1 |
| ISRUC-S3 | 2 | 1e-5 | 1e-2 | 0.1 |
| BCIC2020-3 | 32 | 5e-5 | 5e-2 | 0.1 |
| Mumtaz2016 | 64 | 1e-4 | 5e-2 | 0.1 |
| MentalArithmetic | 64 | 3e-5 | 1e-3 | 0.1 |
| ATTENTION | 32 | 1e-4 | 1e-4 | 0.1 |

## E. Efficiency and Complexity Analysis

In this section, we provide a formal mathematical comparison between Exhaustive Fine-tuning (EFT) and EmBrace. We define EFT as the process of independently performing task-specific fine-tuning for every individual EEG Foundation Model $\mathbf{F}^{(m)}$ within the *model pool* $\mathcal{M}$. This approach implies that the total resource consumption, whether measured in cumulative training time for **serial execution** or total hardware allocation for **parallel execution**, is a summation of the requirements of all $M$ models.

### E.1. Theoretical Formulation

Let $M$ denote the number of models in the *model pool* $\mathcal{M}$, and $N$ represent the total number of samples in dataset $\mathcal{D}$. We use $\Phi_f$ and $\Phi_b$ to represent the FLOPs for a forward and backward pass, respectively, where $\Phi_b \approx 2\Phi_f$.

**Space Complexity.** The peak memory footprint $\mathcal{S}$ is determined by the activation maps required for gradient computation. For parallel EFT, since each model must be capable of independent optimization, the total capacity required to support the entire pool scales with $M$. For a model with sequence length $P_{seq}$ and dimension $d$, the complexity is denoted by a generic function $\Lambda(P_{seq}, d)$.

$$\mathcal{S}_{\text{EFT}} = \sum_{m=1}^{M} O(\Lambda_m(P_{seq}^{(m)}, d^{(m)}) + |\theta_m|) \approx O(M \cdot \Lambda(P_{seq}, d)) \tag{23}$$

$$\mathcal{S}_{\text{Ours}} = O(\Lambda_{\mathbf{T}}(P_{seq}^{(\mathbf{T})}, d^{(\mathbf{T})}) + |\theta_{\mathbf{T}}|) + O(M \cdot P_{seq} \cdot d_{align}) \approx O(1 \cdot \Lambda(P_{seq}, d)) \tag{24}$$

As detailed in **Algorithm 1**, the term $O(M \cdot P_{seq} \cdot d_{align})$ represents the frozen unified individual knowledge $\tilde{\mathbf{K}}_k^{(m)}$. By caching these representations through a serial forward pass initially, EmBrace ensures that only the target model $\mathbf{T}$ and its associated gradients occupy GPU VRAM during the subsequent parallel training, effectively decoupling space complexity from the size of the *model pool* $M$.

**Time Complexity.** The total computational cost is denoted as $\mathbb{C}$. Based on the execution flow of **Algorithm 1**, the complexity is partitioned into three functional phases: **1) Knowledge Externalization**: A one-time serial forward pass of all $N$ samples through $M$ backbones to cache the knowledge $\tilde{\mathbf{K}}_k^{(m)}$. Its cost is $\mathbb{C}_{ext} = N \cdot \sum_{m=1}^{M} \Phi_f^{(m)}$; **2) Carrier Selection**: This phase identifies the optimal $\mathbf{T}$ via the probe set $\mathcal{D}_{probe}$ as detailed in Section Section 3.3. Since all features are pre-cached, this stage involves only a marginal one-time overhead $\mathbb{C}_{sel}$, which is negligible compared to total FLOPs; **3) Knowledge Fusion**: The target model $\mathbf{T}$ is optimized while knowledge is retrieved from the static cache. The per-epoch cost of this parallel optimization is $\mathbb{C}_{sync} = N \cdot (3\Phi_f^{(\mathbf{T})} + \epsilon)$, where $\epsilon$ represents the overhead of modules $\Psi$ and $\mathcal{G}$.

The speedup ratio $\mathcal{R}_{\text{speedup}}$ relative to serial EFT is formulated as:

$$\mathcal{R}_{\text{speedup}}(E) = \frac{\mathbb{C}_{\text{EFT}}}{\mathbb{C}_{\text{Ours}}} = \frac{3E \cdot N \cdot \sum_{m=1}^{M} \Phi_f^{(m)}}{\mathbb{C}_{ext} + \mathbb{C}_{sel} + E \cdot \mathbb{C}_{sync}} \tag{25}$$

---

**Algorithm 1** Knowledge Fusion for EEG Foundation Model

---

**Input:** Raw dataset $\mathcal{D}_{\text{raw}} = \{(\mathbf{X}_{\text{raw}}^{(n)}, \mathbf{y}^{(n)})\}$, pre-trained *model pool* $\mathcal{M} = \{\mathbf{F}^{(m)}\}_{m=1}^{M}$, coefficients $\lambda, \gamma_k$.
**Output:** Optimized carrier model $\mathbf{T}$.

---

**Stage 0: Preprocessing & Meta-feature Caching**
1: Initialize processed dataset $\mathcal{D}_{\text{proc}} \leftarrow \emptyset$.
2: **for** each sample $(\mathbf{X}_{\text{raw}}^{(n)}, \mathbf{y}^{(n)}) \in \mathcal{D}_{\text{raw}}$ **do**
3:     Segment $\mathbf{X}_{\text{raw}}^{(n)}$ into patches $\mathbf{X}^{(n)}$ (Eq. (1)).
4:     Extract physiological meta-features $\mathbf{X}_{\text{meta}}^{(n)}$ (Eq. (12)).
5:     Construct $\mathcal{D}_{\text{proc}} \leftarrow \mathcal{D}_{\text{proc}} \cup \{(\mathbf{X}^{(n)}, \mathbf{X}_{\text{meta}}^{(n)}, \mathbf{y}^{(n)})\}$.
6: **end for**
7: Partition $\mathcal{D}_{\text{proc}}$ into training set $\mathcal{D}_{\text{train}}$, validation set $\mathcal{D}_{\text{val}}$, and test set $\mathcal{D}_{\text{test}}$.

---

**Stage 1: Multi-scale Knowledge Externalization & Unification (Section 3.1)**
8: **for** each model $\mathbf{F}^{(m)} \in$ *model pool* $\mathcal{M}$ **do**
9:     Freeze $\mathbf{F}^{(m)}$ and set it to evaluation mode.
10:     Define the on-demand unified knowledge extractor $\tilde{\mathbf{K}}_k^{(m)}(\mathbf{X}) = \Psi_k^{(m)}(\mathbf{F}^{(m)}, \mathbf{X})$ (Eq. (8)–(9), Eq. (10)–(11)).
11: **end for**

---

**Stage 2: Efficient Carrier Selection via Structural Compatibility (Section 3.3)**
12: Construct probe set $\mathcal{D}_{\text{probe}} = \text{StratifiedSample}(\mathcal{D}_{\text{train}}, \text{ratio} = 0.2)$.
13: **for** each model $\mathbf{F}^{(m)} \in$ *model pool* $\mathcal{M}$ **do**
14:     Compute final representations $\mathbf{Z}_L^{(m)}$ on $\mathcal{D}_{\text{probe}}$ to construct feature matrix $\mathbf{R}^{(m)}$.
15:     Compute compatibility score $\xi^{(m)}$ via alignment mechanism (Eq. (15)).
16: **end for**
17: Identify optimal carrier index $m^* = \arg\max_m \xi^{(m)}$.
18: Initialize a trainable carrier copy $\mathbf{T} \leftarrow \text{Copy}(\mathbf{F}^{(m^*)})$ with parameters $\theta_{\mathbf{T}}$ (Eq. (16)).

---

**Stage 3: Adaptive Fusion & Knowledge Internalization (Section 3.2 & Section 3.3)**
19: Unfreeze $\theta_{\mathbf{T}}$ for joint optimization.
20: **repeat**
21:     Sample mini-batch $\{(\mathbf{X}, \mathbf{X}_{\text{meta}}, \mathbf{y})\}$ from $\mathcal{D}_{\text{train}}$.
22:     Derive adaptive weights $\{w^{(m)}\}_{m=1}^{M} = \mathcal{G}(\mathbf{X}_{\text{meta}})$ (Eq. (13)).
23:     Compute source knowledge $\tilde{\mathbf{K}}_k^{(m)}(\mathbf{X})$ on demand using frozen source models in $\mathcal{M}$.
24:     Synthesize fused reference $\mathbf{K}_k^{(*)} = \sum_{m=1}^{M} w^{(m)} \tilde{\mathbf{K}}_k^{(m)}(\mathbf{X})$ (Eq. (14)).
25:     Externalize and unify carrier knowledge $\mathbf{K}_k^{(\mathbf{T})} = \Psi_k^{(\mathbf{T})}(\mathbf{T}, \mathbf{X})$ from the current state of carrier $\mathbf{T}$.
26:     Compute joint objective $\mathcal{J} = \lambda \mathcal{L}_{\text{task}} + \sum_k \gamma_k \mathcal{L}_{\text{fusion}}(\mathbf{K}_k^{(\mathbf{T})}, \mathbf{K}_k^{(*)})$ (Eq. (17)).
27:     Update carrier parameters $\theta_{\mathbf{T}}$ via gradient descent.
28: **until** convergence or max epochs reached

---

29: Evaluate final performance on the test set $\mathcal{D}_{\text{test}}$.
30: **return** Optimized carrier model $\mathbf{T}$.

---

As the number of training epochs $E$ increases, the constant overheads $\mathbb{C}_{ext}$ and $\mathbb{C}_{sel}$ from the initial serial passes are amortized. For standard convergence schedules ($E \approx 50$), the speedup ratio effectively scales with $M$:

$$\mathcal{R}_{\text{speedup}} \approx \frac{3 \sum_{m=1}^{M} \Phi_{f,m}}{3\Phi_f^{(\mathbf{T})} + \epsilon} \propto M \qquad (26)$$

### E.2. Backbone Parameter Profiling

Table 7 provides a granular breakdown of the backbone parameters for all models within the pool $\mathcal{M}$ specifically for the SEED-V task. By isolating the backbones from the task-specific classification heads, we demonstrate how EmBrace performs knowledge fusion from diverse foundation models with minimal trainable overhead.

*Table 7.* Detailed parameter analysis of individual Backbones and Fusion overhead on SEED-V.

| Category | Component / Source | Backbone Params | Status | Trainable |
|---|---|---|---|---|
| Model Pool $\mathcal{M}$ | Source 01: LaBraM | 5.8 M | Frozen | - |
| | Source 02: CBraMod | 4.9 M | Frozen | - |
| | Source 03: CodeBrain | 15.2 M | Frozen | - |
| | **EFT Total (Parallel)** | **25.9 M** | **Active** | **25.9 M** |
| EmBrace (Ours) | Target: CodeBrain | 15.2 M | Active | 15.2 M |
| | Source Bridge | 0.2 M | Active | 0.2 M |
| | Target Bridge | 0.2 M | Active | 0.2 M |
| | Fusion Weight | 33 | Active | < 0.1 M |
| | **EmBrace Total** | **15.6 M** | **Active** | **15.6 M** |

**Result Analysis.** The profiling results on SEED-V illustrate the structural efficiency of our approach. While the source pool incorporates three distinct backbones with a cumulative size of 25.9M parameters, EmBrace keeps these heavy models frozen. Consequently, in a parallel execution setting where EFT would require gradient updates for the entire 25.9M parameters, our framework only activates 15.6M parameters. This represents a marginal **fusion** overhead $\epsilon$ of only 0.4M relative to the standalone target carrier, confirming that we can leverage a multi-model knowledge while maintaining the hardware requirements of a single model.

### E.3. Empirical Validation of Computational Efficiency

*Table 8.* Empirical Comparison of Training Load in FLOPs (M).

| Input Dimension $(P_{seq} = C \times N_p)$ | Configuration | Total $\Phi_f^{(*)}$ $\Phi_f^{(\mathbf{T})} + \epsilon$ | Increment $\epsilon$ Value (vs. $\Phi_f^{(\mathbf{T})}$) | Speedup $\sum \Phi_f^{(m)}/\Phi_f^{(*)}$ |
|---|---|---|---|---|
| SEED-V $P_{seq} = 62 \times 1$ $M = 3$ | Source 01: LaBraM | 385.99 | - | - |
| | Source 02: CBraMod | 255.20 | - | - |
| | Source 03: CodeBrain | 180.18 | - | - |
| | **EFT Total $\sum \Phi_f^{(m)}$** | 821.37 | - | **1.00×** |
| | EmBrace Target: LaBraM | 431.84 | + 45.86 (11.9%) | 1.90× |
| | EmBrace Target: CBraMod | 301.06 | + 45.86 (18.0%) | 2.73× |
| | EmBrace Target: CodeBrain | 226.04 | + 45.86 (25.4%) | 3.63× |
| | **Average Speedup** | - | - | **2.75×** |
| FACED $P_{seq} = 32 \times 10$ $M = 3$ | Source 01: LaBraM | 2364.28 | - | - |
| | Source 02: CBraMod | 1393.55 | - | - |
| | Source 03: CodeBrain | 1006.30 | - | - |
| | **EFT Total $\sum \Phi_f^{(m)}$** | 4764.13 | - | **1.00×** |
| | EmBrace Target: LaBraM | 2602.86 | + 238.58 (10.1%) | 1.83× |
| | EmBrace Target: CBraMod | 1632.13 | + 238.58 (17.1%) | 2.92× |
| | EmBrace Target: CodeBrain | 1244.88 | + 238.58 (23.7%) | 3.83× |
| | **Average Speedup** | - | - | **2.86×** |

Table 8 provides empirical evidence for the efficiency of EmBrace by profiling forward FLOPs $\Phi_f$ across different input

scales. The analysis utilizes input tensors that strictly mirror the channel counts and sequence lengths $P_{seq}$ of the SEED-V and FACED datasets to compare the online synchronization cost against the EFT baseline.

**Result Analysis.** The results confirm the theoretical efficiency gains of the EmBrace framework based on the $P_{seq}$ of each dataset. **1) Performance Gain**: For computationally efficient targets such as CodeBrain, the speedup ratio reaches $3.83\times$, surpassing the model pool size $M = 3$ because the EFT baseline is heavily penalized by the more computationally intensive backbones like LaBraM. **2) High Efficiency**: Across all configurations, the average speedup is approximately $2.81\times$, which closely approaches the theoretical limit of $M = 3$ and proves the effectiveness of offlining backbones. **3) Stable Cost**: The absolute increment $\epsilon$ remains constant within the same input scale, confirming that fusion costs are governed by $P_{seq}$ rather than carrier complexity. This ensures predictable scaling even when high capacity models are adopted as carriers to leverage their extensive parameter space at a lower computational cost.

## F. Detailed Implementation of Knowledge Unification

This section provides the mathematical details for aligning internal representations across heterogeneous architectures. To integrate $\mathbf{K}_{\text{Attn}}^{(m)}$ from multiple source models, we address structural differences in their attention mechanisms via the *aligner* module to ensure a synchronized coordinate space $\tilde{\mathbf{K}}_{\text{Attn}}^{(m)} \in \mathcal{V}_{\text{Attn}}$ for effective knowledge fusion.

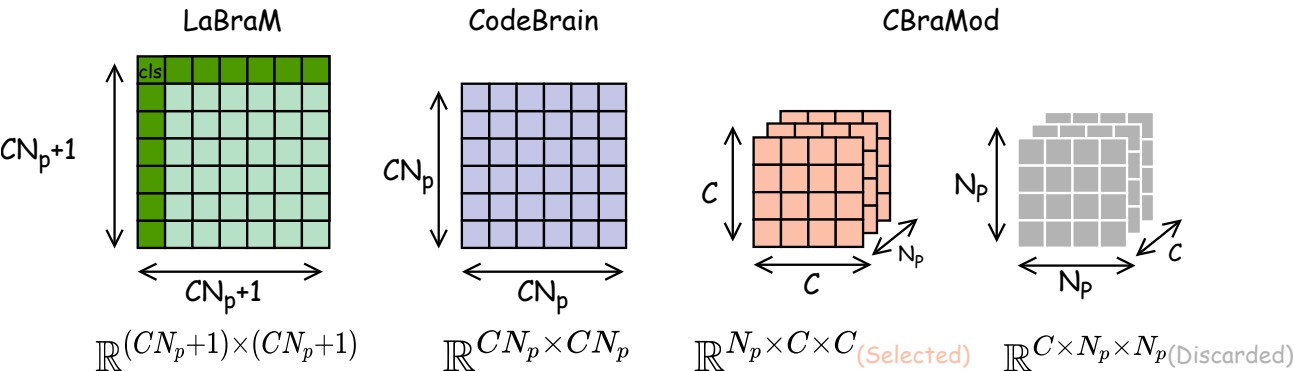

*Figure 4.* Heterogeneous attention representations across different architectures.

Structural incompatibility of $\mathbf{K}_{\text{Attn}}^{(m)}$ presents the primary challenge. As defined in Section Section 3.1, the actual implementation of $\mathbf{S}_{\ell}^{(m)}$ varies by architecture. CodeBrain yields a flattened attention map $\mathbf{S} \in \mathbb{R}^{CN_p \times CN_p}$ via its sliding window mechanism. LaBraM produces an augmented map $\mathbf{S} \in \mathbb{R}^{(CN_p+1) \times (CN_p+1)}$ due to the `[CLS]` token, necessitating a CLS-Exclusion step to extract the $\mathbb{R}^{CN_p \times CN_p}$ sub-matrix. CBraMod generates two parallel 3D attention stacks, specifically a patch-level stack $\in \mathbb{R}^{C \times N_p \times N_p}$ and a channel-level stack $\in \mathbb{R}^{N_p \times C \times C}$. We select the patch-level branch to focus on localized spatial interactions and discard the channel-level stack for consistency.

To harmonize this structural diversity, the *aligner* maps $\mathbf{K}_{\text{Attn}}^{(m)}$ into unified target spaces $\tilde{\mathbf{K}}_{\text{Attn}}^{(m)} \in \mathcal{V}_{\text{Attn}}$. CodeBrain and LaBraM inherently operate in the sequence space $\mathbb{R}^{CN_p \times CN_p}$. **Thus, we propose three methods to reconstruct the 3D stacks of CBraMod into this global space. Additionally, Method 4 provides a universal solution to condense all models into a unified channel space $\mathbb{R}^{C \times C}$ via model-specific aggregation.** Note that only one alignment method is selected for a given experimental setting.

**Method 1. Block-Diagonal**: This method constructs a sparse global matrix by anchoring localized patch maps along the main diagonal. Specifically, the matrix is initialized with zeros rather than a negative-infinity mask. This zero-initialization maintains a neutral activation background, ensuring that non-local regions remain computationally addressable. Such a structure facilitates smoother gradient flow during knowledge fusion and preserves the potential for the model to capture dependencies during the optimization process.

$$\tilde{\mathbf{K}}_{\text{Attn}}^{(m)} \in \mathbb{R}^{CN_p \times CN_p} = \text{diag}(\mathbf{S}_1, \dots, \mathbf{S}_C) \tag{27}$$

**Method 2. Global Tiling**: This strategy projects the mean spatial map across the sequence grid. It uses a Kronecker product with an all-ones matrix $\mathbf{J}$ to enforce a spatial-invariance prior. By distributing average intra-patch

correlations throughout the entire spatio-temporal sequence, it stabilizes the global attention context for knowledge fusion.

$$\tilde{\mathbf{K}}_{\text{Attn}}^{(m)} \in \mathbb{R}^{CN_p \times CN_p} = \mathbf{J}_{C \times C} \otimes \left( \frac{1}{C} \sum_{k=1}^{C} \mathbf{S}_k \right) \tag{28}$$

**Method 3. Row-Broadcasting**: This mechanism replicates localized channel maps row-wise via Kronecker expansion to project local dependencies into a global context. It distributes each patch's feature-wise attention across the entire sequence to approximate the global temporal evolution.

$$\tilde{\mathbf{K}}_{\text{Attn}}^{(m)} \in \mathbb{R}^{CN_p \times CN_p} = \begin{bmatrix} \mathbf{S}_1 \otimes \mathbf{J}_{1 \times C} \\ \vdots \\ \mathbf{S}_C \otimes \mathbf{J}_{1 \times C} \end{bmatrix} \tag{29}$$

**Method 4. Patch-Pooling**: This approach aggregates patch-specific details into a sequence-invariant channel space. By collapsing spatial dimensions, it isolates core inter-channel dynamics. This focuses the knowledge fusion on fundamental neural oscillation patterns and rhythmic coupling across channels.

$$\tilde{\mathbf{K}}_{\text{Attn}}^{(m)} \in \mathbb{R}^{C \times C} = \begin{cases} \frac{1}{N_p} \sum_{k=1}^{N_p} \mathbf{S}_k, & \text{for CBraMod} \\ \frac{1}{N_p^2} \sum_{i=1}^{N_p} \sum_{j=1}^{N_p} \mathbf{S}_{i,j}, & \text{for CodeBrain/LaBraM} \end{cases} \tag{30}$$

where $\mathbf{J}$ denotes an all-ones matrix, $\mathbf{S}_k$ represents the relevant attention slice, and $\mathbf{S}_{i,j} \in \mathbb{R}^{C \times C}$ is the sub-matrix block within the flattened maps of CodeBrain and LaBraM.

## G. Meta-Features Quantification Details

To incorporate physiological priors into the model fusion process, the fusor $\mathcal{G}$ extracts a structured meta-feature matrix $\mathbf{X}_{\text{meta}} \in \mathbb{R}^{C \times 10}$ from the raw EEG signal $\mathbf{X}_{\text{raw}}$. For each channel $c$, the extraction process decomposes the signal into three physiological dimensions as formulated in Eq. (12):

1) **Temporal Domain**: captures baseline statistics and signal volatility through mean $\mu$, standard deviation $\sigma$, and zero-crossing rate ZCR, as specified in $\mathbf{v}_{\text{time}}$.

2) **Spectral Domain**: quantifies log-scale power density across five bands $(P_\delta, P_\theta, P_\alpha, P_\beta, P_\gamma)$ via Welch's method, as specified in $\mathbf{v}_{\text{freq}}$. The raw band power is denoted as $Q_b = \sum_{f \in b} S_{xx}(f)$ for frequency band $b$, and the corresponding spectral feature is computed as $P_b = 10 \log_{10}(Q_b)$. The power spectral density $S_{xx}(f)$ is estimated by averaging the squared discrete Fourier transforms of multiple overlapping signal segments, each smoothed by a Hann window. This process of averaging modified periodograms ensures robust power estimation by reducing spectral variance and noise.

3) **Rhythmic Domain**: evaluates nonlinear interactions via raw band-power ratios $\mathcal{R}_{\alpha/\beta} = Q_\alpha/Q_\beta$ and $\mathcal{R}_{\theta/\alpha} = Q_\theta/Q_\alpha$ to reflect complex transitions in brain states, as specified in $\mathbf{v}_{\text{rhy}}$.

*Table 9.* Detailed specification of the EEG meta-features.

| Domain | Feature | Description | Significance | Formulation |
|---|---|---|---|---|
| Temporal | Mean | Average amplitude | Baseline offset | $\mu = \sum x_t / T$ |
| | Std | Fluctuation intensity | Global activation | $\sigma = \sqrt{\sum (x_t - \mu)^2 / T}$ |
| | ZCR | Zero-crossing rate | Temporal complexity | $\sum \mathbb{I}(x_{t+1} x_t < 0)/(T-1)$ |
| Spectral | $P_\delta$ | 0.5–4 Hz band | Deep sleep (Carskadon & Dement, 2011) | |
| | $P_\theta$ | 4–8 Hz band | Drowsiness (Santamaria & Chiappa, 1987) | |
| | $P_\alpha$ | 8–13 Hz band | Relaxed wake (Klimesch, 1999) | $P_b = 10 \log_{10}(Q_b)$ |
| | $P_\beta$ | 13–30 Hz band | Active state (Engel & Fries, 2010) | |
| | $P_\gamma$ | 30–45 Hz band | Feature binding (Tallon-Baudry & Bertrand, 1999) | |
| Rhythmic | $\mathcal{R}_{\alpha/\beta}$ | Alpha–beta ratio | Mental fatigue (Jap et al., 2009) | $Q_\alpha/Q_\beta$ |
| | $\mathcal{R}_{\theta/\alpha}$ | Theta–alpha ratio | Cognitive load (Raufi & Longo, 2022) | $Q_\theta/Q_\alpha$ |

# H. Detailed Results on Downstream BCI Tasks

## H.1. Motor Imagery Classification

**BCIC-IV-2A** contains EEG data from 9 subjects performing four motor imagery tasks: left hand, right hand, both feet, and tongue (Brunner et al., 2008). Recorded via 22 electrodes at 250 Hz, the signals were band-pass filtered between 0.3 Hz and 50 Hz. Following the protocol in (Wang et al., 2025), we extracted a 4-second window from 2 to 6 seconds for each trial and resampled the data to 200 Hz, totaling 5,088 samples. Subjects 1–5, 6–7, and 8–9 were used for training, validation, and testing respectively. As shown in Table 10, EmBrace attains 0.5047 in ACC-B and 0.3396 in Kappa. While EEGNet remains competitive on this dataset, EmBrace outperforms all evaluated foundation models, exceeding the second-best model, CBraMod, by 2.54% in ACC-B and 5.17% in Kappa.

*Table 10.* Performance comparison on the BCIC-IV-2A (4-Class) dataset.

| Methods | ACC-B | W-F1 | Kappa |
|---|---|---|---|
| EEGNet | $0.5668 \pm 0.0112$ | $0.5498 \pm 0.0161$ | $0.4225 \pm 0.0149$ |
| EEG-Deformer | $0.5488 \pm 0.0170$ | $0.5260 \pm 0.0227$ | $0.3984 \pm 0.0227$ |
| SERA | $0.3188 \pm 0.0381$ | $0.2591 \pm 0.0381$ | $0.0917 \pm 0.0508$ |
| SPaRCNet | $0.5175 \pm 0.0416$ | $0.4996 \pm 0.0514$ | $0.3567 \pm 0.0555$ |
| ContraWR | $0.5293 \pm 0.0281$ | $0.5046 \pm 0.0320$ | $0.3725 \pm 0.0375$ |
| CNN-Transformer | $0.4905 \pm 0.0237$ | $0.4735 \pm 0.0267$ | $0.3206 \pm 0.0316$ |
| FFCL | $0.4941 \pm 0.0424$ | $0.4721 \pm 0.0549$ | $0.3255 \pm 0.0566$ |
| ST-Transformer | $0.4686 \pm 0.0262$ | $0.4639 \pm 0.0275$ | $0.2914 \pm 0.0362$ |
| BIOT | $0.4080 \pm 0.0173$ | $0.3403 \pm 0.0213$ | $0.2106 \pm 0.0231$ |
| LaBraM | $0.3453 \pm 0.0089$ | $0.3110 \pm 0.0449$ | $0.1271 \pm 0.0119$ |
| CBraMod | $\underline{0.4922} \pm 0.0354$ | $\underline{0.4679} \pm 0.0459$ | $\underline{0.3229} \pm 0.0473$ |
| CodeBrain | $0.3747 \pm 0.0233$ | $0.3007 \pm 0.0295$ | $0.1662 \pm 0.0311$ |
| EmBrace | $\mathbf{0.5047} \pm 0.0203$ | $\mathbf{0.4870} \pm 0.0240$ | $\mathbf{0.3396} \pm 0.0271$ |

**PhysioNet-MI** is a high-density motor imagery dataset featuring 64-channel recordings from 109 subjects performing four tasks (left/right fist, both fists, and both feet) (Schalk et al., 2004). Following the 200 Hz resampling and 4-second windowing protocol in (Wang et al., 2025), we utilized a subject-independent split (1–70/71–89/90–109) to assess performance across an extensive population. As shown in Table 11, EmBrace reaches 0.6268 in ACC-B and 0.5023 in Kappa. Notably, the performance advantage of EmBrace over individual EFM baselines is further amplified in this 64-channel configuration compared to lower-density datasets. By surpassing the runner-up CBraMod by 0.65% in ACC-B and 0.87% in Kappa, our framework demonstrates a superior capacity to manage increased channel density and subject variability, maintaining a stable lead over both monolithic foundation models and task-specific architectures like EEG-Deformer.

*Table 11.* Performance comparison on the PhysioNet-MI (4-Class) dataset.

| Methods | ACC-B | W-F1 | Kappa |
|---|---|---|---|
| EEGNet | $0.6070 \pm 0.0107$ | $0.6084 \pm 0.0105$ | $0.4760 \pm 0.0143$ |
| EEG-Deformer | $0.6166 \pm 0.0062$ | $0.6172 \pm 0.0060$ | $0.4888 \pm 0.0082$ |
| SERA | $0.4549 \pm 0.0090$ | $0.4376 \pm 0.0148$ | $0.2727 \pm 0.0119$ |
| SPaRCNet | $0.5996 \pm 0.0058$ | $0.5991 \pm 0.0041$ | $0.4662 \pm 0.0078$ |
| ContraWR | $0.5001 \pm 0.0088$ | $0.5006 \pm 0.0081$ | $0.3334 \pm 0.0118$ |
| CNN-Transformer | $0.3988 \pm 0.0256$ | $0.3800 \pm 0.0305$ | $0.1986 \pm 0.0341$ |
| FFCL | $0.5047 \pm 0.0052$ | $0.5043 \pm 0.0043$ | $0.3395 \pm 0.0069$ |
| ST-Transformer | $0.6069 \pm 0.0129$ | $0.6073 \pm 0.0145$ | $0.4758 \pm 0.0171$ |
| BIOT | $0.3835 \pm 0.0123$ | $0.3826 \pm 0.0136$ | $0.1781 \pm 0.0164$ |
| LaBraM | $0.5872 \pm 0.0189$ | $0.5906 \pm 0.0179$ | $0.4496 \pm 0.0252$ |
| CBraMod | $\underline{0.6203} \pm 0.0153$ | $\underline{0.6199} \pm 0.0165$ | $\underline{0.4936} \pm 0.0205$ |
| CodeBrain | $0.6070 \pm 0.0043$ | $0.6090 \pm 0.0052$ | $0.4760 \pm 0.0057$ |
| EmBrace | $\mathbf{0.6268} \pm 0.0040$ | $\mathbf{0.6272} \pm 0.0038$ | $\mathbf{0.5023} \pm 0.0053$ |

**SHU-MI** is a binary motor imagery dataset (left vs. right hand) designed to assess cross-session stability, comprising 32-channel recordings from 25 subjects across five separate days (Ma et al., 2022). Following a 200 Hz resampling and 4-second windowing protocol, we utilized a subject-independent split (1-15/16-20/21-25) to evaluate the model's resilience to multi-day physiological variations. As detailed in Table 12, EmBrace demonstrates superior discriminative power,

particularly in probabilistic metrics. While CodeBrain maintains a marginal lead in Balanced Accuracy (0.6262 vs. 0.6245), EmBrace establishes a new state-of-the-art in AUC-based performance, outperforming CodeBrain by 0.91% in AUC-PR and achieving the highest AUROC (0.6876) among all evaluated methods. These results indicate that despite the high cross-session variability inherent in SHU-MI, our meta-guided fusion provides a more robust and better-calibrated separation of motor intentions, ensuring more stable performance for practical, long-term BCI applications.

Table 12. Performance comparison on the SHU-MI (2-Class) dataset.

| Methods | ACC-B | AUC-PR | AUROC |
| --- | --- | --- | --- |
| EEGNet | 0.6258 ± 0.0073 | 0.6643 ± 0.0087 | 0.6781 ± 0.0076 |
| EEG-Deformer | 0.6309 ± 0.0080 | 0.6750 ± 0.0224 | 0.6908 ± 0.0180 |
| SERA | 0.5437 ± 0.0105 | 0.5677 ± 0.0204 | 0.5678 ± 0.0237 |
| SPaRCNet | 0.6155 ± 0.0131 | 0.6674 ± 0.0084 | 0.6765 ± 0.0155 |
| ContraWR | 0.5453 ± 0.0442 | 0.6220 ± 0.0070 | 0.6282 ± 0.0069 |
| CNN-Transformer | 0.5387 ± 0.0247 | 0.5743 ± 0.0093 | 0.5746 ± 0.0121 |
| FFCL | 0.5442 ± 0.0286 | 0.6050 ± 0.0116 | 0.6122 ± 0.0099 |
| ST-Transformer | 0.5838 ± 0.0142 | 0.6476 ± 0.0151 | 0.6397 ± 0.0194 |
| BIOT | 0.5605 ± 0.0252 | 0.5984 ± 0.0292 | 0.5980 ± 0.0252 |
| LaBraM | 0.6084 ± 0.0175 | 0.6697 ± 0.0363 | 0.6643 ± 0.0338 |
| CBraMod | 0.6196 ± 0.0048 | 0.6922 ± 0.0085 | 0.6865 ± 0.0052 |
| CodeBrain | **0.6262** ± 0.0071 | 0.6907 ± 0.0128 | 0.6800 ± 0.0119 |
| EmBrace | 0.6245 ± 0.0114 | **0.6998** ± 0.0226 | **0.6876** ± 0.0225 |

## H.2. Emotion Recognition

**SEED-V** is a large-scale benchmark for 5-class emotion recognition (happy, sad, neutral, disgust, and fear), featuring 62-channel EEG signals from 16 subjects (Liu et al., 2021). Following the experimental setup in (Wang et al., 2025), we resampled the 1000 Hz recordings to 200 Hz and segmented them into 1-second non-overlapping windows, yielding a total of 117,744 samples with a 5:5:5 trial split per session. As presented in Table 13, EmBrace delivers the best overall performance, outperforming the current strongest baseline, CodeBrain, across all three evaluation metrics. Specifically, it achieves a Balanced Accuracy of 0.4156 and a Kappa of 0.2746. Although the improvements on this challenging 5-class task are incremental, the consistent lead across ACC-B, W-F1, and Kappa underscores the efficacy of our meta-guided knowledge fusion. These results suggest that EmBrace can better refine emotional representations amidst the high inter-subject variability inherent in SEED-V, providing more robust affective signatures than existing EEG foundation models.

Table 13. Performance comparison on the SEED-V (5-Class) dataset.

| Methods | ACC-B | W-F1 | Kappa |
| --- | --- | --- | --- |
| EEGNet | 0.3014 ± 0.0053 | 0.2918 ± 0.0079 | 0.1261 ± 0.0077 |
| EEG-Deformer | 0.3149 ± 0.0079 | 0.2911 ± 0.0210 | 0.1377 ± 0.0123 |
| SERA | 0.2792 ± 0.0173 | 0.2635 ± 0.0296 | 0.1025 ± 0.0181 |
| SPaRCNet | 0.3386 ± 0.0038 | 0.3435 ± 0.0041 | 0.1691 ± 0.0055 |
| ContraWR | 0.3795 ± 0.0148 | 0.3783 ± 0.0177 | 0.2218 ± 0.0199 |
| CNN-Transformer | 0.2627 ± 0.0173 | 0.2469 ± 0.0199 | 0.0873 ± 0.0224 |
| FFCL | 0.3721 ± 0.0229 | 0.3644 ± 0.0362 | 0.2094 ± 0.0346 |
| ST-Transformer | 0.2501 ± 0.0069 | 0.2503 ± 0.0068 | 0.0644 ± 0.0075 |
| BIOT | 0.3859 ± 0.0097 | 0.3916 ± 0.0099 | 0.2328 ± 0.0124 |
| LaBraM | 0.3688 ± 0.0145 | 0.3740 ± 0.0161 | 0.2124 ± 0.0187 |
| CBraMod | 0.3870 ± 0.0052 | 0.3950 ± 0.0068 | 0.2384 ± 0.0067 |
| CodeBrain | 0.4126 ± 0.0020 | 0.4243 ± 0.0020 | 0.2730 ± 0.0023 |
| EmBrace | **0.4156** ± 0.0029 | **0.4250** ± 0.0027 | **0.2746** ± 0.0036 |

**FACED** is a large-scale dataset for fine-grained emotion recognition, containing 32-channel EEG signals from 123 subjects (Chen et al., 2023). The task involves classifying nine emotional states: neutral, plus four positive and four negative categories. We followed the 200 Hz resampling and 10-second windowing protocol in (Wang et al., 2025), using a subject-independent split (1–80/81–100/101–123). As shown in Table 14, EmBrace reaches 0.6136 in ACC-B and 0.5608 in Kappa, outperforming the most competitive baseline, CodeBrain, by a margin of over 3% in accuracy. These results demonstrate the performance of EmBrace on a diverse population across fine-grained affective states.

*Table 14.* Performance comparison on the FACED (9-Class) dataset.

| Methods | ACC-B | W-F1 | Kappa |
|---|---|---|---|
| EEGNet | $0.2282 \pm 0.0167$ | $0.1781 \pm 0.0248$ | $0.1310 \pm 0.0175$ |
| EEG-Deformer | $0.3439 \pm 0.0251$ | $0.3450 \pm 0.0257$ | $0.2614 \pm 0.0269$ |
| SERA | $0.1444 \pm 0.0149$ | $0.0789 \pm 0.0330$ | $0.0384 \pm 0.0183$ |
| SPaRCNet | $0.3023 \pm 0.0892$ | $0.3069 \pm 0.0905$ | $0.2154 \pm 0.1008$ |
| ContraWR | $0.1469 \pm 0.0170$ | $0.0659 \pm 0.0231$ | $0.0398 \pm 0.0191$ |
| CNN-Transformer | $0.1627 \pm 0.0064$ | $0.1204 \pm 0.0160$ | $0.0566 \pm 0.0073$ |
| FFCL | $0.2241 \pm 0.0127$ | $0.2161 \pm 0.0122$ | $0.1252 \pm 0.0137$ |
| ST-Transformer | $0.2597 \pm 0.0365$ | $0.2571 \pm 0.0341$ | $0.1670 \pm 0.0340$ |
| BIOT | $0.2590 \pm 0.0064$ | $0.2556 \pm 0.0057$ | $0.1642 \pm 0.0066$ |
| LaBraM | $0.2983 \pm 0.0357$ | $0.2986 \pm 0.0359$ | $0.2096 \pm 0.0406$ |
| CBraMod | $0.5425 \pm 0.0077$ | $0.5444 \pm 0.0077$ | $0.4831 \pm 0.0089$ |
| CodeBrain | $\underline{0.5835} \pm 0.0079$ | $\underline{0.5891} \pm 0.0094$ | $\underline{0.5290} \pm 0.0093$ |
| EmBrace | $\mathbf{0.6136} \pm 0.0055$ | $\mathbf{0.6131} \pm 0.0060$ | $\mathbf{0.5608} \pm 0.0062$ |

### H.3. Sleep Staging

**ISRUC-Sleep** is a comprehensive clinical dataset for automatic sleep stage classification based on the AASM standard, categorized into five stages: Wake, N1, N2, N3, and REM (Khalighi et al., 2016). We evaluate our framework on two subsets: **ISRUC-S1** (100 subjects with sleep disorders) and **ISRUC-S3** (10 healthy subjects). Adopting the protocol in (Wang et al., 2025), we utilized 6-channel EEG recordings (F3-A2, C3-A2, O1-A2, F4-A1, C4-A1, and O2-A1) sampled at 200 Hz. The signals were segmented into 30-second epochs and grouped into sequences of length 20 to capture critical temporal transition patterns. To ensure a fair comparison, we standardized the classification interface across all foundation models. Specifically, for architectures such as LaBraM that originally utilize a single [CLS] token, we replaced the vanilla head with feature flattening, which preserves the complete spatial temporal patch representations by flattening all output tokens into a vector. These features are subsequently processed by a one layer Transformer sequence encoder, consistent with the default heads of CBraMod and CodeBrain. Consequently, all models can benefit equally from high granularity features and sequence level dependency modeling.

As summarized in Table 15, EmBrace achieves the top performance on **ISRUC-S1**, reaching a Kappa of 0.7617 and a Weighted F1 of 0.8110. It maintains a clear lead over advanced foundation models like LaBraM and CodeBrain, suggesting that our framework effectively extracts epoch-level features that are highly conducive to sequence-level decoding.

*Table 15.* Performance comparison on the ISRUC-S1 (5-Class) dataset.

| Methods | ACC-B | W-F1 | Kappa |
|---|---|---|---|
| EEGNet | $0.6504 \pm 0.0181$ | $0.6825 \pm 0.0158$ | $0.6236 \pm 0.0229$ |
| EEG-Deformer | $0.7095 \pm 0.0169$ | $0.7360 \pm 0.0100$ | $0.6571 \pm 0.0148$ |
| SERA | $0.6860 \pm 0.0126$ | $0.7095 \pm 0.0074$ | $0.6375 \pm 0.0160$ |
| SPaRCNet | $0.7722 \pm 0.0277$ | $0.7856 \pm 0.0305$ | $0.7262 \pm 0.0377$ |
| ContraWR | $0.7935 \pm 0.0027$ | $0.7928 \pm 0.0114$ | $0.7344 \pm 0.0126$ |
| CNN-Transformer | $0.7865 \pm 0.0129$ | $0.7991 \pm 0.0078$ | $0.7447 \pm 0.0065$ |
| FFCL | $0.7822 \pm 0.0082$ | $0.7885 \pm 0.0129$ | $0.7292 \pm 0.0136$ |
| ST-Transformer | $0.6651 \pm 0.0415$ | $0.6752 \pm 0.0619$ | $0.6019 \pm 0.0704$ |
| BIOT | $0.7795 \pm 0.0061$ | $0.8028 \pm 0.0071$ | $0.7451 \pm 0.0084$ |
| LaBraM | $\underline{0.7880} \pm 0.0067$ | $\underline{0.8023} \pm 0.0107$ | $\underline{0.7455} \pm 0.0124$ |
| CBraMod | $0.7701 \pm 0.0070$ | $0.7915 \pm 0.0093$ | $0.7323 \pm 0.0061$ |
| CodeBrain | $0.7751 \pm 0.0078$ | $0.7945 \pm 0.0104$ | $0.7393 \pm 0.0147$ |
| EmBrace | $\mathbf{0.7927} \pm 0.0119$ | $\mathbf{0.8110} \pm 0.0107$ | $\mathbf{0.7617} \pm 0.0111$ |

For the **ISRUC-S3** dataset (Table 16), EmBrace remains highly competitive with an ACC-B of 0.7897. While some foundation models exhibit significant performance fluctuations on this smaller dataset, EmBrace demonstrates superior stability. This consistency across diverse populations underscores the model's ability to decode complex sleep architectures by effectively leveraging the structural dependencies between different cortical regions.

*Table 16.* Performance comparison on the ISRUC-S3 (5-Class) dataset.

| Methods | ACC-B | W-F1 | Kappa |
|---|---|---|---|
| EEGNet | $0.6607 \pm 0.0087$ | $0.5518 \pm 0.0165$ | $0.4958 \pm 0.0141$ |
| EEG-Deformer | $0.6108 \pm 0.0363$ | $0.5946 \pm 0.0533$ | $0.5052 \pm 0.0477$ |
| SERA | $0.5815 \pm 0.0592$ | $0.5115 \pm 0.0547$ | $0.4260 \pm 0.0656$ |
| SPaRCNet | $0.6001 \pm 0.0723$ | $0.5499 \pm 0.0967$ | $0.4631 \pm 0.1012$ |
| ContraWR | $0.6557 \pm 0.0221$ | $0.6311 \pm 0.0310$ | $0.5395 \pm 0.0433$ |
| CNN-Transformer | $0.7118 \pm 0.0482$ | $0.6930 \pm 0.0361$ | $0.6105 \pm 0.0504$ |
| FFCL | $0.7379 \pm 0.0267$ | $0.7199 \pm 0.0279$ | $0.6485 \pm 0.0331$ |
| ST-Transformer | $0.5175 \pm 0.0227$ | $0.3879 \pm 0.0487$ | $0.3356 \pm 0.0351$ |
| BIOT | $0.6879 \pm 0.0405$ | $0.6200 \pm 0.0362$ | $0.5511 \pm 0.0435$ |
| LaBraM | $\underline{0.7895} \pm 0.0082$ | $\underline{0.7618} \pm 0.0120$ | $\underline{0.7035} \pm 0.0133$ |
| CBraMod | $0.6069 \pm 0.0455$ | $0.5401 \pm 0.0467$ | $0.4463 \pm 0.0528$ |
| CodeBrain | $0.7013 \pm 0.0217$ | $0.6616 \pm 0.0209$ | $0.5807 \pm 0.0247$ |
| EmBrace | $\mathbf{0.7897} \pm 0.0186$ | $\mathbf{0.7692} \pm 0.0188$ | $\mathbf{0.7097} \pm 0.0233$ |

## H.4. Seizure Detection

**CHB-MIT** is a widely recognized clinical EEG benchmark containing long-term recordings from 23 pediatric subjects with intractable seizures. To address the inconsistent channel configurations across subjects and sessions, we implemented a rigorous preprocessing pipeline where we parsed clinical metadata to extract a consistent subset of 16 bipolar montage channels. For cases with incomplete channel sets, we padded missing channels with zeros to maintain a uniform input dimensionality as per our implementation. Following the BIOT protocol (Yang et al., 2023), all signals were resampled to 200 Hz and partitioned into non-overlapping 10-second windows, resulting in 326,993 samples annotated with precise seizure onset and offset metadata. We utilized a subject-independent split with subjects 1–19 for training, 20–21 for validation, and 22–23 for testing to evaluate cross-subject generalization. As summarized in Table 17, while EmBrace does not achieve the highest Balanced Accuracy, trailing CBraMod by a notable margin (0.6157 vs. 0.7166), it establishes a new state-of-the-art in probabilistic discriminative power. Specifically, EmBrace secures the highest AUC-PR (0.5274) and AUROC (0.9296) among all evaluated methods. These objective results indicate that despite a lower global accuracy, EmBrace provides superior calibration and sensitivity in identifying seizure events within highly imbalanced clinical data, outperforming foundation models like LaBraM by 25.81% in AUC-PR.

*Table 17.* Performance comparison on the CHB-MIT (2-Class) dataset.

| Methods | ACC-B | AUC-PR | AUROC |
|---|---|---|---|
| EEGNet | $0.6093 \pm 0.0429$ | $0.2890 \pm 0.0952$ | $0.9194 \pm 0.0114$ |
| EEG-Deformer | $0.6255 \pm 0.0253$ | $0.3551 \pm 0.0095$ | $0.9081 \pm 0.0130$ |
| SERA | $0.5362 \pm 0.0297$ | $0.2774 \pm 0.1055$ | $0.9143 \pm 0.0160$ |
| SPaRCNet | $0.6279 \pm 0.0817$ | $0.2889 \pm 0.1354$ | $0.8837 \pm 0.0597$ |
| ContraWR | $0.5618 \pm 0.0276$ | $0.2775 \pm 0.0676$ | $0.8672 \pm 0.0084$ |
| CNN-Transformer | $0.5294 \pm 0.0658$ | $0.2249 \pm 0.0843$ | $0.7269 \pm 0.1387$ |
| FFCL | $0.5932 \pm 0.0728$ | $0.3004 \pm 0.1036$ | $0.8598 \pm 0.0609$ |
| ST-Transformer | $0.5788 \pm 0.0527$ | $0.2544 \pm 0.0705$ | $0.8939 \pm 0.0154$ |
| BIOT | $0.5764 \pm 0.0693$ | $0.2981 \pm 0.0088$ | $0.8734 \pm 0.0095$ |
| LaBraM | $0.5776 \pm 0.0089$ | $0.4192 \pm 0.0929$ | $0.8997 \pm 0.0384$ |
| CBraMod | $\mathbf{0.7166} \pm 0.0241$ | $\underline{0.4779} \pm 0.1830$ | $\underline{0.9025} \pm 0.0355$ |
| CodeBrain | $\underline{0.6265} \pm 0.0432$ | $0.3848 \pm 0.0588$ | $0.8690 \pm 0.0271$ |
| EmBrace | $0.6157 \pm 0.0913$ | $\mathbf{0.5274} \pm 0.1725$ | $\mathbf{0.9296} \pm 0.0204$ |

## H.5. Imagined Speech

**BCIC2020-3** is a dataset for imagined speech classification, released as part of the 2020 International BCI Competition (Jeong et al., 2022). It contains 64-channel EEG recordings from 15 subjects who were instructed to imagine the silent pronunciation of five words or phrases: "hello", "help me", "stop", "thank you", and "yes". Following the standard competition split, we utilized 60 trials per class for training, 10 for validation, and 10 for testing per subject. Each 3-second trial was resampled to 200 Hz for consistency. As shown in Table 18, decoding imagined speech is challenging, as reflected

by the fact that several foundation models perform worse than task-specific models like SPaRCNet. In this task, EmBrace achieves the best performance with 0.5867 in ACC-B and 0.4833 in Kappa, outperforming CodeBrain by 2.60 percentage points in Kappa. These results indicate that our integrated approach can better extract relevant patterns from imagined speech signals compared to standard pre-trained models.

*Table 18.* Performance comparison on the BCIC2020-3 (5-Class) dataset.

| Methods | ACC-B | W-F1 | Kappa |
|---|---|---|---|
| EEGNet | $0.2600 \pm 0.0151$ | $0.2573 \pm 0.0163$ | $0.0750 \pm 0.0189$ |
| EEG-Deformer | $0.4563 \pm 0.0113$ | $0.4567 \pm 0.0116$ | $0.3203 \pm 0.0141$ |
| SERA | $0.2456 \pm 0.0198$ | $0.2012 \pm 0.0256$ | $0.0570 \pm 0.0247$ |
| SPaRCNet | $0.5101 \pm 0.0137$ | $0.5098 \pm 0.0137$ | $0.3877 \pm 0.0171$ |
| ContraWR | $0.3000 \pm 0.0298$ | $0.2713 \pm 0.0411$ | $0.1250 \pm 0.0372$ |
| CNN-Transformer | $0.2139 \pm 0.0092$ | $0.1142 \pm 0.0104$ | $0.0173 \pm 0.0115$ |
| FFCL | $0.2248 \pm 0.0060$ | $0.1912 \pm 0.0067$ | $0.0310 \pm 0.0075$ |
| ST-Transformer | $0.3432 \pm 0.0173$ | $0.3431 \pm 0.0170$ | $0.1790 \pm 0.0216$ |
| BIOT | $0.3592 \pm 0.0187$ | $0.3579 \pm 0.0191$ | $0.1990 \pm 0.0234$ |
| LaBraM | $0.2288 \pm 0.0255$ | $0.2139 \pm 0.0347$ | $0.0360 \pm 0.0319$ |
| CBraMod | $0.3120 \pm 0.0155$ | $0.3112 \pm 0.0158$ | $0.1400 \pm 0.0193$ |
| CodeBrain | $\underline{0.5659} \pm 0.0110$ | $\underline{0.5659} \pm 0.0115$ | $\underline{0.4573} \pm 0.0138$ |
| EmBrace | $\mathbf{0.5867} \pm 0.0115$ | $\mathbf{0.5869} \pm 0.0115$ | $\mathbf{0.4833} \pm 0.0144$ |

## H.6. Mental Disorder Diagnosis

**Mumtaz2016** comprises a pathological EEG repository for mental health screening, involving 34 patients with major depressive disorder (MDD) and 30 healthy controls (Mumtaz, 2016). Data acquisition employed 19 electrodes arranged per the 10-20 international system at a 256 Hz sample rate. Adhering to the setup in (Wang et al., 2025), we focused on the eyes-open and eyes-closed states, with signals resampled to 200 Hz and divided into 7,143 5-second segments. For subject-independent assessment, we allocated 43 participants for training, 9 for validation, and 10 for testing. As shown in Table 19, EmBrace reaches 0.9048 in ACC-B, 0.9811 in AUC-PR, and 0.9798 in AUROC.

*Table 19.* Performance comparison on the Mumtaz2016 (2-Class) dataset.

| Methods | ACC-B | AUC-PR | AUROC |
|---|---|---|---|
| EEGNet | $0.8867 \pm 0.0542$ | $0.9586 \pm 0.0219$ | $0.9505 \pm 0.0211$ |
| EEG-Deformer | $0.9097 \pm 0.0314$ | $0.9768 \pm 0.0023$ | $0.9705 \pm 0.0030$ |
| SERA | $0.9011 \pm 0.0134$ | $0.9685 \pm 0.0071$ | $0.9590 \pm 0.0100$ |
| SPaRCNet | $0.8921 \pm 0.0177$ | $0.9705 \pm 0.0087$ | $0.9632 \pm 0.0130$ |
| ContraWR | $0.8984 \pm 0.0073$ | $0.9670 \pm 0.0123$ | $0.9611 \pm 0.0164$ |
| CNN-Transformer | $0.9038 \pm 0.0309$ | $0.9631 \pm 0.0286$ | $0.9438 \pm 0.0537$ |
| FFCL | $0.8973 \pm 0.0047$ | $0.9705 \pm 0.0076$ | $0.9662 \pm 0.0122$ |
| ST-Transformer | $0.8790 \pm 0.0206$ | $0.9838 \pm 0.0086$ | $0.9817 \pm 0.0101$ |
| BIOT | $0.8866 \pm 0.0158$ | $0.9763 \pm 0.0149$ | $0.9737 \pm 0.0171$ |
| LaBraM | $\underline{0.9053} \pm 0.0069$ | $0.9767 \pm 0.0043$ | $0.9728 \pm 0.0071$ |
| CBraMod | $0.8867 \pm 0.0036$ | $0.9784 \pm 0.0050$ | $\underline{0.9791} \pm 0.0056$ |
| CodeBrain | $\mathbf{0.9054} \pm 0.0068$ | $\underline{0.9806} \pm 0.0018$ | $0.9786 \pm 0.0023$ |
| EmBrace | $0.9048 \pm 0.0039$ | $\mathbf{0.9811} \pm 0.0040$ | $\mathbf{0.9798} \pm 0.0049$ |

## H.7. Mental Stress Detection

**MentalArithmetic** is a public EEG dataset used to study cognitive workload and mental stress (Goldberger et al., 2000; Zyma et al., 2019). It contains 20-channel recordings from 36 healthy subjects during both resting states and active mental subtraction tasks. Following the preprocessing in (Wang et al., 2025), we resampled the 500 Hz signals to 200 Hz and applied a 0.5–45 Hz band-pass filter, resulting in 1,707 5-second segments. We employed a subject-independent evaluation protocol, allocating subjects 1–28 for training, 29–32 for validation, and 33–36 for testing. As detailed in Table 20, EmBrace attains 0.7292 in ACC-B, 0.7377 in AUC-PR, and 0.8813 in AUROC. These metrics represent the highest values across all tested models, including foundation-based and specialized architectures, particularly showing an improvement in AUC-PR compared to CodeBrain.

*Table 20.* Performance comparison on the MentalArithmetic (2-Class) dataset.

| Methods | ACC-B | AUC-PR | AUROC |
|---|---|---|---|
| EEGNet | $0.5847 \pm 0.0619$ | $0.5770 \pm 0.1300$ | $0.7875 \pm 0.0633$ |
| EEG-Deformer | $0.5090 \pm 0.0129$ | $0.4899 \pm 0.0323$ | $0.7297 \pm 0.0231$ |
| SERA | $0.5639 \pm 0.0626$ | $0.5766 \pm 0.0549$ | $0.7167 \pm 0.0399$ |
| SPaRCNet | $0.6326 \pm 0.0553$ | $0.4637 \pm 0.0558$ | $0.6963 \pm 0.0645$ |
| ContraWR | $0.5778 \pm 0.0494$ | $0.4782 \pm 0.1231$ | $0.6821 \pm 0.0301$ |
| CNN-Transformer | $0.6229 \pm 0.0835$ | $0.5367 \pm 0.1035$ | $0.7277 \pm 0.0366$ |
| FFCL | $0.5708 \pm 0.0456$ | $0.4481 \pm 0.0295$ | $0.7076 \pm 0.0146$ |
| ST-Transformer | $0.5042 \pm 0.0213$ | $0.3101 \pm 0.0478$ | $0.5881 \pm 0.0284$ |
| BIOT | $0.6229 \pm 0.0743$ | $0.5428 \pm 0.1534$ | $0.7717 \pm 0.0744$ |
| LaBraM | $0.6840 \pm 0.0673$ | $0.5619 \pm 0.1650$ | $0.7073 \pm 0.1132$ |
| CBraMod | $0.7063 \pm 0.0613$ | $0.6109 \pm 0.0686$ | $0.7979 \pm 0.0500$ |
| CodeBrain | $\underline{0.7174 \pm 0.0144}$ | $\underline{0.6417 \pm 0.0507}$ | $\underline{0.8487 \pm 0.0190}$ |
| EmBrace | $\mathbf{0.7292 \pm 0.0391}$ | $\mathbf{0.7377 \pm 0.0415}$ | $\mathbf{0.8813 \pm 0.0201}$ |

## H.8. Mental Attention Detection

**ATTENTION** is an EEG dataset derived from a simultaneous EEG-NIRS study focusing on the Discrimination/Selection Response task (Shin et al., 2018). Each of the 26 subjects performed three sessions to distinguish between target and non-target stimuli, which appeared with probabilities of 30% and 70% respectively. Following the electrode selection in (Ding et al., 2024), we utilized 28 EEG channels by excluding the non-cephalic HEOG and VEOG signals. The original 1000 Hz recordings were downsampled to 200 Hz and processed with a 0.5–50 Hz band-pass filter and ICA-based artifact removal. To balance the classes, we extracted the initial 20 seconds of each task trial and segmented the data into 4-second windows with 50% overlap. Subjects 1–16, 17–21, and 22–26 were used for training, validation, and testing respectively. As reported in Table 21, EmBrace attains 0.7217 in ACC-B, 0.7884 in AUC-PR, and 0.8031 in AUROC.

*Table 21.* Performance comparison on the ATTENTION (2-Class) dataset.

| Methods | ACC-B | AUC-PR | AUROC |
|---|---|---|---|
| EEGNet | $0.7344 \pm 0.0369$ | $0.8046 \pm 0.0367$ | $0.8271 \pm 0.0386$ |
| EEG-Deformer | $0.5664 \pm 0.0457$ | $0.8021 \pm 0.0080$ | $0.8138 \pm 0.0091$ |
| SERA | $0.6474 \pm 0.0197$ | $0.6815 \pm 0.0245$ | $0.7150 \pm 0.0288$ |
| SPaRCNet | $0.7467 \pm 0.0292$ | $0.8470 \pm 0.0305$ | $0.8262 \pm 0.0328$ |
| ContraWR | $0.7012 \pm 0.0302$ | $0.8140 \pm 0.0109$ | $0.8155 \pm 0.0132$ |
| CNN-Transformer | $0.5746 \pm 0.0613$ | $0.6667 \pm 0.0190$ | $0.6757 \pm 0.0328$ |
| FFCL | $0.7256 \pm 0.0263$ | $0.8239 \pm 0.0183$ | $0.8264 \pm 0.0172$ |
| ST-Transformer | $0.7462 \pm 0.0126$ | $0.8358 \pm 0.0099$ | $0.8280 \pm 0.0079$ |
| BIOT | $0.5784 \pm 0.0402$ | $0.6839 \pm 0.0542$ | $0.7081 \pm 0.0310$ |
| LaBraM | $0.6170 \pm 0.0287$ | $0.6610 \pm 0.0176$ | $0.6789 \pm 0.0255$ |
| CBraMod | $\underline{0.7033 \pm 0.0139}$ | $\underline{0.7803 \pm 0.0054}$ | $\underline{0.7915 \pm 0.0045}$ |
| CodeBrain | $0.6704 \pm 0.0082$ | $0.7363 \pm 0.0070$ | $0.7503 \pm 0.0055$ |
| EmBrace | $\mathbf{0.7217 \pm 0.0036}$ | $\mathbf{0.7884 \pm 0.0129}$ | $\mathbf{0.8031 \pm 0.0104}$ |

## I. Sensitivity Analysis of Loss Weights

**Experimental Settings.** To evaluate the contribution of each loss term, we refine the optimization objective $\mathcal{J}$ defined in Eq. (17) as:

$$\mathcal{J} = \lambda \mathcal{L}_{\text{task}} + (1 - \lambda)\mathcal{L}_{\text{fusion}}$$
$$\mathcal{L}_{\text{fusion}} = \gamma \mathcal{L}_{\text{Attn}} + (2 - \gamma)\mathcal{L}_{\text{Emb}} \tag{31}$$

where $\mathcal{L}_{\text{Attn}}$ and $\mathcal{L}_{\text{Emb}}$ denote the discrepancies $\mathcal{L}(\tilde{\mathbf{K}}_k^{(\mathbf{T})}, \mathbf{K}_k^{(*)})$ for the respective scales. Here, $\lambda$ governs the **inter-loss** balance between the ground truth supervision ($\mathcal{L}_{\text{task}}$) and the knowledge fusion loss ($\mathcal{L}_{\text{fusion}}$), while $\gamma \in [0, 2]$ controls the **intra-loss** allocation between the two types of knowledge $\mathcal{K}^{(*)} = \{\mathbf{K}_{\text{Attn}}^{(*)}, \mathbf{K}_{\text{Emb}}^{(*)}\}$. In this formulation, the combined weight of the fusion terms is $2(1 - \lambda)$. This design provides a stronger gradient signal from the source models than a standard normalized sum. We use $\lambda = 0.6$ and $\gamma = 1.0$ as the default hyperparameters, resulting in a weight distribution of

$(0.6, 0.4, 0.4)$ for $\mathcal{L}_{\text{task}}$, $\mathcal{L}_{\text{Attn}}$, and $\mathcal{L}_{\text{Emb}}$, respectively.

The sensitivity analysis is conducted in two stages:

- **Phase I: Inter-loss Balancing ($\lambda$-Search).** This stage evaluates the trade-off between the ground truth supervision and the knowledge fusion of $\mathcal{K}^{(*)}$. We vary $\lambda \in \{0.5, 0.6, 0.7, 0.8, 0.9\}$ while keeping $\gamma = 1.0$. This determines the relative importance of the fused expertise compared to the task-specific labels across different EEG datasets.

- **Phase II: Intra-loss Allocation ($\gamma$-Search).** We vary $\gamma \in \{0.0, 0.5, 1.0, 1.5, 2.0\}$ while fixing $\lambda = 0.6$ to analyze the individual impact of $\mathbf{K}_{\text{Attn}}^{(*)}$ and $\mathbf{K}_{\text{Emb}}^{(*)}$. This stage examines whether structural attention knowledge or localized embedding knowledge is more effective for the target architecture $\mathbf{T}$.

The comprehensive results of this two-stage analysis are summarized in Table 22, with the corresponding performance trends visualized in Figure 5.

*Table 22.* Two-stage sensitivity analysis of loss weights across different EEG datasets.

| Stage | Param. | BCIC-IV-2A (*4-class*) | | | ATTENTION (*2-class*) | | |
|---|---|---|---|---|---|---|---|
| | | ACC-B | W-F1 | Kappa | ACC-B | AUC-PR | AUROC |
| **Stage I:** Inter-loss | $\lambda = 0.9$ | $0.4715 \pm 0.0307$ | $0.4354 \pm 0.0538$ | $0.2954 \pm 0.0409$ | $0.7111 \pm 0.0100$ | $0.7821 \pm 0.0047$ | $0.7934 \pm 0.0080$ |
| | $\lambda = 0.8$ | $0.4799 \pm 0.0434$ | $0.4464 \pm 0.0681$ | $0.3065 \pm 0.0578$ | $0.7181 \pm 0.0129$ | $0.7874 \pm 0.0112$ | $0.8007 \pm 0.0148$ |
| | $\lambda = 0.7$ | $0.4896 \pm 0.0376$ | $\underline{0.4667} \pm 0.0531$ | $0.3194 \pm 0.0501$ | $\mathbf{0.7241} \pm 0.0166$ | $\mathbf{0.7916} \pm 0.0135$ | $\mathbf{0.8059} \pm 0.0130$ |
| | $\lambda = 0.6$ | $\mathbf{0.5047} \pm 0.0203$ | $\mathbf{0.4870} \pm 0.0240$ | $\mathbf{0.3396} \pm 0.0271$ | $\underline{0.7217} \pm 0.0036$ | $\underline{0.7884} \pm 0.0129$ | $\underline{0.8031} \pm 0.0104$ |
| | $\lambda = 0.5$ | $\underline{0.4915} \pm 0.0408$ | $0.4610 \pm 0.0610$ | $\underline{0.3220} \pm 0.0544$ | $0.7111 \pm 0.0088$ | $0.7853 \pm 0.0071$ | $0.7980 \pm 0.0064$ |
| **Stage II:** Intra-loss | $\gamma = 2.0$ | $0.4806 \pm 0.0312$ | $0.4526 \pm 0.0549$ | $0.3074 \pm 0.0416$ | $\underline{0.7211} \pm 0.0102$ | $\mathbf{0.7953} \pm 0.0053$ | $\mathbf{0.8060} \pm 0.0083$ |
| | $\gamma = 1.5$ | $0.4911 \pm 0.0116$ | $\underline{0.4740} \pm 0.0164$ | $0.3215 \pm 0.0155$ | $0.7183 \pm 0.0148$ | $\underline{0.7920} \pm 0.0071$ | $\underline{0.8035} \pm 0.0134$ |
| | $\gamma = 1.0$ | $\mathbf{0.5047} \pm 0.0203$ | $\mathbf{0.4870} \pm 0.0240$ | $\mathbf{0.3396} \pm 0.0271$ | $\mathbf{0.7217} \pm 0.0036$ | $0.7884 \pm 0.0129$ | $0.8031 \pm 0.0104$ |
| | $\gamma = 0.5$ | $0.4911 \pm 0.0404$ | $0.4605 \pm 0.0606$ | $0.3215 \pm 0.0538$ | $0.7149 \pm 0.0104$ | $0.7783 \pm 0.0077$ | $0.7948 \pm 0.0075$ |
| | $\gamma = 0.0$ | $\underline{0.4976} \pm 0.0446$ | $0.4723 \pm 0.0616$ | $\underline{0.3301} \pm 0.0595$ | $0.7144 \pm 0.0178$ | $0.7893 \pm 0.0135$ | $0.7973 \pm 0.0057$ |

| Stage | Param. | ISRUC-S3 (*5-class*) | | | BCIC2020-3 (*5-class*) | | |
|---|---|---|---|---|---|---|---|
| | | ACC-B | W-F1 | Kappa | ACC-B | W-F1 | Kappa |
| **Stage I:** Inter-loss | $\lambda = 0.9$ | $0.7889 \pm 0.0196$ | $0.7682 \pm 0.0196$ | $0.7085 \pm 0.0244$ | $0.5731 \pm 0.0108$ | $0.5730 \pm 0.0107$ | $0.4663 \pm 0.0135$ |
| | $\lambda = 0.8$ | $0.7887 \pm 0.0195$ | $0.7679 \pm 0.0197$ | $0.7081 \pm 0.0245$ | $\underline{0.5768} \pm 0.0067$ | $0.5768 \pm 0.0064$ | $\underline{0.4710} \pm 0.0084$ |
| | $\lambda = 0.7$ | $0.7895 \pm 0.0187$ | $\underline{0.7690} \pm 0.0187$ | $\underline{0.7094} \pm 0.0231$ | $\underline{0.5768} \pm 0.0049$ | $\underline{0.5770} \pm 0.0047$ | $\underline{0.4710} \pm 0.0061$ |
| | $\lambda = 0.6$ | $\underline{0.7897} \pm 0.0186$ | $\mathbf{0.7692} \pm 0.0188$ | $\mathbf{0.7098} \pm 0.0233$ | $\mathbf{0.5867} \pm 0.0115$ | $\mathbf{0.5869} \pm 0.0115$ | $\mathbf{0.4833} \pm 0.0144$ |
| | $\lambda = 0.5$ | $\mathbf{0.7898} \pm 0.0198$ | $\underline{0.7690} \pm 0.0200$ | $\mathbf{0.7098} \pm 0.0250$ | $0.5715 \pm 0.0178$ | $0.5716 \pm 0.0175$ | $0.4643 \pm 0.0222$ |
| **Stage II:** Intra-loss | $\gamma = 2.0$ | $0.7889 \pm 0.0197$ | $0.7682 \pm 0.0197$ | $0.7085 \pm 0.0244$ | $0.5667 \pm 0.0142$ | $0.5667 \pm 0.0144$ | $0.4583 \pm 0.0178$ |
| | $\gamma = 1.5$ | $0.7895 \pm 0.0187$ | $\underline{0.7690} \pm 0.0187$ | $\underline{0.7094} \pm 0.0231$ | $0.5741 \pm 0.0140$ | $0.5742 \pm 0.0142$ | $0.4677 \pm 0.0175$ |
| | $\gamma = 1.0$ | $\mathbf{0.7897} \pm 0.0186$ | $\mathbf{0.7692} \pm 0.0188$ | $\mathbf{0.7098} \pm 0.0233$ | $\mathbf{0.5867} \pm 0.0115$ | $\mathbf{0.5869} \pm 0.0115$ | $\mathbf{0.4833} \pm 0.0144$ |
| | $\gamma = 0.5$ | $0.7894 \pm 0.0198$ | $0.7685 \pm 0.0199$ | $0.7091 \pm 0.0249$ | $\underline{0.5747} \pm 0.0119$ | $\underline{0.5748} \pm 0.0117$ | $\underline{0.4683} \pm 0.0149$ |
| | $\gamma = 0.0$ | $\underline{0.7896} \pm 0.0206$ | $0.7688 \pm 0.0203$ | $\underline{0.7094} \pm 0.0255$ | $0.5656 \pm 0.0065$ | $0.5659 \pm 0.0065$ | $0.4570 \pm 0.0081$ |

**Note:** Shaded rows highlight the default parameters ($\lambda = 0.6, \gamma = 1.0$).

**Results Analysis. 1) Inter-loss Balancing.** Variations in $\lambda$ show how the target model coordinates task labels with source model signals. A high $\lambda$ of 0.9 causes a performance drop across all datasets, confirming that model behavior is sensitive to reduced fusion guidance. In BCIC-IV-2A and BCIC2020-3, peaks concentrate at $\lambda = 0.6$, where the gradient signal from source models is stronger. However, the ATTENTION dataset peak shifts to $\lambda = 0.7$, suggesting this paradigm requires more task-specific supervision to anchor the transferred representations. **2) Intra-loss Allocation.** The $\gamma$ settings highlight the relative influence of low-level and high-level knowledge. BCIC-IV-2A and BCIC2020-3 reach maximum metrics at $\gamma = 1.0$, where low-level embeddings and high-level attention maps contribute equally. In contrast, the ATTENTION dataset shows a preference for high-level attention maps ($\gamma = 2.0$), yielding the highest AUROC and AUC-PR. This indicates that capturing long-range token dependencies is more influential for this paradigm than inheriting low-level patch embeddings. The stability of default parameters across four heterogeneous datasets validates the capacity of EmBrace to maintain consistent behavior without exhaustive per-dataset tuning.

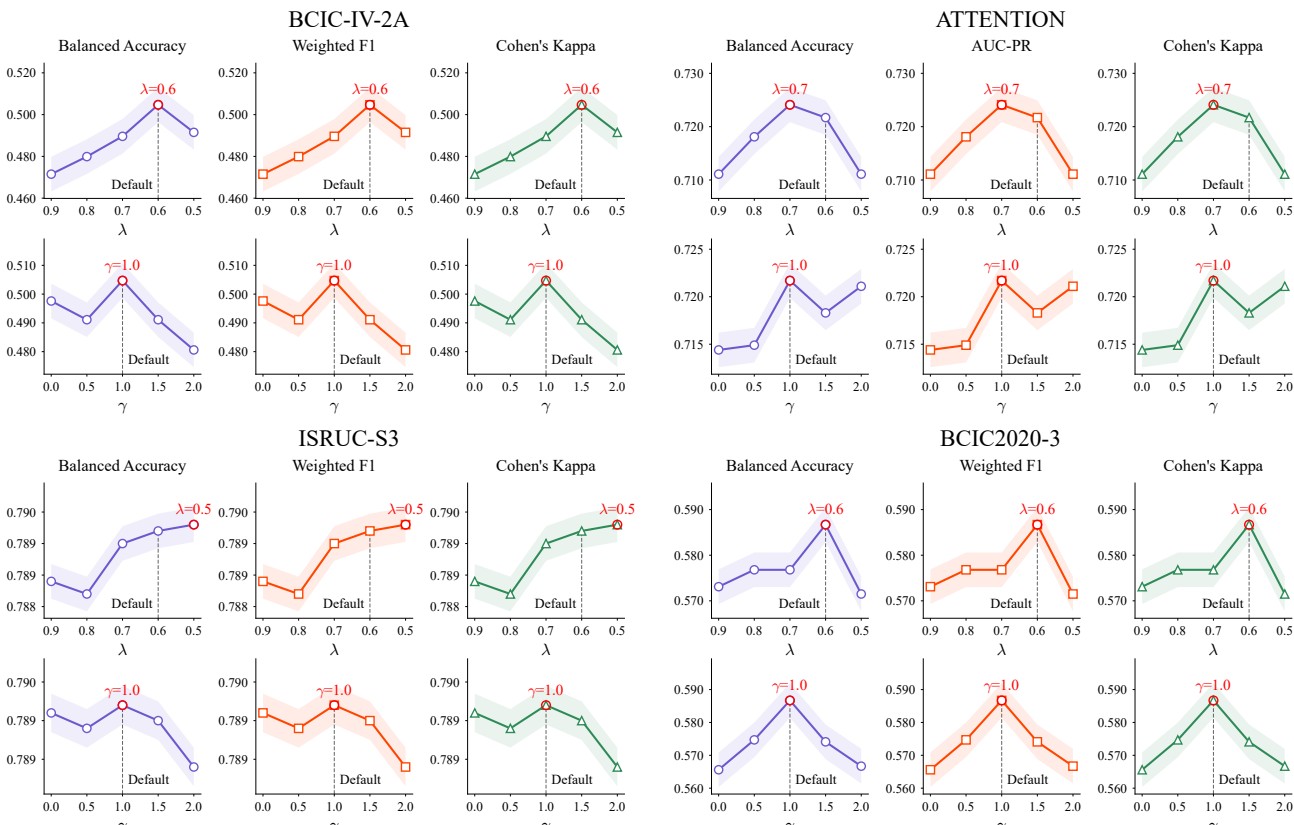

*Figure 5.* Visualizing the sensitivity analysis results across four EEG datasets. The top row of each dataset block shows the **Phase 1** (λ**-Search**) results, reflecting the balance between CE loss and Fusion loss. The bottom row shows the **Phase 2 (γ-Search)** results, reflecting the allocation between Attention and Embedding knowledge. Red circles mark the peak performance points, and the dashed vertical lines indicate the default hyperparameter settings ($\lambda = 0.6, \gamma = 1.0$).

## J. Ablation Study

### J.1. Ablation on Design Choices

**Ablation Settings.** The knowledge fusion process uses meta-guided weighting and a **bridge** module to handle brain state changes across different samples. Here, we evaluate the design of the *Fusor* $\mathcal{G}$ and *bridge* modules to see how they contribute to the fusion performance. The results are summarized in Table 23.

**Configurations of the *Fusor* Module.** The *Fusor* $\mathcal{G}$ assigns weights $\mathbf{w}$ to different source models based on the physiological state of each EEG sample $\mathbf{X}_{\text{raw}}$. We examine this through four incremental variants: **1) Static Weighting**: We fix the weights $w^{(m)}$ to $1/M$, assuming every source model contributes equally regardless of the input. **2) Learnable Weighting**: Weights are optimized as global parameters but remain invariant across different samples, failing to respond to physiological variability. **3) Decoupled Dynamic Weighting**: This variant generates independent weights $\mathbf{w}_{\text{Emb}}$ and $\mathbf{w}_{\text{Attn}}$, allowing the model to separately adjust the information flow of embedding and attention knowledge. **4) Shared Dynamic Weighting (Ours)**: Our design uses a meta-guided process $\mathcal{G}$ to map physiological features $\mathbf{X}_{\text{meta}}$ into shared weights where $\mathbf{w}_{\text{Emb}} = \mathbf{w}_{\text{Attn}} = \mathbf{w}$, testing the efficacy of using a single biological context to calibrate all scales of source knowledge in $\mathcal{K}^{(*)}$ simultaneously.

**Configurations of the *Bridge* Module.** The *bridge* projects *embedding knowledge* $\mathbf{K}_{\text{Emb}}^{(m)}$ into a shared latent space $\mathcal{V}_{\text{Emb}}$. We compare different implementations: **5) W/O Bridge**: We use raw source embeddings directly to test if explicit manifold unification $\mathcal{V}_{\text{Emb}}$ is necessary. **6) Shared Linear Bridge**: All source models utilize a shared single-layer linear projection to map dimensions between source models and the carrier $\mathbf{T}$. **7) Independent MLP Bridge**: Each source model is assigned a separate 2-layer MLP, allowing for source-specific mapping at the expense of higher complexity. **8) Shared MLP Bridge (Ours)**: A shared 2-layer MLP is applied across all source models as described in Eq. (10) to promote a universal latent representation and reduce reliance on specific source domain patterns.

*Table 23.* Ablation study of component designs across different EEG datasets.

| Comp. | Variant | BCIC-IV-2A (*4-class*) | | | ATTENTION (*2-class*) | | |
|---|---|---|---|---|---|---|---|
| | | ACC-B | W-F1 | Kappa | ACC-B | AUC-PR | AUROC |
| **Fusor** | Static | 0.4865 ± 0.0450 | 0.4631 ± 0.0589 | 0.3153 ± 0.0600 | 0.7078 ± 0.0110 | 0.7856 ± 0.0065 | 0.7959 ± 0.0048 |
| | Learnable | 0.4865 ± 0.0450 | 0.4631 ± 0.0589 | 0.3153 ± 0.0600 | 0.7070 ± 0.0077 | 0.7856 ± 0.0062 | 0.7960 ± 0.0044 |
| | Decoupled | 0.4656 ± 0.0222 | 0.4316 ± 0.0327 | 0.2875 ± 0.0296 | 0.7163 ± 0.0090 | 0.7839 ± 0.0082 | 0.8013 ± 0.0080 |
| | Dynamic | **0.5047 ± 0.0203** | **0.4870 ± 0.0240** | **0.3396 ± 0.0271** | **0.7217 ± 0.0036** | **0.7884 ± 0.0129** | **0.8031 ± 0.0104** |
| **Bridge** | W/O | 0.4691 ± 0.0215 | 0.4364 ± 0.0304 | 0.2921 ± 0.0287 | 0.6637 ± 0.0952 | 0.7413 ± 0.0829 | 0.7541 ± 0.0856 |
| | Linear | 0.4613 ± 0.0281 | 0.4238 ± 0.0412 | 0.2817 ± 0.0375 | 0.7089 ± 0.0148 | 0.7852 ± 0.0129 | 0.7987 ± 0.0123 |
| | Indep. | 0.4526 ± 0.0230 | 0.4158 ± 0.0316 | 0.2701 ± 0.0306 | 0.7120 ± 0.0169 | 0.7816 ± 0.0121 | 0.7952 ± 0.0132 |
| | Shared | **0.5047 ± 0.0203** | **0.4870 ± 0.0240** | **0.3396 ± 0.0271** | **0.7217 ± 0.0036** | **0.7884 ± 0.0129** | **0.8031 ± 0.0104** |

| Comp. | Variant | ISRUC-S3 (*5-class*) | | | BCIC2020-3 (*5-class*) | | |
|---|---|---|---|---|---|---|---|
| | | ACC-B | W-F1 | Kappa | ACC-B | W-F1 | Kappa |
| **Fusor** | Static | 0.7660 ± 0.0268 | 0.7344 ± 0.0485 | 0.6732 ± 0.0502 | 0.5741 ± 0.0060 | 0.5740 ± 0.0058 | 0.4677 ± 0.0075 |
| | Learnable | 0.7656 ± 0.0275 | 0.7337 ± 0.0497 | 0.6726 ± 0.0514 | 0.5677 ± 0.0188 | 0.5678 ± 0.0185 | 0.4597 ± 0.0235 |
| | Decoupled | 0.7655 ± 0.0188 | 0.7403 ± 0.0268 | 0.6693 ± 0.0189 | 0.5779 ± 0.0110 | 0.5779 ± 0.0111 | 0.4723 ± 0.0138 |
| | Dynamic | **0.7897 ± 0.0186** | **0.7692 ± 0.0188** | **0.7098 ± 0.0233** | **0.5867 ± 0.0115** | **0.5869 ± 0.0115** | **0.4833 ± 0.0144** |
| **Bridge** | W/O | 0.7859 ± 0.0174 | 0.7595 ± 0.0214 | 0.7019 ± 0.0253 | 0.5667 ± 0.0111 | 0.5672 ± 0.0114 | 0.4583 ± 0.0139 |
| | Linear | 0.7861 ± 0.0172 | 0.7594 ± 0.0208 | 0.7020 ± 0.0249 | 0.5773 ± 0.0128 | 0.5777 ± 0.0127 | 0.4717 ± 0.0159 |
| | Indep. | 0.7583 ± 0.0306 | 0.7351 ± 0.0190 | 0.6679 ± 0.0282 | 0.5755 ± 0.0064 | 0.5754 ± 0.0065 | 0.4693 ± 0.0080 |
| | Shared | **0.7897 ± 0.0186** | **0.7692 ± 0.0188** | **0.7098 ± 0.0233** | **0.5867 ± 0.0115** | **0.5869 ± 0.0115** | **0.4833 ± 0.0144** |

**Note:** Shaded rows highlight our proposed designs (Dynamic for Fusor and Unified for Bridge).

## J.2. Ablation on Temperature Scaling

**Ablation Settings.** This section investigates the sensitivity of the temperature scaling factor $\tau$ introduced in Eq. (13). By adjusting $\tau \in \{1, 2, 4, 8, 16, 32\}$, we control the smoothness of the dynamic weight distribution $\mathbf{w}$ assigned to diverse foundation models during the adaptive knowledge fusion process. The comprehensive results of this analysis are summarized in Table 24 and visualized in Figure 6.

*Table 24.* Sensitivity analysis of the temperature scaling factor $\tau$ across different EEG datasets.

| Param. | BCIC-IV-2A (*4-class*) | | | ATTENTION (*2-class*) | | |
|---|---|---|---|---|---|---|
| | ACC-B | W-F1 | Kappa | ACC-B | AUC-PR | AUROC |
| $\tau = 1$ | 0.4946 ± 0.0437 | 0.4669 ± 0.0645 | 0.3262 ± 0.0583 | 0.7156 ± 0.0273 | 0.7854 ± 0.0203 | 0.7989 ± 0.0197 |
| $\tau = 2$ | 0.4929 ± 0.0408 | 0.4656 ± 0.0619 | 0.3238 ± 0.0544 | 0.7157 ± 0.0274 | 0.7854 ± 0.0203 | 0.7989 ± 0.0197 |
| $\tau = 4$ | 0.4934 ± 0.0434 | 0.4663 ± 0.0645 | 0.3245 ± 0.0578 | 0.7163 ± 0.0276 | 0.7853 ± 0.0203 | 0.7989 ± 0.0197 |
| $\tau = 8$ | **0.5047 ± 0.0203** | **0.4870 ± 0.0240** | **0.3396 ± 0.0271** | **0.7217 ± 0.0036** | 0.7884 ± 0.0129 | 0.8031 ± 0.0104 |
| $\tau = 16$ | 0.5003 ± 0.0436 | 0.4712 ± 0.0653 | 0.3338 ± 0.0582 | 0.7206 ± 0.0160 | **0.7900 ± 0.0129** | **0.8041 ± 0.0078** |
| $\tau = 32$ | 0.4908 ± 0.0339 | 0.4616 ± 0.0552 | 0.3211 ± 0.0451 | 0.7211 ± 0.0164 | 0.7872 ± 0.0145 | 0.8004 ± 0.0149 |

| Param. | ISRUC-S3 (*5-class*) | | | BCIC2020-3 (*5-class*) | | |
|---|---|---|---|---|---|---|
| | ACC-B | W-F1 | Kappa | ACC-B | W-F1 | Kappa |
| $\tau = 1$ | 0.7894 ± 0.0194 | 0.7690 ± 0.0193 | 0.7094 ± 0.0241 | 0.5784 ± 0.0123 | 0.5789 ± 0.0120 | 0.4730 ± 0.0154 |
| $\tau = 2$ | 0.7896 ± 0.0189 | 0.7693 ± 0.0189 | 0.7098 ± 0.0235 | 0.5731 ± 0.0134 | 0.5732 ± 0.0132 | 0.4663 ± 0.0167 |
| $\tau = 4$ | 0.7898 ± 0.0186 | 0.7695 ± 0.0186 | 0.7101 ± 0.0230 | 0.5765 ± 0.0094 | 0.5765 ± 0.0092 | 0.4707 ± 0.0117 |
| $\tau = 8$ | 0.7897 ± 0.0186 | 0.7692 ± 0.0188 | 0.7098 ± 0.0233 | 0.5867 ± 0.0115 | 0.5869 ± 0.0115 | 0.4833 ± 0.0144 |
| $\tau = 16$ | **0.7901 ± 0.0182** | **0.7698 ± 0.0183** | **0.7104 ± 0.0226** | 0.5779 ± 0.0096 | 0.5780 ± 0.0095 | 0.4723 ± 0.0119 |
| $\tau = 32$ | **0.7901 ± 0.0182** | **0.7698 ± 0.0183** | **0.7104 ± 0.0226** | **0.5952 ± 0.0136** | **0.5955 ± 0.0131** | **0.4940 ± 0.0170** |

**Note:** Shaded row highlights the default temperature ($\tau = 8$).

**Results Analysis.** The scaling factor $\tau$ modulates the contribution of each source model by adjusting the entropy of the fusion process: **1) Impact of Distribution Sparsity.** A low temperature ($\tau = 1$) results in a near-one-hot weight assignment,

forcing the target architecture to derive knowledge from a single dominant source. The suboptimal performance in this range indicates that localized selection fails to utilize the complementary spatial-temporal priors available in the model pool. Increasing $\tau$ to 8 or 16 improves metrics by facilitating multi-scale feature aggregation, allowing the framework to synthesize representations from multiple foundation models simultaneously. **2) Dynamic Adaptation vs. Static Configuration.** Although higher temperatures like $\tau = 32$ produce smoother and more uniform coefficients, the performance consistently exceeds the Static Weighting variant described in Appendix J.1. This gap confirms that the meta-guided mechanism successfully encodes sample-level biological context; even when numerical fluctuations become subtle, they remain synchronized with underlying brain state transitions. This trend is particularly evident in complex paradigms like ISRUC-S3 and BCIC2020-3, where maintaining a broad integration of diverse knowledge provides greater robustness than focusing on a single model's output. The stability across this range validates that EmBrace effectively balances representational diversity with the dynamic requirements of downstream EEG tasks.

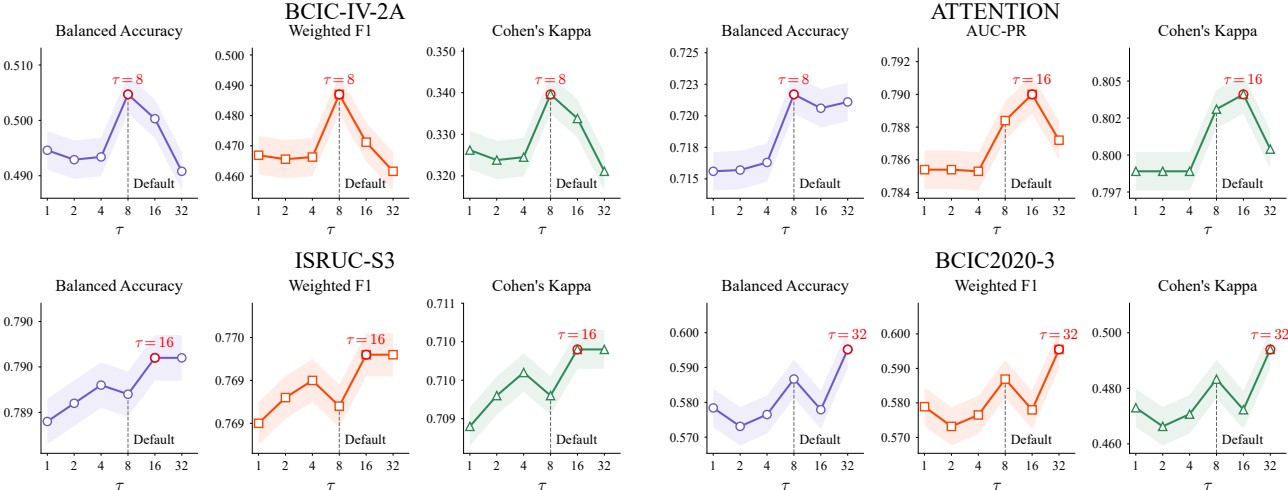

*Figure 6.* Sensitivity analysis of evaluation metrics across four datasets with respect to the temperature scaling factor $\tau$. Red open circles denote the best performance value achieved for each respective metric.

### J.3. Ablation on Model Selection Methods

**Ablation Settings.** This ablation study is conducted to justify the choice of our primary selection method and to verify its effectiveness in identifying the most compatible source models for diverse EEG target tasks. To rigorously assess the reliability of this selection mechanism, we compare the adopted method against two widely recognized compatibility assessment methods, namely Negative Conditional Entropy (NCE) and Log Maximum Evidence (LogME), which serve as established baselines for evaluating the structural congruence between pre-trained feature spaces and target label distributions:

- **Pairwise Annotation Representation Comparison (PARC)**: This method assesses the structural congruence between the feature space and label space (Bolya et al., 2021). For a feature matrix $\mathbf{F} \in \mathbb{R}^{B \times d}$ and one-hot label matrix $\mathbf{Y} \in \{0,1\}^{B \times C}$, the relationship is captured by two distance matrices:

$$\begin{cases} D_{\mathbf{F}} = 1 - \text{corrcoef}(\mathbf{F}) \in \mathbb{R}^{B \times B} \\ D_{\mathbf{Y}} = 1 - \text{corrcoef}(\mathbf{Y}) \in \mathbb{R}^{B \times B} \end{cases} \tag{32}$$

The score is defined as the Spearman rank correlation between the flattened lower-triangular parts of these matrices:

$$\text{PARC} = \rho_{\text{Spearman}}(\text{vec}(\text{lt}(D_{\mathbf{F}})), \text{vec}(\text{lt}(D_{\mathbf{Y}}))) \tag{33}$$

Higher scores indicate that the model's representation naturally clusters samples in a manner that mirrors the target categories, consistent with the compatibility score $\xi^{(m)}$ defined in Section 3.3.

- **Log Maximum Evidence (LogME)**: This method measures **structural alignment** by estimating the maximum log-marginal likelihood of a linear head (You et al., 2021). It models the relationship as $\mathbf{y} = \mathbf{F}w + \epsilon$, assuming a Bayesian

framework where weights $w$ follow a Gaussian prior $\mathcal{N}(0, \alpha^{-1}I)$. The evidence is obtained by marginalizing over the weight distribution:

$$p(\mathbf{y}|\mathbf{F}, \alpha, \beta) = \int p(\mathbf{y}|\mathbf{F}, w, \beta)p(w|\alpha)dw \tag{34}$$

The final score balances the model's fitting ability with its complexity:

$$\text{LogME} = \frac{1}{N} \sum_{c=1}^{C} \max_{\alpha_c, \beta_c} \left( \frac{d}{2} \ln \alpha_c + \frac{N}{2} \ln \beta_c - \frac{\beta_c}{2} \|\mathbf{y}_c - \mathbf{F}m_c\|^2 - \frac{\alpha_c}{2} \|m_c\|^2 - \frac{1}{2} \ln |A| \right) \tag{35}$$

where $m_c$ is the posterior mean and $A$ is the Hessian matrix.

- **Negative Conditional Entropy (NCE)**: Based on information theory, NCE method quantifies the information that source model predictions $Z$ provide about target labels $Y$ (Tran et al., 2019). It characterizes the uncertainty of $Y$ given $Z$:

$$\text{NCE} = -H(Y|Z) = \sum_{z \in \mathcal{Z}} P(z) \sum_{y \in \mathcal{Y}} P(y|z) \log P(y|z) \tag{36}$$

The joint probability $P(y, z)$ is estimated empirically from the counts of sample pairs $(y_i, z_i)$. A value closer to zero suggests higher relevance.

*Table 25.* Ablation analysis of structural alignment methods across six EEG datasets. The scores evaluate the compatibility between candidate source models and target task distributions.

| Source | BCIC-IV-2A | | | ATTENTION | | | MentalArithmetic | | |
|---|---|---|---|---|---|---|---|---|---|
| | NCE | LogME | PARC | NCE | LogME | PARC | NCE | LogME | PARC |
| LaBraM | -1.3809 | -0.7243 | 0.0081 | -0.6923 | -1.0729 | 0.0011 | -0.5592 | -0.9929 | 0.0523 |
| CBraMod | **-1.3639** | **-0.7206** | 0.0273 | **-0.6921** | -1.0711 | 0.0108 | **-0.5429** | -0.9910 | 0.0077 |
| CodeBrain | -1.3863 | -0.7253 | 0.0121 | -0.6928 | **-1.0673** | 0.0077 | -0.5641 | **-0.9836** | 0.0566 |

| Source | SEED-V | | | FACED | | | ISRUC-S1 | | |
|---|---|---|---|---|---|---|---|---|---|
| | NCE | LogME | PARC | NCE | LogME | PARC | NCE | LogME | PARC |
| LaBraM | -1.5864 | -0.6023 | 0.0037 | -2.1876 | **-0.3174** | 0.0042 | -1.7863 | **-0.4765** | 0.2899 |
| CBraMod | -1.5891 | -0.5977 | 0.0072 | -2.1871 | -0.3192 | 0.0003 | **-1.7841** | -0.5102 | 0.1166 |
| CodeBrain | **-1.5885** | **-0.5812** | 0.0141 | **-2.1856** | -0.3177 | 0.0054 | -1.8115 | -0.4895 | 0.1709 |

**Note:** Shaded cells denote the results of the specific selection method used to identify the optimal carrier for each task. **calculated** values indicate the highest compatibility score predicted by each method within a dataset.

**Results Analysis. 1) Superiority of Structural Resonance:** The experimental results reported in Table 25 demonstrate that the structural alignment method (Bolya et al., 2021) provides a more stable and discriminative assessment of model compatibility, consistently identifying the optimal source models for diverse EEG paradigms, such as selecting CBraMod for BCIC-IV-2A and CodeBrain for MentalArithmetic. **2) Robustness Against Baseline Inconsistencies:** In contrast, while NCE and LogME show competitive results in specific datasets, they exhibit critical inconsistencies; for instance, LogME prioritizes CodeBrain for the ATTENTION task despite CBraMod yielding better downstream alignment, and both methods fail to correctly rank the most effective backbone for ISRUC-S1, whereas our adopted method effectively captures the structural congruence (0.2899) necessary to minimize the conditional uncertainty stipulated in Lemma 3.3.

### J.4. Ablation on Fusion Criteria

**Ablation Settings.** To investigate the impact of different optimization criteria on cross-modal knowledge fusion, we evaluate various combinations of loss functions for the attention maps $\mathcal{L}_{\text{Attn}}$ and the latent embeddings $\mathcal{L}_{\text{Emb}}$. Specifically, we compare the performance of **Mean Squared Error** (MSE), **Mean Absolute Error** (MAE), and **Kullback-Leibler** (KL) Divergence with a temperature scaling factor $T = 4$. These criteria are selected to capture different facets of the neural signal, ranging from raw amplitude patterns to high-level structural dependencies:

- **Mean Squared Error (MSE)**: This criterion enforces a quadratic penalty on discrepancies to ensure a strict element-wise approximation. By prioritizing the reduction of larger deviations, MSE effectively preserves the high-amplitude rhythmic components and precise feature magnitudes essential for reconstructing time-frequency EEG characteristics during knowledge fusion:

$$\mathcal{L}_{MSE} = \frac{1}{N} \sum_{n=1}^{N} (\mathbf{K}_{k,n}^{(*)} - \tilde{\mathbf{K}}_{k,n}^{(\mathbf{T})})^2 \tag{37}$$

- **Mean Absolute Error (MAE)**: As a robust linear constraint, MAE treats all deviations equally, focusing on the global consistency of the fused patterns. This approach provides a complementary objective that prevents isolated outliers or noise-induced spikes from disproportionately influencing the gradient updates during the knowledge internalization process:

$$\mathcal{L}_{MAE} = \frac{1}{N} \sum_{n=1}^{N} |\mathbf{K}_{k,n}^{(*)} - \tilde{\mathbf{K}}_{k,n}^{(\mathbf{T})}| \tag{38}$$

- **Kullback-Leibler (KL) Divergence**: Beyond point-wise approximation, this criterion matches the probabilistic topology of the *fused attention knowledge*. Following the distillation paradigm (Hinton et al., 2015), we utilize a temperature scaling factor $T$ to soften the probability maps. This mechanism reveals latent structural relations within the attention weights that are typically suppressed, guiding the target model to mirror the long-range dependencies and relative importance assigned by the source models:

$$\mathcal{L}_{KL} = T^2 \sum_{x \in \mathbf{X}} P_{\mathbf{K}_k^{(*)}}(x, T) \log \left( \frac{P_{\mathbf{K}_k^{(*)}}(x, T)}{Q_{\tilde{\mathbf{K}}_k^{(\mathbf{T})}}(x, T)} \right) \tag{39}$$

The quantitative results across the BCIC-IV-2A and BCIC2020-3 datasets are summarized in Table 26.

*Table 26.* Ablation analysis of different fusion criteria for knowledge fusion.

| Loss Criteria | | BCIC-IV-2A (4-class) | | | BCIC2020-3 (5-class) | | |
|---|---|---|---|---|---|---|---|
| $\mathcal{L}_{\text{Attn}}$ | $\mathcal{L}_{\text{Emb}}$ | ACC-B | W-F1 | Kappa | ACC-B | W-F1 | Kappa |
| **MSE** | **MSE** | **0.5047** ± 0.0203 | **0.4870** ± 0.0240 | **0.3396** ± 0.0271 | **0.5867** ± 0.0115 | **0.5869** ± 0.0115 | **0.4833** ± 0.0144 |
| MSE | MAE | 0.4880 ± 0.0208 | 0.4649 ± 0.0260 | 0.3174 ± 0.0277 | 0.5811 ± 0.0083 | 0.5813 ± 0.0084 | 0.4763 ± 0.0104 |
| KL$_{T=4}$ | MSE | 0.4899 ± 0.0203 | 0.4726 ± 0.0187 | 0.3199 ± 0.0271 | 0.5685 ± 0.0063 | 0.5685 ± 0.0064 | 0.4607 ± 0.0079 |
| KL$_{T=4}$ | MAE | 0.4924 ± 0.0162 | 0.4716 ± 0.0171 | 0.3231 ± 0.0216 | 0.5733 ± 0.0093 | 0.5735 ± 0.0093 | 0.4667 ± 0.0117 |

**Note:** Shaded row indicates our proposed multi-objective loss configuration. KL$_{T=4}$ denotes Kullback-Leibler divergence with a temperature factor of 4.

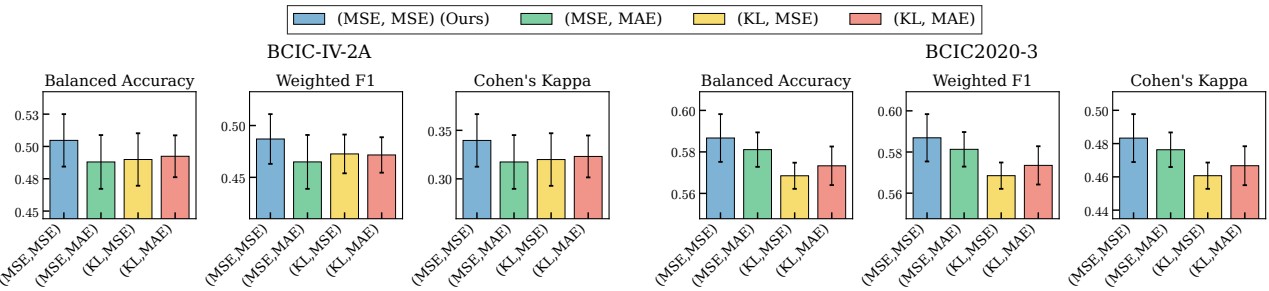

*Figure 7.* Performance comparison of different fusion criteria across BCIC-IV-2A and BCIC2020-3 datasets.

**Results Analysis.** The results reveal that the choice of fusion criteria significantly influences the knowledge transfer process.
**1) Superiority of Dual-MSE Supervision.** Our proposed configuration utilizing **(MSE, MSE)** for both attention and embedding approximation consistently outperforms other combinations. As shown in Table 26, this setup achieves the highest ACC-B of 0.5047 on BCIC-IV-2A and 0.5867 on BCIC2020-3. The effectiveness of MSE stems from its quadratic penalty on larger deviations which enforces a stricter constraint on the primary rhythmic components. This ensures that

the target model accurately captures original attention and embedding features from source models while preserving the sample-specific importance assigned by the **Fusor**. **2) Sensitivity to Distributional Approximation.** Replacing MSE with **KL Divergence** for attention maps $\mathcal{L}_{\text{Attn}}$ leads to a noticeable performance drop. For instance, on BCIC2020-3, the ACC-B decreases from 0.5867 using **(MSE, MSE)** to 0.5685 with **(KL, MSE)**. Although KL divergence is robust for distribution matching, it may over-smooth the underlying structures within the attention maps, causing the target model to lose fine-grained local dependencies and failing to capture the distinct dynamics prioritized by the **Fusor**. **3) Robustness Across Tasks.** The consistency of the **(MSE, MSE)** configuration across both 4-class BCIC-IV-2A and 5-class BCIC2020-3 tasks suggests that a variance-focused objective is more suitable for the geometry of synchronized EEG representations. By penalizing significant discrepancies more heavily, the dual-MSE objective ensures that the fused knowledge maintains high spatial and temporal fidelity, providing a reliable foundation for multi-scale knowledge fusion.

### J.5. Ablation on Attention Knowledge Unification Strategies

**Ablation Settings.** To evaluate the impact of different synchronization methods for heterogeneous attention, we compare four knowledge unification strategies defined in Appendix F: our default **Block-Diagonal** method, **Global Tiling**, **Row-Broadcasting**, and **Patch-Pooling**.

While the first three approaches establish distinct topological frameworks to map localized interactions into the unified $\mathbb{R}^{CN_P \times CN_P}$ coordinate space required by the *aligner*, **Patch-Pooling** instead compresses the spatial dimension into a localized $\mathbb{R}^{C \times C}$ representation. This comparison examines how different structural priors in $\tilde{\mathbf{K}}_{\text{Attn}}^{(m)}$ influence the efficiency of knowledge fusion. The quantitative results are summarized in Table 27, while Figure 8 illustrates the performance distribution across various EEG metrics.

*Table 27.* Ablation analysis of different attention knowledge unification strategies across EEG datasets.

| Alignment Strategy | BCIC-IV-2A (*4-class*) | | | ATTENTION (*2-class*) | | |
|---|---|---|---|---|---|---|
| | ACC-B | W-F1 | Kappa | ACC-B | AUC-PR | AUROC |
| **Block-Diagonal** | **0.5047** ± 0.0203 | **0.4870** ± 0.0240 | **0.3396** ± 0.0271 | **0.7217** ± 0.0036 | 0.7884 ± 0.0129 | **0.8031** ± 0.0104 |
| Global Tiling | 0.4899 ± 0.0203 | 0.4726 ± 0.0187 | 0.3199 ± 0.0271 | 0.7158 ± 0.0134 | 0.7871 ± 0.0127 | 0.8014 ± 0.0115 |
| Row-Broadcasting | 0.4972 ± 0.0108 | 0.4760 ± 0.0141 | 0.3296 ± 0.0144 | 0.7146 ± 0.0109 | 0.7843 ± 0.0108 | 0.7979 ± 0.0076 |
| Patch-Pooling | 0.4899 ± 0.0203 | 0.4787 ± 0.0205 | 0.3199 ± 0.0271 | 0.7190 ± 0.0092 | **0.7900** ± 0.0092 | 0.8022 ± 0.0112 |

**Note:** Shaded row indicates the default unification strategy (Block-Diagonal).

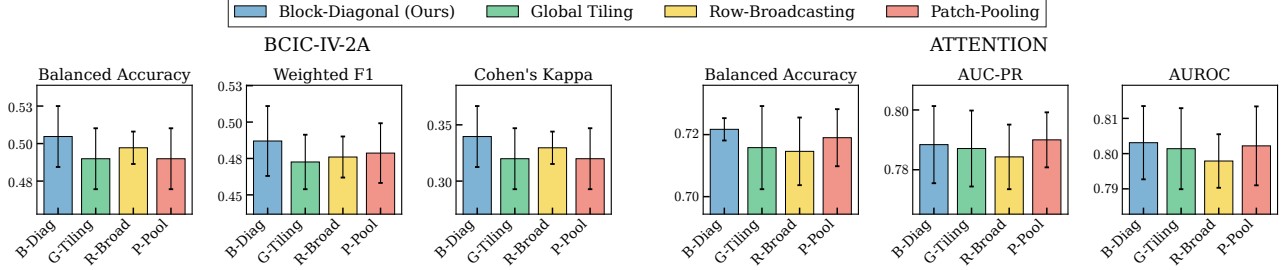

*Figure 8.* Performance comparison of unification strategies across BCIC-IV-2A and ATTENTION datasets.

**Results Analysis.** The choice of alignment strategy determines how localized maps are reconstructed, which directly governs the spatial fidelity of synchronized representations: **1) Locality vs. Global Context.** The **Block-Diagonal** strategy consistently achieves the best performance across both datasets. By isolating intra patch correlations within a sparse format, it preserves the accurate local features extracted by source models. In contrast, **Global Tiling** and **Row-Broadcasting** introduce a rigid spatial invariance prior by repeating or stretching local maps. This leads to a measurable performance drop, such as the ACC-B decrease from 0.5047 to 0.4899 on BCIC-IV-2A, suggesting that excessive smoothing smears fine grained temporal features and reduces discriminative power. **2) Impact of Zero-Initialization.** A key advantage of the **Block-Diagonal** approach is utilizing zero initialization for non local regions instead of negative infinity masking. This maintains a neutral activation background where off-diagonal blocks remain computationally addressable for the

**Aligner**. This design facilitates smoother gradient flow, allowing the model to focus on local rhythmic coupling while preserving the latent capacity to optimize long range dependencies. Although **Patch-Pooling** shows competitive results on the ATTENTION dataset with a peak AUC-PR of 0.7900, its failure on the more complex BCIC-IV-2A task involving four classes highlights its limitations. Collapsing $N_p$ dimensions into a sequence invariant $\mathbb{R}^{C \times C}$ space sacrifices the contextual detail necessary for multi class motor imagery decoding. The overall stability of the **Block-Diagonal** method validates its effectiveness in balancing local integrity with the requirements of cross model knowledge fusion.

### J.6. Ablation on Backbone Fusion Gain and Carrier Selection

**Ablation Settings.** This experiment evaluates the efficacy of the proposed framework by comparing two paradigms: **1) Individual Fine-Tuning (FT)**: Each model $\mathbf{F}^{(m)}$ is directly fine tuned on the target dataset using a standard classification head. **2) Knowledge Fusion (Fusion)**: The same backbone acts as a carrier model $\mathbf{T}$, integrated with our fusion framework to leverage knowledge from the source *model pool* $\mathcal{M}$. We conduct this comparison across three distinct EEG tasks: ISRUC-S3, BCIC-IV-2A, and MentalArithmetic. For each dataset, we highlight the optimal backbone identified by our PARC based selection strategy to verify if the fusion gain remains consistent across different architectural foundations.

**Results Analysis.** Table 28 illustrates the advantages of our framework through two primary insights: **1) Universal Knowledge Synergy**: The **Fusion** paradigm consistently surpasses standalone **FT** across diverse backbones. This improvement demonstrates that integrating knowledge from multiple stages effectively compensates for individual architectural limitations by providing complementary neural information that a single model cannot capture. **2) Validation of Carrier Selection and Synchronization**: Optimal backbones identified by PARC achieve significant performance peaks under the fusion framework, such as CodeBrain reaching a 0.8813 AUROC in MentalArithmetic. This success confirms that our approach effectively synchronizes noncommensurable latent distributions into a unified representation, maximizing the utility of the most compatible source knowledge.

*Table 28.* Performance comparison between individual model fine-tuning (FT) and knowledge fusion (Fusion) across three EEG datasets.

| Source | Metric | ISRUC-S3 (5-class) | | BCIC-IV-2A (4-class) | | MentalArithmetic (2-class) | |
| --- | --- | --- | --- | --- | --- | --- | --- |
| | | **FT** | **Fusion** | **FT** | **Fusion** | **FT** | **Fusion** |
| LaBraM | ACC-B | $0.7895 \pm 0.0082$ | $\mathbf{0.7897} \uparrow \pm 0.0186$ | $0.3453 \pm 0.0089$ | $\mathbf{0.3464} \uparrow \pm 0.0091$ | $0.6840 \pm 0.0673$ | $0.6208 \downarrow \pm 0.0935$ |
| | W-F1 | $0.7618 \pm 0.0120$ | $\mathbf{0.7692} \uparrow \pm 0.0188$ | $0.3110 \pm 0.0449$ | $\mathbf{0.3237} \uparrow \pm 0.0147$ | $0.5619 \pm 0.1650$ | $\mathbf{0.6122} \uparrow \pm 0.1168$ |
| | Kappa | $0.7035 \pm 0.0133$ | $\mathbf{0.7098} \uparrow \pm 0.0233$ | $0.1271 \pm 0.0119$ | $\mathbf{0.1285} \uparrow \pm 0.0531$ | $0.7073 \pm 0.1132$ | $\mathbf{0.7234} \uparrow \pm 0.1067$ |
| CBraMod | ACC-B | $0.6069 \pm 0.0455$ | $\mathbf{0.6383} \uparrow \pm 0.0792$ | $0.4922 \pm 0.0354$ | $\mathbf{0.5047} \uparrow \pm 0.0203$ | $0.7063 \pm 0.0613$ | $\mathbf{0.7486} \uparrow \pm 0.0153$ |
| | W-F1 | $0.5401 \pm 0.0467$ | $0.5179 \downarrow \pm 0.1007$ | $0.4679 \pm 0.0459$ | $\mathbf{0.4870} \uparrow \pm 0.0240$ | $0.6109 \pm 0.0686$ | $0.5304 \downarrow \pm 0.0645$ |
| | Kappa | $0.4463 \pm 0.0528$ | $\mathbf{0.4678} \uparrow \pm 0.1018$ | $0.3229 \pm 0.0473$ | $\mathbf{0.3396} \uparrow \pm 0.0271$ | $0.7979 \pm 0.0500$ | $\mathbf{0.8060} \uparrow \pm 0.0211$ |
| CodeBrain | ACC-B | $0.7013 \pm 0.0217$ | $\mathbf{0.7224} \uparrow \pm 0.0319$ | $0.3747 \pm 0.0233$ | $\mathbf{0.3856} \uparrow \pm 0.0182$ | $0.7174 \pm 0.0144$ | $\mathbf{0.7292} \uparrow \pm 0.0391$ |
| | W-F1 | $0.6616 \pm 0.0209$ | $\mathbf{0.6813} \uparrow \pm 0.0325$ | $0.3007 \pm 0.0295$ | $\mathbf{0.3116} \uparrow \pm 0.0211$ | $0.6417 \pm 0.0507$ | $\mathbf{0.7377} \uparrow \pm 0.0415$ |
| | Kappa | $0.5807 \pm 0.0247$ | $\mathbf{0.6090} \uparrow \pm 0.0374$ | $0.1662 \pm 0.0311$ | $\mathbf{0.1808} \uparrow \pm 0.0242$ | $0.8487 \pm 0.0190$ | $\mathbf{0.8813} \uparrow \pm 0.0201$ |

**Note:** Shaded cells highlight the optimal carrier for each task as determined by the selection method. **Bold** values indicate superior performance of the fusion method over individual fine-tuning.

### J.7. Ablation on Attention Knowledge Extraction Depth

**Ablation Settings.** To investigate the impact of multi-layer feature integration within the knowledge extraction-unification operator $\Psi$, we vary the number of extracted internal layers from the source models. Specifically, we evaluate the set of **last** $k$ **layers** $\mathbb{L}_{\text{sel}} = \{L^{(m)} - k + 1, \ldots, L^{(m)}\}$ for $k \in \{1, 2, 3, 4\}$, where $L^{(m)}$ denotes the terminal layer index of each model $\mathbf{F}^{(m)}$. This setting examines whether incorporating shallower intermediate representations benefits the fusion process. All other hyperparameters and the carrier model $\mathbf{T}$ remain constant to ensure a fair comparison across both BCIC-IV-2A and BCIC2020-3 datasets.

**Results Analysis.** The experimental results indicate that incorporating earlier layers from the source models does not yield proportional performance gains, but rather leads to a degradation in representation quality: **1) High-level Semantic Sufficiency.** The peak performance achieved at $k = 1$ (i.e., extracting only the terminal layer $\mathbb{L}_{\text{sel}} = \{L^{(m)}\}$) suggests that the final layer of the pre-trained foundation models already captures sufficient semantic information for cross-model knowledge fusion. Since these backbones are optimized for generalized neural representations, the terminal layer provides

the most refined expertise. In contrast, increasing $k$ to 4 (including more intermediate layers) results in a measurable drop in ACC-B from 0.5867 to 0.5733 on BCIC2020-3, indicating that shallower features may introduce lower-level redundant information that interferes with high-level knowledge synthesis. **2) Noise Propagation and Integration Complexity.** The decline in Kappa scores as more layers are integrated reflects the challenge of managing multi-scale feature noise. EEG signals possess a low signal-to-noise ratio, and shallower layers often contain more raw, unrefined temporal dynamics. By focusing on the most abstract and discriminative manifolds ($k = 1$), the framework achieves an optimal trade-off between knowledge richness and the robustness required for stable knowledge fusion.

*Table 29.* Ablation analysis of last $k$ layers *attention knowledge* extraction depth within the $\Psi$ module.

| Depth ($k$) | BCIC-IV-2A (4-class) | | | BCIC2020-3 (5-class) | | |
| --- | --- | --- | --- | --- | --- | --- |
| | ACC-B | W-F1 | Kappa | ACC-B | W-F1 | Kappa |
| 1 | **0.5047** $\pm$ 0.0203 | **0.4870** $\pm$ 0.0240 | **0.3396** $\pm$ 0.0271 | **0.5867** $\pm$ 0.0115 | **0.5869** $\pm$ 0.0115 | **0.4833** $\pm$ 0.0144 |
| 2 | 0.4976 $\pm$ 0.0125 | 0.4787 $\pm$ 0.0147 | 0.3312 $\pm$ 0.0166 | 0.5840 $\pm$ 0.0201 | 0.5840 $\pm$ 0.0202 | 0.4800 $\pm$ 0.0252 |
| 3 | 0.4901 $\pm$ 0.0120 | 0.4728 $\pm$ 0.0184 | 0.3201 $\pm$ 0.0266 | 0.5797 $\pm$ 0.0143 | 0.5800 $\pm$ 0.0137 | 0.4746 $\pm$ 0.0179 |
| 4 | 0.4899 $\pm$ 0.0203 | 0.4726 $\pm$ 0.0187 | 0.3199 $\pm$ 0.0271 | 0.5733 $\pm$ 0.0215 | 0.5734 $\pm$ 0.0218 | 0.4667 $\pm$ 0.0269 |

**Note:** Shaded row indicates the default depth ($\mathbb{L}_{\text{sel}} = \{L^{(m)}\}$) used in our EmBrace framework.

## J.8. Ablation on Bridge Projection Dimension Scaling

**Ablation Settings.** The *bridge* mechanism Bridge$^{(m)}$ projects embedding spaces from different source models into a unified dimension. We test how the dimension $d_{\text{align}} \in \{128, 200, 256, 512\}$ affects knowledge fusion. This experiment evaluates if the projection can successfully map $\mathbf{K}_{\text{Emb}}^{(m)}$ from various sources into a single space while preserving local patterns. We aim to find a dimension that maintains source information without making training unstable or inefficient for the target model $\mathbf{T}$.

**Results Analysis.** Table 30 shows that performance follows a non-monotonic trend on both BCIC-IV-2A and BCIC2020-3 datasets. The configuration with $d_{\text{align}} = 256$ performs best, reaching an ACC-B of 0.5047 and 0.5867. Increasing the dimension from 128 to 256 improves results because a smaller space cannot adequately store the complex dynamics from sources. However, expanding the dimension to 512 leads to a decline in performance. For instance, the Kappa metric on BCIC2020-3 drops to 0.4661. An oversized $d_{\text{align}}$ introduces noise and dilutes the supervision signal. Furthermore, a dimension of 512 significantly increases the computational cost for the target model $\mathbf{T}$. We choose $d_{\text{align}} = 256$ as the optimal balance for our model.

*Table 30.* Ablation analysis of different *bridge* projection dimensions $d_{\text{align}}$ in the *bridge* mechanism.

| $d_{\text{align}}$ | BCIC-IV-2A (4-class) | | | BCIC2020-3 (5-class) | | |
| --- | --- | --- | --- | --- | --- | --- |
| | ACC-B | W-F1 | Kappa | ACC-B | W-F1 | Kappa |
| 128 | 0.4816 $\pm$ 0.0324 | 0.4517 $\pm$ 0.0407 | 0.3088 $\pm$ 0.0432 | 0.5631 $\pm$ 0.0109 | 0.5631 $\pm$ 0.0109 | 0.4539 $\pm$ 0.0136 |
| 200 | 0.4708 $\pm$ 0.0229 | 0.4462 $\pm$ 0.0355 | 0.2944 $\pm$ 0.0305 | 0.5667 $\pm$ 0.0162 | 0.5667 $\pm$ 0.0161 | 0.4583 $\pm$ 0.0202 |
| 256 | **0.5047** $\pm$ 0.0203 | **0.4870** $\pm$ 0.0240 | **0.3396** $\pm$ 0.0271 | **0.5867** $\pm$ 0.0115 | **0.5869** $\pm$ 0.0115 | **0.4833** $\pm$ 0.0144 |
| 512 | 0.4936 $\pm$ 0.0398 | 0.4706 $\pm$ 0.0552 | 0.3248 $\pm$ 0.0530 | 0.5729 $\pm$ 0.0056 | 0.5729 $\pm$ 0.0058 | 0.4661 $\pm$ 0.0069 |

**Note:** Shaded row indicates the default projection dimension ($d_{\text{align}} = 256$) used in our main experiments.

## K. Discussion

### K.1. Implication

Our framework introduces a new paradigm for EEG knowledge integration with the following implications: **1) Cross-Architecture Knowledge Integration.** Current EFMs like LaBraM or CBraMod focus on single encoders. However, no single architecture fits all EEG paradigms. EmBrace shifts the focus to active knowledge integration. By reconciling structural differences, our framework combines multiple source models into one target model, capturing cognitive processes that single models often overlook. **2) Structural Alignment and Weight Inheritance.** We use the PARC metric to select the best base model as a structural anchor. Since EFMs replace decoders during fine-tuning, prediction-level alignment

is impossible. EmBrace performs fusion at the structural level using embeddings and attention maps from source models. Initializing the target model $\mathbf{T}$ with full pre-trained weights ensures a strong representational starting point. **3) Biologically-Informed Dynamic Adaptation.** EmBrace incorporates explicit physiological priors to guide the fusion process. By encoding brain state transitions, the system dynamically adjusts the contribution of different source models. This ensures the fusion follows biological signals rather than just statistical patterns, making BCI systems more robust across diverse paradigms.

### K.2. Limitation

Our current framework has certain limitations that warrant further investigation: **1) Model Pool Dependency.** Fusion quality depends on the selected source models. If source models have large domain gaps or poor representational quality relative to the target task, the supervision signals $\mathcal{L}_{\text{Attn}}$ and $\mathcal{L}_{\text{Emb}}$ may introduce noise. **2) Hyperparameter Sensitivity.** As detailed in Appendix I and Table 22, model performance is sensitive to the loss weights $\lambda$ and $\gamma$. The optimal balance between ground-truth and fusion knowledge varies across datasets, and finding these parameters currently requires manual grid search.

### K.3. Future Work

Based on these observations, our future research will focus on the following directions: **1) Automated Hyperparameter Optimization.** We plan to develop adaptive methods such as FAMO (Liu et al., 2023) to tune $\lambda$ and $\gamma$ dynamically during training to reduce reliance on manual grid searches and improve robustness across diverse datasets. **2) Interpretability of Fusion Weights.** We will focus on linking the dynamic weights $\mathbf{w}$ more closely to neurophysiological rhythms to enhance transparency and provide insights into how architectures prioritize specific brain patterns. **3) Cross-Modality Knowledge Fusion.** We intend to extend EmBrace by incorporating source models pre-trained on a broader range of neurophysiological signals such as MEG and fNIRS. By leveraging multi-modal foundation models like BrainOmni (Xiao et al., 2026), future iterations can unify heterogeneous patterns from electromagnetic fields to hemodynamic responses via joint training. This would enable the framework to extract universal representational embeddings that capture the underlying neural language across diverse sensors and recording modalities.

