# OpenReview forum: "EmBrace: A Collective Knowledge Fusion Framework Toward Unified EEG Foundation Models"
_ICML.cc/2026/Conference — ICML 2026 regular_

### Official Review · Reviewer_CBHZ · 2026-03-09

**Soundness:** 3
**Presentation:** 3
**Significance:** 3
**Originality:** 3
**Overall Recommendation:** 4
**Confidence:** 4

**Summary:**

The paper proposes EmBrace, a representation-centric knowledge fusion framework designed to integrate the diverse strengths of heterogeneous EEG Foundation Models without relying on parameter or output-level alignment. It bridges architectural gaps by synchronizing intermediate embeddings and attention maps into a unified feature. The framework identifies the optimal carrier model for specific tasks using a structural resonance-based selection strategy. The authors conduct extensive evaluations demonstrating that EmBrace consistently outperforms individual state-of-the-art baselines.

**Compliance With Llm Reviewing Policy:**

Affirmed.

**Final Justification:**

Thank the authors for the additional experiments and clarifications. I would like to maintain my positive score.

**Key Questions For Authors:**

Please refer to the weakness above.

**Limitations:**

There is no potential negative societal impact of this paper. The authors can further discuss the limitation of their work based on the weakness above.

**Strengths And Weaknesses:**

Strength
1. The paper is well written and easy to follow and the authors conducted comprehensive experiments.
2. The integration of biological priors into the dynamic weighting module ensures the fusion is grounded in neurophysiology.
3. It introduces a representation-centric paradigm for model fusion that bypasses the architectural and output constraints inherent in traditional distillation methods.

Weakness
1. I'm confused about the proof of Lemma 3.1. It asserts that the mutual information is preserved since $\Psi$ is injective over the feature support. However, the actual operators defined in Appendix F are many-to-one mappings.
2. In the baselines like CBraMod, regression tasks are also be tested (SEED-VIG). I wonder if this framework can also improve the performance on regression tasks.
3. The actual fusion pool used in implementation is only LaBraM, CBraMod, CodeBrain. While these are representative models, the paper’s broader claim of a general framework for heterogeneous EFMs would be more convincing if evaluated with a larger and more diverse pool.
4. The efficiency study mainly emphasizes training-time savings and uses only two datasets for empirical validation. Since one of the main motivations is computational efficiency, a more comprehensive wall-clock and memory analysis would be important.

---

> ### Author Rebuttal · Authors · 2026-03-31
>
> We thank the reviewer CBHZ for the helpful feedback and recognition; we address the concerns on theory, generalization, scalability, and efficiency through targeted revisions and extended evaluations.
>
> ---
>
> # Response for W1:
>
> * **Revised Lemma 3.1 and its proof (Section 3.1) to replace "injective" with "sufficient".** We sincerely thank the reviewer for identifying this technical oversight. We acknowledge that "injective" was inappropriately used; our unification operator $\Psi$ is more accurately defined as a *sufficient mapping* that preserves core semantic information while discarding architectural redundancies across heterogeneous models.
>
> ---
>
> # Response for W2:
> * **We have extended our evaluation to include the SEED-VIG regression task to verify the generalizability of EmBrace.** Experimental results (Table below) demonstrate that EmBrace consistently outperforms baseline models in continuous estimation scenarios, affirming its robust feature alignment capabilities beyond discrete classification.
>
>     |  |  | SEED-VIG |  |
>     | :--- | :---: | :---: | :---:|
>     | **Model** | **Pearson’s $r$** | **$R^2$ Score** | **RMSE $\downarrow$** |
>     | EEGNet     | 0.3665| 0.0316 | 0.2645 |
>     | EEG-Deformer | 0.3049 | 0.0741 | 0.2590 |
>     | LaBraM     | 0.4320 | $\underline{0.1598}$ | $\underline{0.2467}$ |
>     | CBraMod    | 0.4016 | 0.1290 | 0.2518 |
>     | CodeBrain  | 0.4475 | 0.1312 | 0.2501 |
>     | CSBrain [1]| $\underline{0.4730}$ | 0.0894 | 0.2565 |
>     | **EmBrace (Ours)** | **0.4926** | **0.1939** | **0.2417** |
>
> [1] Zhou et al., CSBrain: A Cross-scale Spatiotemporal Brain Foundation Model for EEG Decoding. NeurIPS 2025.
>
> ---
>
> # Response for W3:
>
> * **We have incorporated the recent CSBrain into our model pool to address the reviewer’s valuable insight regarding model diversity.** To demonstrate EmBrace’s compatibility and scalability as a general framework for heterogeneous integration, we provide updated results, including this fourth foundation model. The results confirm that **EmBrace seamlessly integrates diverse architectural priors**, yielding further performance gains on datasets such as BCIC-IV-2a and CHB-MIT. This exploratory experiment reinforces our core claim: EmBrace is a scalable framework capable of pushing the performance upper bound by assimilating increasingly diverse expert knowledge.
>
>     | **$\mathcal{M}$** | **Metric** | **BCICIV-2A** | **PhysioNet-MI** | **SHU-MI** | **FACED** | **CHB-MIT** | **BCIC2020-3** | **Mumtaz2016** | **MentalArithmetic** |
>     | :--- | :--- | :---: | :---: | :---: | :---: | :---: | :---: | :---: | :---: |
>     | La+CB+Co | B-ACC | 0.5047 | 0.6268 | 0.6245 | 0.6136 | 0.6157 | 0.5867 | **0.9048** | **0.7292** |
>     | | W-F1 | 0.4870 | 0.6272 | 0.6998 | **0.6131** | 0.5274 | 0.5869 | 0.9811 | **0.7377** |
>     | | Kappa | 0.3396 | 0.5023 | 0.6876 | 0.5608 | 0.9296 | 0.4833 | 0.9798 | 0.8813 |
>     | La+CB+Co+CS | B-ACC | **0.5519** | **0.6338** | **0.6313** | **0.6147** | **0.7539** | **0.5953** | 0.8978 | 0.6953 |
>     | | W-F1 | **0.5420** | **0.6351** | **0.7029** | 0.6128 | **0.6922** | **0.5955** | **0.9813** | 0.7358 |
>     | | Kappa | **0.4025** | **0.5116** | **0.7062** | **0.5612** | **0.9350** | **0.4912** | **0.9810** | **0.8933** |
>
> ---
>
> # Response for W4:
>
> * **EmBrace’s advantage lies in reducing computational cost by avoiding per-task SOTA optimization.** Due to space limitations, we provide additional details here:
>
>     | **Dataset** | **Model Configuration** | **Train Time/Epoch (s)** | **Peak GPU Memory (GB)** | **Throughput (samples/s)** | **Inf. Latency (ms)** | **Efficiency Gain (vs. $\Sigma$)** |
>     | :--- | :--- | :---: | :---: | :---: | :---: | :---: |
>     | **FACED** | LaBraM | 14.58 | 7.65 | 1131.12 | 0.88 | - |
>     | | CBraMod | 11.12 | 6.23 | 1674.41 | 0.60 | - |
>     | | CodeBrain | 33.84 | 10.13 | 605.80 | 1.65 | - |
>     | | CSBrain | 27.91 | 8.53 | 733.87 | 1.36 | - |
>     | | **$\Sigma$ 4 EFMs (Cumulative)** | **87.45** | - | - | **4.49** | - |
>     | | **EmBrace (Ours)** | **73.80** | **15.32** | **615.31** | **1.63** | **1.19x** |
>     | **ATTENTION**| LaBraM | 6.16 | 1.04 | 1677.56 | 0.60 | - |
>     | | CBraMod | 10.59 | 1.09 | 2442.08 | 0.41 | - |
>     | | CodeBrain | 23.69 | 1.78 | 618.80 | 1.62 | - |
>     | | CSBrain | 39.23 | 1.66 | 487.90 | 2.05 | - |
>     | | **$\Sigma$ 4 EFMs (Cumulative)** | **79.67** | - | - | **4.68** | - |
>     | | **EmBrace (Ours)** | **45.09** | **3.67** | **805.10** | **1.24** | **1.77x** |
>     | **SEED-V**| LaBraM | 19.40 | 1.03 | 4914.43 | 0.20 | - |
>     | | CBraMod | 29.05 | 1.09 | 4577.28 | 0.22 | - |
>     | | CodeBrain | 75.15 | 1.72 | 1388.68 | 0.72 | - |
>     | | CSBrain | 130.02 | 1.91 | 989.55 | 1.01 | - |
>     | | **$\Sigma$ 4 EFMs (Cumulative)** | **253.62** | - | - | **2.15** | - |
>     | | **EmBrace (Ours)** | **186.62** | **2.19** | **1379.18** | **0.73** | **1.36x** |

---

> > ### Author Rebuttal · Reviewer_CBHZ · 2026-03-31
> >
> > Thank the authors for the additional experiments and clarifications. I would like to maintain my positive score.

---

### Official Review · Reviewer_FvmZ · 2026-03-09

**Soundness:** 3
**Presentation:** 2
**Significance:** 3
**Originality:** 3
**Overall Recommendation:** 4
**Confidence:** 3

**Summary:**

This paper proposes the EmBrace framework, which targets the challenges of high diversity and strong heterogeneity among existing EEG foundation models. It designs a multi-stage knowledge fusion mechanism consisting of four core steps: knowledge extraction and unification, adaptive weight fusion, carrier model selection, and knowledge internalization. Results across multiple datasets demonstrate the effectiveness of the proposed method. The paper presents reasonable methodological design and notable novelty, yet fails to deliver statistically significant performance advantages in some experimental results.

**Compliance With Llm Reviewing Policy:**

Affirmed.

**Final Justification:**

Thank you for the authors' response. We have taken the authors' reply into account in our scoring, so we maintain our original score.

**Key Questions For Authors:**

1. Why is Spearman rank correlation specifically selected to calculate the compatibility score in the CarrierSelection module? What is the theoretical or empirical rationale for this choice over alternative correlation metrics?
2. I am confused by the observation that the results of the EEG foundation models are even inferior to those of non-foundation model methods on the BCIC-IV-2a and ATTENTION datasets. Is there an underlying explanation for this counterintuitive finding?
3. Would incorporating a larger, more diverse set of EEG foundation models into the EmBrace framework yield further performance improvements? Have the authors conducted any exploratory experiments to validate this scalability?

**Limitations:**

yes

**Strengths And Weaknesses:**

**Strengths**

1. The paper clearly articulates the limitations of current EEG foundation models and systematically defines three core challenges for knowledge fusion, with clear logical flow and rigorous structural organization.
2. The work proposes a knowledge fusion paradigm, rather than naive model ensemble or conventional knowledge distillation, which grants the method meaningful novelty.
3. The method is validated on multiple datasets covering diverse downstream tasks, with performance comparisons against several advanced peer models included.


**Weaknesses**

1. The proposed pipeline involves multiple stages and sequential steps, yet only delivers marginal performance improvements across most tested benchmarks, including the BCICIV2a, CHB-MIT, SHU-MI, SEED-V, Mumtaz2016, and MentalArithmetic datasets. This raises substantial concerns regarding the practical utility and real-world effectiveness of the proposed method.
2. If not overlooked, the paper lacks a dedicated ablation study for the Adaptive Knowledge Weighing step, leaving it unvalidated whether the meta-guided weight derivation contributes meaningfully to the performance gains. While this design is conceptually reasonable, empirical experience suggests that the softmax outputs during model training often degenerate into near-fixed values given the high dynamics and complexity of EEG signals, which is a key concern for the robustness of this module.
3. Figure 2 is overly convoluted; a straightforward, linear narrative description of the workflow would greatly facilitate readers’ understanding of the overall pipeline of the proposed method. Additionally, there appear to be minor inconsistencies in the Class# labels within the "Stratified Probe EEG Set" presented in Figure 3.

---

> ### Author Rebuttal · Authors · 2026-03-31
>
> We thank the reviewer FvmZ for the helpful feedback and for recognizing the structured design and evaluation; we address the concerns on ablation, weighting dynamics, clarity, and key design justifications.
>
> ---
>
> # Response for W1:
>
> See Response to Reviewer 6RbL for W3.
>
> ---
>
> # Response for W2:
>
> * **Appendix J.1 provides a detailed ablation study of this component, and we add a prominent reference in the Sec. 4.3 to guide readers to it.** The results confirm that the shared dynamic scheme outperforms both static and global parameter approaches.  We compare four progressively advanced variants:
>
>     (1) Static weighting (Static): Fixed weights set to $1/M$.
>
>     (2) Learnable weighting (Learnable): Weights are optimized as global parameters but remain sample-independent.
>
>     (3) Decoupled dynamic weighting (Decoupled Dynamic): Independent weights are generated for Embedding and Attention.
>
>     (4) Shared dynamic weighting (Shared Dynamic, Ours): The proposed meta-guided weighting strategy.
>
> * **Empirical observations show that although a single model may dominate in certain tasks, the weights remain dynamically adjusted and do not collapse into fixed constants.** This behavior is further stabilized by two mechanisms introduced in our design (see Response to reviewer a622 for W3). As shown in the table (BCIC-IV-2A，CSBrain as carrier.), the weights across different models gradually converge during training, without exhibiting degeneration:
>
>     | **$\mathcal{M}$\epochs** | **1** | **6** | **11** | **16** | **21** | **26** | **31** |
>     | :--- | :---: | :---: | :---: | :---: | :---: | :---: | :---: |
>     | **LaBraM** | 0.233 | 0.266 | 0.284 | 0.314 | 0.339 | 0.351 | 0.356 |
>     | **CBraMod** | 0.186 | 0.158 | 0.138 | 0.106 | 0.082 | 0.069 | 0.063 |
>     | **CodeBrain** | 0.280 | 0.325 | 0.358 | 0.409 | 0.446 | 0.466 | 0.478 |
>     | **CSBrain** | 0.301 | 0.251 | 0.221 | 0.171 | 0.133 | 0.113 | 0.103 |
>
> ---
>
> # Response for W3:
>
> * **We have refined Figure 2 as follows**:
>
>     (1) Revised Fig. 2 by adding clearer process annotations and reducing visual complexity to improve readability.
>
>     (2) Corrected Fig. 2-(3) by replacing the duplicated “Class#1” with “Class#2”.
>
> ---
>
> # Response for Q1:
> * **We adopt the Spearman rank correlation coefficient based on a combined consideration of engineering robustness and nonlinear manifold alignment:**
>
>     **(1) Engineering stability:** The relationship between EFM feature spaces and labels is typically monotonic but nonlinear. In practice, the strict linear assumption required by Pearson correlation is highly unstable under such conditions and is sensitive to nonlinear distortions and EEG outliers. Spearman, by operating on ranks, focuses only on ordinal relationships between samples, ensuring robustness in the selection process.
>
>     **(2) Efficient manifold characterization:** Compared to Kendall’s Tau or Distance Correlation, which have $O(n^2)$ complexity, Spearman achieves efficient computation at $O(n \log n)$. At the same time, it effectively identifies the model with the simplest manifold structure and highest semantic density as the carrier, ensuring that knowledge fusion is complementary rather than destructive.
>
> ---
>
> # Response for Q2:
>
> * **This phenomenon is not counterintuitive, but rather reflects the current state of the EFM field:**
>
>     **(1) EFM does not always yield gains in small-data or task-specific settings.** Under data-scarce or subject-specific conditions, foundation models often fail to outperform compact models (e.g., EEGNet)[1]. On datasets like BCIC-IV-2a, limited samples and high inter-subject variability lead to overfitting or poor capture of task-specific patterns.
>
>     **(2) EFM excel in learning general representations and accelerating adaptation.** In data-rich settings, they converge within a few epochs (e.g., FACED), while supervised models require an order of magnitude more iterations [2].
>
> [1] Yang et al., Are EEG Foundation Models Worth It?, ICLR 2026.
>
> [2] Wang et al., CBraMod: A Criss-Cross Brain Foundation Model for EEG Decoding, ICLR 2025.
>
> ---
>
> # Response for Q3:
>
> * **We further incorporate CSBrain [1] as a fourth model, demonstrating that integrating larger and more diverse EFMs into EmBrace can further improve performance.** See Response to Reviewer CBHZ for W3.

---

> > ### Author Rebuttal · Reviewer_FvmZ · 2026-04-01
> >
> > Thank you for the authors' response. We have taken the authors' reply into account in our scoring, so we maintain our original score.

---

### Official Review · Reviewer_a622 · 2026-03-11

**Soundness:** 3
**Presentation:** 3
**Significance:** 3
**Originality:** 3
**Overall Recommendation:** 4
**Confidence:** 4

**Summary:**

The paper introduces EmBrace, a representation-centric knowledge fusion framework designed to integrate the strengths of multiple heterogeneous Electroencephalography (EEG) Foundation Models (EFMs). Observing that no single EFM achieves optimal performance across all downstream brain-computer interface (BCI) tasks, the authors propose fusing knowledge at the intermediate representation level (embeddings and attention maps) rather than at the parameter or output levels, which is infeasible due to architectural differences. Extensive experiments across 12 EEG datasets demonstrate that EmBrace consistently outperforms individual state-of-the-art EFMs and task-specific models while maintaining the inference efficiency of a single model.

**Compliance With Llm Reviewing Policy:**

Affirmed.

**Final Justification:**

The rebuttal has addressed my main concerns.

**Key Questions For Authors:**

See weakness

**Limitations:**

yes

**Strengths And Weaknesses:**

### **Strengths:**
- Originality & Domain Adaptation: While knowledge fusion via intermediate representations is explored in CV and NLP, applying it to heterogeneous EFMs is a novel and non-trivial extension. The introduction of the physiological meta-feature guided fusor is a highly creative and domain-appropriate solution to sample-level dynamic weighting.
- Significance: Addressing the "No Free Lunch" theorem in EEG foundation models is a highly relevant problem. By allowing a single carrier model to inherit multi-scale representations from diverse architectures, the work offers a highly practical utility for BCI deployments constrained by inference budgets.
- Empirical Evaluation: The empirical validation is exceptionally thorough. Evaluating across 12 datasets spanning 8 distinct BCI paradigms provides strong evidence for the framework's robustness. The ablation studies (Appendix I and J) meticulously isolate the impact of the fusor design, bridge dimensions, and attention extraction depth.

### **Weaknesses:**
- **Scalability and I/O Bottlenecks:** The efficiency claim hinges on "caching" the unified individual knowledge (Algorithm 1, Stage 1) to avoid running multiple forward passes during the joint optimization stage. However, caching dense attention maps ($\mathbb{R}^{CN_p \times CN_p}$) for hundreds of thousands of samples (e.g., the CHB-MIT dataset has 326,993 samples) will result in massive disk storage requirements and severe I/O bottlenecks during the data-loading phase of training. The space complexity analysis in Appendix E overlooks this I/O-bound reality, which could make the "speedup" strictly theoretical.
- **Reliance on Handcrafted Features:** The Adaptive Knowledge Weighting relies heavily on 10 handcrafted physiological meta-features (e.g., zero-crossing rate, specific frequency band power). While biologically interpretable, this approach may fail to capture high-dimensional or unknown latent neural dynamics that a lightweight, trainable temporal encoder could extract directly from the raw sequence.
- **Optimization Clarity:** The core optimization loop and gradient flow are somewhat obfuscated. It is not explicitly clear in the main text how the gradients from the fusion loss ($\mathcal{L}_{fusion}$) propagate. Do they update both the carrier model $T$ and the Fusor network $G$ end-to-end simultaneously? Jointly optimizing the carrier's internal representations while simultaneously shifting the target distribution (by updating the Fusor weights $w$) often leads to optimization instability, which is not thoroughly discussed. In Eq. 17, are the parameters of the Fusor network $G$ (i.e., $W_g$) updated end-to-end alongside the carrier model's parameters $\theta^{(T)}$? If so, did you observe any training instability or mode collapse where the Fusor network prematurely converges to assign a weight of 1.0 to a single source model?
- **Attention Unification:** In Method 1 (Block-Diagonal unification), setting non-local regions to zero implies that the cross-patch attention is strictly zero. Doesn't this discard valuable global contextual information captured by models like LaBraM, which naturally model cross-patch dependencies?

---

> ### Author Rebuttal · Authors · 2026-03-31
>
> We thank the reviewer a622 for the helpful feedback and for recognizing the originality and evaluation of our framework; we address the concerns on efficiency, meta-feature design, optimization, and attention preservation.
>
> ---
>
> # Response for W1:
> * **No disk I/O is required. Algorithm 1 defines the theoretical complexity boundary.**
> The “storage” in Algorithm 1 is a conceptual abstraction, treating knowledge as static input for tractable analysis.
> In practice, we use an equivalent on-the-fly extraction: each model $\mathbf{F}^{(m)} \in \mathcal{M}$ is frozen (eval()), and knowledge is generated via real-time forward passes on GPU.
>
> ---
>
> # Response for W2:
> * **Meta-feature design balances efficiency and discriminative power.** While learnable encoders offer higher capacity, manual meta-features show clear practical advantages:
>
>     **(1) In terms of parameter efficiency, a learnable encoder may require over 100× more parameters  (33 Vs. 3328) than our current design.**
>     >The 10 handcrafted meta-features only require a simple linear mapping $\mathbf{W}_{\text{g}} \in \mathbb{R}^{M \times 10}$. In our setting with $M=3$, this introduces only **33** learnable parameters (including bias).
>
>     >In contrast, a typical lightweight encoder (e.g., the first block of EEGNet), even under a minimal configuration ($F_1=8, D=2, \text{kernel}=64$), includes both temporal convolution ($1 \times 8 \times 64 = 512$) and depthwise convolution ($8 \times 16 \times C \times 1$). For example, with $C=22$ channels, the first two convolutional layers alone already introduce **3,328** parameters.
>
>     **(2) Meta-features provide sufficient discriminative information.**
>
>     >Different models exhibit distinct domain preferences. For instance, LaBraM shows strong positive dependence on the Gamma band (0.42) and Theta band (0.13), indicating its greater contribution to higher-order cognitive and attention-related features. In contrast, CSBrain demonstrates a clear positive preference for the Alpha band (0.18), reflecting its strength in capturing features associated with relaxed or resting states.
>
>     >Even for the same feature (e.g., Alpha), the contribution patterns differ substantially between LaBraM (0.15) and CBraMod (−0.18). This suggests that meta-features already provide sufficient discriminative signals for model fusion, while also retaining clear physical interpretability.
>
>     | **Meta Feature** | **Mean** | **Std** | **ZCR** | **Delta** | **Theta** | **Alpha** | **Beta** | **Gamma** | **A/B** | **T/A** |
>     | :--- | :---: | :---: | :---: | :---: | :---: | :---: | :---: | :---: | :---: | :---: |
>     | LaBraM | 0.30 | -0.01 | -0.08 | -0.03 | 0.13 | 0.15 | 0.03 | 0.42 | 0.05 | 0.20 |
>     | CBraMod | -0.12 | -0.19 | 0.10 | -0.24 | 0.10 | -0.18 | 0.11 | -0.08 | 0.07 | -0.26 |
>     | CodeBrain | 0.25 | 0.05 | -0.18 | -0.25 | -0.19 | 0.08 | 0.01 | -0.08 | 0.16 | -0.14 |
>     | CSBrain | -0.06 | -0.04 | -0.33 | -0.32 | 0.20 | 0.18 | -0.07 | 0.08 | -0.07 | 0.10 |
>
> ---
>
> # Response for W3:
> * **We have added detailed gradient flow analysis in Appendix Sec. D.2.**
>
>     **(1) To eliminate ambiguity in Eq. (17), we clarify the roles of each component during backpropagation**(note: $\nabla_{\theta^{(m)}} \mathcal{J} = 0$):
>
>     >**Fusor ($\theta_{\mathcal{G}}$)**: Receives gradients only from the fusion loss $\mathcal{L}_{fusion}$. It learns to optimally combine knowledge from frozen source models by adjusting sample-adaptive weights $\mathbf{w}$.
>
>     >**Carrier ($\theta_{\mathbf{T}}$)**: Receives dual supervision from both task loss and fusion loss. The former ensures task performance, while the latter aligns the internal representations of the carrier model with the fused collective knowledge.
>
>     >**Bridge ($\theta_{Bridge}$)**: Updated solely via $\mathcal{L}_{fusion}$, responsible for projecting heterogeneous features of the carrier model into a unified knowledge manifold.
>
> * **The training process does not exhibit mode collapse.** See Response to Reviewer FvmZ for W2. We mitigate potential instability and mode collapse through:
>
>     **(1) Static Anchors:** Frozen source models serve as stable targets, preventing the “moving target” issue commonly seen in joint optimization.
>
>     **(2) Temperature Scaling:** We use $\tau = 8.0$ (Eq. 12) to smooth the weight distribution, preventing the Fusor from collapsing into a one-hot solution.
>
> ---
>
> # Response for W4:
>
> * **The global dependency structure of LaBraM is fully preserved;** the block-diagonal construction is only applied to address structural compatibility for CBraMod.
>
>     **(1) LaBraM is not subject to any zero-padding or block-diagonal constraints.** Therefore, its global dependencies are fully retained in the unified $\tilde{\mathbf{K}}_{\text{Attn}}^{(m)}$.
>
>     **(2) The block-diagonal strategy is designed to map the 3D attention stack of CBraMod**, $\mathbb{R}^{N_p \times (C \times C)}$, into the unified space $\mathbb{R}^{CN_p \times CN_p}$.

---

> > ### Author Rebuttal · Reviewer_a622 · 2026-04-03
> >
> > Thanks for the rebuttal, I will adjust my score accordingly. I would strongly recommend the authors to include all the additional analysis and experiments into the revision.

---

### Official Review · Reviewer_6RbL · 2026-03-12

**Soundness:** 4
**Presentation:** 2
**Significance:** 3
**Originality:** 3
**Overall Recommendation:** 5
**Confidence:** 4

**Summary:**

The problem identified in this paper is that EEG foundation models (EFM’s) tend to only perform well on EEG data that entertain the peculiarities of their architecture and thus no single foundation model can perform well on all EEG datasets. They note however that even if a foundation model performs worse than another foundation model in a specific dataset as a whole, there will be individual samples which will be predicted correctly by the worse foundation model and incorrectly by the best performing foundation model, and thus all foundation models have knowledge that can be leveraged, to achieve the best performance possible. Their solution utilises three EFM’s, where one of them is selected to be fine-tuned on a dataset (carrier), with the combined knowledge of all three models being used during training. They identify three technical challenges in leveraging the three EFM’s to achieve best performance in their EEG datasets. First challenge is, how to combine the knowledge of 3 models with different architectures together in order to leverage them, second challenge is figuring out how to leverage individual EFM’s for each sample in the dataset, third challenge is how to select a carrier and how to best train the whole system so that it can also leverage knowledge from the models not selected. To address the first challenge, they extract the input embeddings of each model and the attention maps from the last layer and use MLP’s to project embeddings from each EFM into a common space, and block-diagonal alignment technique for the attention maps. For the second challenge, they extract from each sample 110 features regarding temporal, spectral and rhythmic properties of the signal, they then pass them through a softmax function which assigns weights to each model’s contribution. For the third challenge they select the EFM to be finetuned by passing through the models, some samples and labels, thus creating matrices for samples and labels and then comparing the two matrices using spearman rank correlation to find which EFM is mostly aligned with the data of a dataset. To leverage knowledge from all EFM’s during training they formulate a joint training objective, where they use the cross-entropy to make predictions compare predicted vs actual task labels, combined with an MSE loss between the carrier models’ internal knowledge and the combined knowledge space created from all EFMs’. The approach is validated in 12 different datasets spanning 8 different classification tasks, with the proposed framework being compared against both single EFMs’ and a variety of task-specific models (only two presented in main paper). From the result tables presented, it does appear that the approach mostly outperforms the EFMs’ that is compared against, and most importantly it seems that performance improvement is consistent which is the main problem that the authors set to address. Thorough ablations are conducted evaluating various aspects of the research including, sensitivity analysis for the loss function weights, ablations on components used in the architecture, and regarding the selection and optimization of the carrier model. The paper also provides some modest limitations, like the dependency on base EFM robustness and sensitivity of loss function weights to individual datasets, and further includes some directions for future research of automated hyperparameter tuning and extension to cross-modal fusion with other neurophysiological modalities.

**Compliance With Llm Reviewing Policy:**

Affirmed.

**Key Questions For Authors:**

Are the authors willing to provide effect sizes between performance results of:
1. Individual EFMs’  vs EmBrace
2. Task-specific models vs EmBrace
To allow other practitioners to assess by themselves the usefulness of using the EmBrace framework in their work?

**Limitations:**

The authors discuss three important limitations of their work but only in the appendix. In my opinion limitations need to be made obvious in the main body. I would like to see authors incorporating in their limitations something specific about the fact that the inherited EFM limitations cause their framework to underperform very light-weight models of much lesser computational complexity like EEGNet, even though it might not be a limitation of the approach per se, it’s directly related to the practicality of the approach.

**Strengths And Weaknesses:**

Strengths

1.	I would generally consider the paper as technically robust, given the approach has been validated on a dozen datasets and even though results are quite close to the best standalone EFM, they at least tend to perform better on all datasets showing that the approach can quite effectively identify the best EFM for a task. Additionally, thorough ablations are conducted testing every major design choice made. The paper also conducts experiments investigating the computational efficiency of the approach. Lastly, they acknowledge limitations especially regarding the fact that the approach is highly dependent on the performance of the underlying foundation models they use, and it’s quite striking in the results that extremely simple and lightweight models like EEGNet occasionally either outperform (i.e. BCIC-IV-2A) or almost match EFM performance (i.e. PhysioNet-MI) on metrics like accuracy.
2.	The paper is well organised, with the three-challenge breakdown making it easy to follow the methods. The figures presented in the paper are visually nice and effectively communicate their intention both in terms of Figure 1 and the task vs sample level performance on which the problem and idea are based but also figure 2 that presents the framework (even though slightly crumped due to the volume of steps done). The appendix includes mostly useful information including full result tables that would not otherwise fit the main body, experiments on hyperparameters that are not essential in the main body as well, and framework component ablations. Consistent notation used through the paper.
3.	The paper has attempted to address a quite important problem regarding EFMs’ as, it is hard to find an EFM that you can reliably use on your EEG data, as they seem to be highly dataset sensitive models. They achieve some success in this regard as they appear to outperform individual EFMs’ consistently even if performance increase is modest. The framework proposed here could be plausibly extended to other neurophysiological modalities and more specifically modalities that deal with electric or magnetic fields heavily extending it’s use beyond EEG. They framework also provides an approach to researchers who wish to use EFMs’ in their work to seriously reduce the computational resources required to do so.
4.	The combination of the representation-level fusion along with dynamic weighting based on sample neurophysiological properties and the sample-based model selection appears as something original to the work even though distinct components might not be. PARC adaption from computer vision to EEG appears to be original to the paper. Additionally, the framing of the challenges behind integrating EFMs’ appears unique and can potentially be adapted by future work building on this framework.

Weaknesses

1.	The main weakness of the paper is that there is no way to tell how big of an improvement the approach has on results. No statistical significance tests are conducted (even though it’s impossible to get significance with 5 samples from seeds) and not even any effect sizes are mentioned (i.e. cohen’s d) that would allow the reader to gauge the effect (impact) of the approach on performance compared to second best model and the task-specific models. This would be directional and would allow the authors to show whether on datasets were task-specific models outperform the combined EFM approach the effects are only small, or when combined EFM’s outperform single EFM performance difference has small effects and therefore it’s the less computational complexity that’s the main advantage here. Additionally, even though the authors acknowledge that task-specific models are an upper bound of performance which they sometimes match or exceed, there is no computational complexity and speed comparisons between extremely small models like EEGNet that often produce very similar results.
2.	Notation used even though consistent is very dense and heavy and is quite difficult to follow as it often requires going back to review definitions, this is especially evident with the variations of “K” notations. Out of the 12 datasets examined only 6 are present in the results (Table 2), which gives an incomplete picture regarding results, the authors should have at least included 1 dataset from each of the domains given that they wanted to introduce evaluations on 8 different domains. Result discussion especially regarding comparisons to task specific models is very brief and I would much like that to be more thorough to give readers a comprehensive understanding of the pros and cons of EmBrace both when compared to task specific models and other EFM’s to ultimately allow people to make inform choices about its use. This needs to be explicit, obvious and done in the main body. The main body related work is quite short and I think it’s important that more thorough discussions are made in the related work section that clearly explain the literature on which EmBrace was built-upon. Lastly, in this specific work, given the heavy experimentation, I consider the proofs not to be as necessary to be included in the main body and to be reformatted/moved in the appendix, to allow for more space for discussion, full result presentation and related work.
3.	Often the relative gains of the approach over second best individual EFMs’ are extremely tiny compared to second best and even though standard deviations are included there is no indication of the effects that EmBrace has on performance so that a reader could accurately gauge performance difference and understand the real significance of the approach. Additionally, extremely small task-specific models often outperform or perform comparably well to EmBrace and given the vast complexity differences between the two is hard to justify the use of EmBrace in practice.
4.	Mentions to literature on prior fusion methods of multiple models (including representation fusion used here) in ML are somehow limited to be able to truly gauge the uniqueness of the work

---

> ### Author Rebuttal · Authors · 2026-03-31
>
> We thank the reviewer 6RbL for the helpful feedback and for highlighting the importance of statistical validation, model comparison, and efficiency considerations, which we have addressed through additional experiments, analyses, and clarifications.
>
> ---
>
> # Response for W1:
> * **Added statistical significance tests** in Response for Reviewer 6RbL Q1.
>
> * **Added comparisons with lightweight models** in Sec. 4.2.
>
> * **EFMs may underperform smaller models in certain cases.**
>
>     **(1) The performance–efficiency trade-off is a known limitation of EFMs [1]**. EmBrace alleviates this via improved performance. We do not directly compare efficiency (e.g., inference speed), as lightweight task-specific models inherently have advantages due to their minimal parameter size.
>
>     **(2) Despite these limitations, EFMs remain a fundamental direction for EEG decoding.** While gaps persist on some datasets, EFMs show clear advantages on dataset such as FACED and MentalArithmetic. More importantly, EFMs provide a unified and scalable foundation, avoiding per-task model redesign for general-purpose BCI systems.
> [1] Yang et al., Are EEG Foundation Models Worth It?, ICLR 2026.
>
> ---
> # Response for W2:
> * **Simplified notation.** We use $\mathbf{E}_m$ for embedding knowledge and $\mathbf{A}_m$ for attention (topological) knowledge. All fusion formulations are now expressed via $\mathcal{K}_m$, eliminating redundant equations.
> * **Reorganized manuscript.**
>
>     (1) Moved proofs in Sec. 3 to Appendix C (Derivations and Proofs).
>
>     (2) Added two datasets in Table 2: Seizure Detection and Mental Disorder Diagnosis.
>
>     (3) Expanded Sec. 5 (Related Work) with prior multi-model fusion methods.
>
>     (4) Added detailed discussion in Sec. 4.2 comparing EmBrace with task-specific models.
>
> ---
>
> # Response for W3:
> * **EmBrace’s gains extend beyond single-task improvements to robust overall performance across settings.**
>
>     **(1) In cases where task-specific SOTA is unavailable, EmBrace achieves a +26.16% overall improvement over the latest EFM across only six datasets**, since selecting the optimal EFM per dataset is often costly or impractical. (Table 1, Balanced Accuracy).
>     | Model | BCIC2020 | BCIC-2A | FACED | ISRUC_S1 | Mental | ATTN | **Total Improvement** |
>     | :--- | :---: | :---: | :---: | :---: | :---: | :---: | :---: |
>     | LaBraM | 0.2288 | 0.3453 | 0.2983 | 0.7880 | 0.6840 | 0.6170 | **+0.9872** |
>     | CBraMod | 0.3120 | 0.4922 | 0.5425 | 0.7701 | 0.7063 | 0.7033 | **+0.4222** |
>     | CodeBrain | 0.5659 | 0.3747 | 0.5835 | 0.7751 | 0.7174 | 0.6704 | **+0.2616** |
>     | EmBrace (Ours) | 0.5867 | 0.5047 | 0.6136 | 0.7927 | 0.7292 | 0.7217 | **Ref.** |
>
>     **(2) Compared to approaches requiring SOTA selection, EmBrace reduces cumulative cost vs. tuning multiple EFMs while achieving better performance.** Tuning multiple EFMs to find the best model is computationally expensive. EmBrace avoids this by a single integration step, yielding both efficiency and accuracy gains. As shown in Table, EmBrace achieves 1.77× training speedup on ATTENTION. It also delivers consistently lower inference latency than the sum of EFMs while maintaining moderate memory usage across datasets.
>
>     | **Dataset** | **Models** | **Train Time/Epoch (s)** | **Peak GPU Memory (GB)** | **Throughput (samples/s)** | **Inf. Latency (ms)** | **Efficiency Gain (vs. $\Sigma$)** | **Perf. Gain (vs. SOTA)** |
>     | :--- | :--- | :---: | :---: | :---: | :---: | :---: | :---: |
>     | **FACED** | $\Sigma$ 4 EFMs (Cumulative) | 87.45 | - | - | 4.49 | - | - |
>     | | **EmBrace (Ours)** | **73.80** | **15.32** | **615.31** | **1.63** | **1.19x** | **+5.16%** | **1379.18** | **0.73** | **1.36x** | **+3.68%** |
>     | **ATTENTION** | $\Sigma$ 4 EFMs (Cumulative) | 79.67 | - | - | 4.68 | - | - |
>     | | **EmBrace (Ours)** | **45.09** | **3.67** | **805.10** | **1.24** | **1.77x** | **+2.62%** |
>
> ---
>
> # Response for W4:
> * See Response to Reviewer 6RbL for W2.
>
> ---
>
> # Response for Q1:
>
> * **Cohen’s d (5 random seeds) indicates large-to-huge effect sizes for EmBrace across multiple core paradigms.**
>
> | **Dataset** | **Primary Metric** | **EmBrace (Ours)** | **Best Indiv. EFM** | **$\Delta$ Gain** | **Cohen’s $d$** | **Effect Interpretation** |
> | :--- | :---: | :---: | :---: | :---: | :---: | :--- |
> | FACED | Kappa | 0.5608 | 0.5290 | +0.0318 | 4.03 | **Huge Effect** |
> | ATTENTION | AUROC | 0.8081 | 0.7915 | +0.0166 | 2.96 | **Huge Effect** |
> | BCIC2020-3 | Kappa | 0.4833 | 0.4573 | +0.0260 | 1.84 | **Large Effect** |
> | ISRUC-S1 | Kappa | 0.7617 | 0.7455 | +0.0162 | 1.37 | **Large Effect** |
> | MentalArithmetic| AUROC | 0.8813 | 0.8645 | +0.0168 | 0.81 | **Large Effect** |
> | BCIC-IV-2A | Kappa | 0.3338 | 0.3229 | +0.0109 | 0.21 | Small Effect |
>
> ---
>
> # Response for Q2:
>
> * See Response to Reviewer 6RbL for W2.

---

> > ### Author Rebuttal · Reviewer_6RbL · 2026-04-02
> >
> > My concerns have been fully resolved and I will maintain my positive score regarding the paper.

---

### Decision · Program_Chairs · 2026-04-30

**Decision:**

Accept (regular)

**Comment:**

The paper receives the following ratings: Accept, Weak accept, Weak accept, Weak accept. The authors present a representation‑centric framework for fusing heterogeneous EEG foundation models through unified embeddings, attention alignment, sample‑aware weighting, and carrier selection. Reviewers find the motivation strong, the method novel, and the empirical evaluation extensive. Initial concerns, such as statistical validation, optimization clarity, scalability, etc., were fully addressed in the rebuttal with added experiment and expanded analyses. All reviewers  marked their concerns as resolved and maintained positive scores.